# A New Theoretical Perspective on Data Heterogeneity in Federated Averaging

## Abstract

In federated learning, data heterogeneity is the main reason that existing theoretical analyses are pessimistic about the convergence error caused by local updates. However, empirical studies have shown that more local updates can improve the convergence rate and reduce the communication cost when data are heterogeneous. This paper aims to bridge this gap between the theoretical understanding and the practical performance by providing a theoretical analysis for federated averaging (FedAvg) with non-convex objective functions from a new perspective on data heterogeneity. Identifying the limitations in the commonly used assumption of bounded gradient divergence, we propose a new assumption, termed the heterogeneity-driven Lipschitz assumption, which characterizes the fundamental effect of data heterogeneity on local updates. In the convergence analysis, we use the heterogeneity-driven Lipschitz constant and the global Lipschitz constant to substitute the widely used local Lipschitz constant and we show that our assumptions are weaker than those used in the literature. Based on the new assumption, we derive novel convergence bounds for both full participation and partial participation, which are tighter compared to the state-of-the-art analysis of FedAvg. This result can also imply that more local updates can improve the convergence rate even when data are highly heterogeneous. Further, we discuss some insights of the proposed heterogeneity-driven Lipschitz assumption. One interesting finding is that for some quadratic objective functions, we can identify a region where FedAvg (also known as local SGD) can outperform mini-batch SGD even when the gradient divergence is arbitrarily large.

## 1 Introduction

Federated learning (FL) has emerged as an important technique for locally training machine learning models over geographically distributed workers. It has advantages in improving training efficiency and preserving data privacy. In this paper, we consider the following optimization problem in FL:

$$\min_{\mathbf{x}} \left\{ f(\mathbf{x}) := \frac{1}{N} \sum_{i=1}^{N} F_i(\mathbf{x}) \right\}, \tag{1}$$

where $N$ is the number of workers; $F_i(\mathbf{x})$ is the expected loss function of worker $i$[1] given by

$$F_i(\mathbf{x}) := \mathbb{E}_{\xi_i \sim \mathcal{D}_i}[\ell(\mathbf{x}; \xi_i)], \tag{2}$$

where $\ell(\cdot)$ is the loss function, $\xi_i$ is the random data sample on worker $i$, and $\mathcal{D}_i$ is the data distribution on worker $i$. In addition, we use $\mathcal{D}$ to denote the global data distribution. In FL, each worker performs $I > 1$ local iterations using its local dataset to reduce the communication cost, which is called *local updates*. Federated averaging (FedAvg), also known as local stochastic gradient descent (local SGD), is one of the most popular algorithms to solve the above optimization problem (McMahan et al., 2017). In addition to FedAvg, a number of FL algorithms (Yu et al., 2019a; Karimireddy et al., 2020; Reddi et al., 2020; Li et al., 2020b; Wang et al., 2020a;b) have been proposed, whereas the core mechanism, local updates, is still the foundation of FL. Nevertheless, existing theoretical analyses are pessimistic on the convergence error caused by local updates. In

---

[1]The objective function can be extended to weighted average by multiplying each local objective function by a possibly distinct constant.

Table 1: Estimated $L_h$, $\tilde{L}$, $L_g$ with the MNIST dataset. Heterogeneity is shown by the percentage of data on each worker that are not uniformly sampled from the global dataset. Results for linear regression can be found in Appendix C.

| Obj. Function | Two-layer Neural Network | | | |
|---|---|---|---|---|
| Heterogeneity | 25% | 50% | 75% | 100% |
| $\tilde{L}$ | $127.62 \pm 10.21$ | $130.97 \pm 11.67$ | $134.24 \pm 12.23$ | $141.92 \pm 12.78$ |
| $L_h$ | $0.35 \pm 0.06$ | $0.82 \pm 0.11$ | $1.66 \pm 0.23$ | $2.36 \pm 0.29$ |
| $L_g$ | $122.23 \pm 9.75$ | $122.23 \pm 9.75$ | $122.23 \pm 9.75$ | $122.23 \pm 9.75$ |

particular, some works (Yu et al., 2019a;b) showed that convergence error of FL algorithms grows as the number of local updates increases, which is inconsistent with empirical studies. This will be explained in detail as follows.

**There is a gap between the theoretical understanding and the experimental results.** Unlike the centralized SGD running on a single machine, where the gradients are directly sampled from the global data distribution $\mathcal{D}$, the local gradients in FedAvg are sampled from the local data distributions $\{\mathcal{D}_i\}$, which are often highly heterogeneous (Kairouz et al., 2021). This can deteriorate FL's performance since the local models could drift to different directions during local updates (Zhao et al., 2018; Karimireddy et al., 2020). Therefore, a common understanding is that local SGD can have a larger convergence error than that of centralized SGD due to local updates. Existing theoretical analyses for non-convex objective functions (Yu et al., 2019a;b; Wang & Joshi, 2019; Yang et al., 2020) confirmed this intuition and showed that the convergence error caused by local updates grows fast when the number of local updates $I$ increases. However, in practice, local updates have been successfully applied (Li et al., 2020a; Niknam et al., 2020; Rieke et al., 2020) and showed superior experimental performance compared to mini-batch SGD (each worker performing $I = 1$ local iteration per round) (McMahan et al., 2017; Lin et al., 2020). This means that increasing $I$ can improve the convergence rate and reduce the communication cost when data are highly non-IID in practice. This inconsistency between the pessimistic theoretical results and the good experimental results for the local updates implies that the existing theoretical analysis may overestimate the error caused by local updates. It is indeed challenging to show theoretically when local SGD ($I > 1$) can outperform mini-batch SGD ($I = 1$) (Woodworth et al., 2020a;b).

**Although local models could drift to different directions, the average of local models can still be close to the centralized model.** To the best of our knowledge, the only metric of data heterogeneity in existing works (Yu et al., 2019b; Wang & Joshi, 2019; Karimireddy et al., 2020; Woodworth et al., 2020b) is gradient divergence ($\zeta$), which characterizes the difference between the expected local gradient $\nabla F_i(\mathbf{x})$ of worker $i$ and the expected global gradient $\nabla f(\mathbf{x})$. As shown in Figure 1, the intuition behind the gradient divergence is that when $\zeta$ is large, the difference between local gradients and the global gradient is large. Then after multiple local iterations, the local models will drift to different directions. Previous theoretical results based on the gradient divergence show that when $\zeta$ is large, $I$ has to be small to avoid the divergence of the FL algorithms. However, in FL, the final output is the global model on the server, which is the average of local models after local updates. As shown in Figure 1, although $\zeta$ is large, the averaged model $\hat{\mathbf{x}}^{r,k}$ can still be close to the centralized model $\mathbf{x}_c^{r,k}$ obtained using centralized SGD. This means that the convergence error caused by local updates is close to zero. While $\zeta$ successfully characterizes the variance among local gradients, it cannot capture the difference

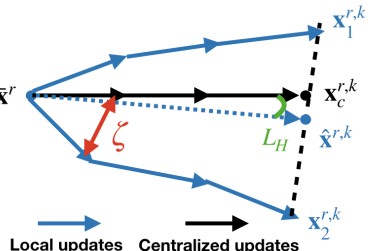

Figure 1: An illustrative comparison between local updates and centralized updates. $\bar{\mathbf{x}}^r$ is the global model at $r$th round. The local models after $k$ local iterations at the $r$th round are denoted by $\mathbf{x}_1^{r,k}$ and $\mathbf{x}_2^{r,k}$. The average of $\mathbf{x}_1^{r,k}$ and $\mathbf{x}_2^{r,k}$ is $\hat{\mathbf{x}}^{r,k}$. The centralized model after $k$ centralized iterations is denoted by $\mathbf{x}_c^{r,k}$. It can be seen that $\zeta$ shows the difference between $\mathbf{x}_c^{r,k}$ and $\mathbf{x}_i^{r,k}$, $i = 1, 2$ and $L_h$ shows the difference between $\mathbf{x}_c^{r,k}$ and $\hat{\mathbf{x}}^{r,k}$.

between the averaged model and the centralized model. Consequently, relying solely on the gradient divergence in convergence analysis can lead to an overestimation of the convergence error caused by

local updates. To obtain a better convergence upper bound, it is necessary to introduce a new metric which can characterize the difference between the averaged model and the centralized model.

To address the inconsistency between the theory and the practice, we introduce a new metric $L_h$, referred to as the *heterogeneity-driven Lipschitz constant*. As shown in Figure 1, the proposed metric $L_h$ captures the difference between the averaged model and the centralized model, which cannot be characterized by $\zeta$. In our analysis, we use the heterogeneity-driven Lipschitz constant $L_h$ and the global Lipschitz constant $L_g$ to substitute the widely used local Lipschitz constant $\tilde{L}$. This is based on our important observation that $\tilde{L}$ is affected by data heterogeneity, which is neglected by previous theoretical results. In the literature (Yu et al., 2019b; Yang et al., 2020; Khaled et al., 2020), $\tilde{L}$ is used to characterize the smoothness of the gradients for all local objective functions under any degree of the data heterogeneity. However, as shown in Table 1, $\tilde{L}$ increases fast as the percentage of non-IID data increases. We use $L_h$ to characterize the information on data heterogeneity contained in $\tilde{L}$ and we use $L_g$ to show the smoothness. We will show that using $L_h$ and $L_g$ to substitute $\tilde{L}$ can make the assumptions weaker.

**Contribution of this paper.** In this paper, we reveal the fundamental effect of the data heterogeneity on FedAvg by introducing a new perspective shown by $L_h$, the *heterogeneity-driven Lipschitz constant* in Assumption 4.2. In particular, our main contributions are as follows. (1) Using the new assumptions, which are proved to be weaker than those in the literature, we develop a novel analysis for FedAvg with general non-convex objective functions, which shows that if $L_h$ is small enough, even for a large $\zeta$, the convergence error caused by local updates can still be small so that a large $I$ can be used to obtain a good convergence performance and to reduce communication costs. (2) Our analysis can incorporate partial participation where only a subset of workers are sampled to perform the local updates in each round. We show that with partial participation, increasing $I$ can still improve the convergence rate. (3) We discuss the significance of the proposed $L_h$, by which, we identify a region where local SGD can outperform mini-batch SGD for some quadratic objective functions. (4) Our theoretical results are validated using experiments.

## 2 RELATED WORKS

FedAvg, also known as local SGD, was first proposed in McMahan et al. (2017) and there have been a considerable amount of works analyzing the convergence rate of local SGD with non-convex objective functions including FedAvg (Haddadpour, Farzin et al., 2019; Yu et al., 2019b; Reddi et al., 2020). There is also a line of work focusing on the partial participation (Yang et al., 2020), compression and quantization (Jiang & Agrawal, 2018; Richtárik et al., 2021) in local SGD. However, in these works, the gradient divergence is the only metric of data heterogeneity. Thus, it is hard to show that local updates can improve the convergence rate when data are highly heterogeneous (Woodworth et al., 2020a;b) while our paper show that more local updates can improve the convergence rate. Based on local SGD, some works aim to overcome the data heterogeneity by introducing additional control variates, such as SCAFFOLD (Karimireddy et al., 2020). However, the communication overhead is larger compared to local SGD. There are two papers (Wang et al., 2022; Das et al., 2022) closely related to our work, which also aim to find new assumptions that can better characterize the effect of data heterogeneity in local SGD. However, both works cannot capture the information on the data heterogeneity contained in the local Lipschitz constant as shown in our paper so their convergence upper bounds can be worse compared to ours. A detailed discussion for related works can be found in Appendix A.1.

## 3 PRELIMINARIES

In FedAvg, each round is composed of the local update phase and the global update phase. The global model is initialized as $\bar{\mathbf{x}}^0$. At the start of round $r$, the server distributes the global model $\bar{\mathbf{x}}^r$ to all workers. During the local update phase, each worker updates its local model with the local learning rate $\gamma$ and the stochastic gradients sampled from their own local data distribution $\mathcal{D}_i$,

$$\mathbf{x}_i^{r,k+1} = \mathbf{x}_i^{r,k} - \gamma \mathbf{g}(\mathbf{x}_i^{r,k}; \zeta_i), \tag{3}$$

where $\mathbf{x}_i^{r,k}$ is the local model at the $r$th round and $k$th iteration. For simplicity, we use $\mathbf{g}_i(\cdot)$ to denote the gradient $\mathbf{g}(\cdot; \zeta_i)$. In addition, $\bar{\mathbf{g}}(\cdot)$ denotes the gradient sampled from the global dataset $\mathcal{D}$. We assume that the local stochastic gradient is an unbiased estimate of the expected local gradient $\mathbb{E}\left[\mathbf{g}_i(\mathbf{x}_i^{r,k}) \big| \mathbf{x}_i^{r,k}\right] = \nabla F_i(\mathbf{x}_i^{r,k})$. After $I$ local iterations, worker $i$ sends the local model update at

the $r$th round $\Delta_i^r := \bar{\mathbf{x}}^r - \mathbf{x}_i^{r,I}$ to the server. During the global update phase, the server updates the global model using the following equality:

$$\bar{\mathbf{x}}^{r+1} = \bar{\mathbf{x}}^r - \eta \cdot \frac{1}{N} \sum_{i=1}^N \Delta_i^r, \tag{4}$$

where $\eta$ is the global learning rate. Let $\hat{\mathbf{x}}^{r,k}$ be the "virtual" averaged model during the local update phase and

$$\hat{\mathbf{x}}^{r,k+1} := \frac{1}{N} \sum_{i=1}^N \mathbf{x}_i^{r,k+1} = \hat{\mathbf{x}}^{r,k} - \gamma \cdot \frac{1}{N} \sum_{i=1}^N \mathbf{g}_i(\mathbf{x}_i^{r,k}), \tag{5}$$

where $k \in \{0, 1, 2, \ldots, I-1\}$. Note that the virtual model $\hat{\mathbf{x}}^{r,k}$ may not be observed in the system, and is mainly used for the theoretical analysis. We define $\mathbf{x}_c^{r,k}$ as the model obtained by applying centralized updates[2] at $k$th iteration of $r$th round given the averaged model $\hat{\mathbf{x}}^{r,k}$, which means that the gradient is sampled from the global data distribution $\mathcal{D}$. Specifically,

$$\mathbf{x}_c^{r,k+1} := \hat{\mathbf{x}}^{r,k} - \gamma \bar{\mathbf{g}}(\hat{\mathbf{x}}^{r,k}), \tag{6}$$

where $\mathbb{E}\left[\bar{\mathbf{g}}(\hat{\mathbf{x}}^{r,k})\right] = \nabla f(\hat{\mathbf{x}}^{r,k})$.

The following assumptions are widely used in the literature for the analysis of algorithms including FedAvg (Karimireddy et al., 2020; Yu et al., 2019b; Khaled et al., 2020; Wang et al., 2020a).

**Assumption 3.1** (Local Lipschitz Gradient)**.**

$$\|\nabla F_i(\mathbf{x}) - \nabla F_i(\mathbf{y})\| \le \tilde{L} \|\mathbf{x} - \mathbf{y}\|, \forall \mathbf{x}, \mathbf{y}, i. \tag{7}$$

There are also some works (Khaled et al., 2020) assuming that Lipschitz gradient condition holds for each data sample $\|\nabla\ell(\mathbf{x};\xi) - \nabla\ell(\mathbf{y};\xi)\| \le L' \|\mathbf{x} - \mathbf{y}\|, \forall \mathbf{x}, \mathbf{y}, \xi$, which is stronger and can imply local Lipschitz gradient condition.

**Assumption 3.2** (Bounded Stochastic Gradient Variance)**.**

$$\mathbb{E}\left[\|\mathbf{g}_i(\mathbf{x}) - \nabla F_i(\mathbf{x})\|^2\right] \le \sigma^2, \forall i, \mathbf{x}. \tag{8}$$

**Assumption 3.3** (Bounded Gradient Divergence)**.**

$$\|\nabla F_i(\mathbf{x}) - \nabla f(\mathbf{x})\|^2 \le \zeta^2, \forall i, \mathbf{x}. \tag{9}$$

Assumption 3.3 is often the only metric of data heterogeneity in the literature (Yu et al., 2019a; Wang & Joshi, 2019), where it was shown that there is a term $O(\gamma^2 \tilde{L}^2 I^2 \zeta^2)$ in the convergence upper bound. This means that the gradient divergence ($\zeta$) and the number of local updates ($I$) are coupled, and the error caused by $\zeta$ grows fast as $I$ increases and the effect of $I^2\zeta^2$ is amplified by $\tilde{L}^2$. In this paper, we find that this result can be pessimistic since it can be seen from Table 1 that $\tilde{L}$ can be very large, which means that the error caused by $I^2\zeta^2$ can become much larger due to the large $\tilde{L}^2$. In the next section, we will solve this problem using our new assumption and analysis.

## 4 MAIN RESULTS

In this section, we present the convergence upper bound for non-convex objective functions using the proposed new assumption. We also consider the partial participation in the analysis. We summarize the technical novelties and provide proofs for all theorems and propositions in Appendix B.

In the literature, three classes of assumptions on *stochastic gradient variance*, *gradient divergence* and *smoothness* are often made for theoretical analysis (Yu et al., 2019b; Wang et al., 2020a; Khaled et al., 2020). We keep Assumption 3.2 for stochastic gradient variance and Assumption 3.3 for gradient divergence. Assumptions 4.1 and 4.2 will replace Assumption 3.1. In Section 5, we will show that Assumptions 4.1 and 4.2 are weaker than Assumption 3.1.

**Assumption 4.1** (Global Lipschitz Gradient)**.** The global objective function $f(\mathbf{x})$ satisfies

$$\|\nabla f(\mathbf{x}) - \nabla f(\mathbf{y})\| \le L_g \|\mathbf{x} - \mathbf{y}\|, \forall \mathbf{x}, \mathbf{y}. \tag{10}$$

---

[2]Note that the model $\mathbf{x}_c^{r,k}$ is different from the model obtained by applying the centralized updates from the beginning of the algorithm. We use this for ease of analysis, and leave the consideration of the "actual" centralized model for future work.

In our analysis, the Lipschitz gradient condition is only needed for the global objective function instead of for each local objective function as in Assumption 3.1 or for each data sample as in Khaled et al. (2020).

**Assumption 4.2** (Heterogeneity-driven Lipschitz Condition on Averaged Gradients). There exists a constant $L_h \geq 0$ such that $\forall \mathbf{x}_i$,

$$\left\| \frac{1}{N} \sum_{i=1}^{N} \nabla F_i(\mathbf{x}_i) - \nabla f(\bar{\mathbf{x}}) \right\|^2 \leq \frac{L_h^2}{N} \sum_{i=1}^{N} \|\mathbf{x}_i - \bar{\mathbf{x}}\|^2, \tag{11}$$

where $\bar{\mathbf{x}} = \frac{1}{N} \sum_{i=1}^{N} \mathbf{x}_i$ and $L_h$ is referred to as the heterogeneity-driven Lipschitz constant.

We consider Assumption 4.2 as a new perspective on data heterogeneity for the following reasons. First, $L_h$ can be used to characterize the convergence error caused by local updates. In particular, we will show that $\tilde{L}$ can be replaced by $L_h$ in the convergence error caused by local updates in the literature. Second, unlike Assumption 3.3, $L_h$ can characterize the difference between averaged model and the centralized model. This difference captures the real impact of data heterogeneity as discussed in Section 1. We will discuss these new perspectives of Assumption 4.2 and $L_n$ in detail in this section and in Section 5. Next, we present the theoretical results for full participation.

**Theorem 4.3** (General Non-convex Objective Functions). *Assuming Assumptions 3.2, 3.3, 4.1, 4.2 hold, when $\gamma \leq \frac{1}{30(L_h + L_g)I}$ and $\gamma\eta \leq \frac{1}{4IL_g}$, then after $R$ rounds of FedAvg, we have*

$$\min_{r \in [R]} \mathbb{E} \|\nabla f(\bar{\mathbf{x}}^r)\|^2 = \mathcal{O}\left( \underbrace{\frac{\mathcal{F}}{\gamma\eta RI} + \frac{\gamma\eta L_g \sigma^2}{N}}_{\text{error caused by SGD}} + \underbrace{\gamma^2 L_h^2 (I-1)^2 \zeta^2 + \gamma^2 L_h^2 (I-1)\sigma^2}_{\text{error caused by local updates}} \right), \tag{12}$$

*where $\mathcal{F} := f(\mathbf{x}_0) - f^*$.*

**An improved bound by using new assumption.** In (12), the stochastic gradient variance in the error caused by SGD depends on $L_g$ while the error caused by local updates depends on $L_h$. It can be observed that, in Yu et al. (2019b;a); Yang et al. (2020), the variance in the error caused by SGD is $\mathcal{O}(\frac{\gamma\eta\tilde{L}\sigma^2}{N})$, and the error caused by local updates is $\mathcal{O}(\gamma^2 \tilde{L}^2 (I-1)^2 \zeta^2 + \gamma^2 \tilde{L}^2 (I-1)\sigma^2)$, where $\tilde{L}$ is substituted by $L_g$ and $L_h$, respectively, in (12). As shown by the experimental results in Table 1, both $L_h$ and $L_g$ are smaller than $\tilde{L}$. In Section 5, we also prove theoretically that $L_h$ and $L_g$ are smaller than $\tilde{L}$. In particular, $L_h$ can be far less than $\tilde{L}$. Therefore, existing theoretical results overestimate both the error caused by SGD and the error caused by local updates while the convergence upper bound using new assumption is better.

**New insights about the effect of data heterogeneity.** It can be observed that in the error caused by local updates, both $\zeta^2$ and $\sigma^2$ are multiplied by $L_h$. *A key message is that when $\zeta^2$ is large, as long as $L_h^2$ is small enough, the error caused by local updates can still be small.* Since $L_h$ and $\zeta$ characterize the effect of the data heterogeneity in different perspectives, we show that it is possible that $L_h = 0$ while $\zeta$ can be arbitrarily large by providing an example in Section 5. In this case, no matter how large $\zeta$ is, the convergence error of local SGD is the same as that of centralized SGD, which means that $I$ can be arbitrarily large and only one aggregation is sufficient. Moreover, when $L_h = 0$, we can see that the impacts of $\gamma$ and $\eta$ on the convergence upper bound are the same since the error caused by local updates is zero and the error caused by SGD is a function of $\gamma\eta$. In this case, the two-sided learning rates may not help and only a single learning rate, e.g., let $\eta = 1$, suffices to achieve the desired convergence upper bound.

It is worth noting that although $L_h$ increases with the percentage of heterogeneous data, $L_h$ can still be small even if the percentage of heterogeneous data is large as shown by the experimental results for the two-layer neural network in Table 1 and the experimental results for CNN in Section 6. The following corollary shows that more local iterations can improve the convergence rate.

**Corollary 4.4.** *With $\gamma\eta = \sqrt{\frac{\mathcal{F}N}{RIL_g\sigma^2}}$, there exists $\gamma \leq \frac{1}{30(L_h + L_g)I}$ such that*

$$\min_{r \in [R]} \mathbb{E} \|\nabla f(\bar{\mathbf{x}}^r)\|^2 = \mathcal{O}\left( \sqrt{\frac{\mathcal{F}L_g\sigma^2}{RIN}} + \frac{\zeta^2 + \sigma^2/I}{RIN} \right). \tag{13}$$

From (13), it can be seen that FedAvg achieves *linear speedup* in the number of iterations $RI$ with respect to the total number of workers $N$ since the convergence rate is given by $\mathcal{O}\left(\frac{1}{\sqrt{RIN}} + \frac{1}{RIN}\right)$. If we keep the product of $R$ and $I$, the convergence rate will be the same. This means that we can choose a large $I$ and a small $R$ such that the communication cost can be reduced. From the second term $\mathcal{O}\left(\frac{\zeta^2 + \sigma^2/I}{RIN}\right)$, it can be observed that although $\zeta$ can be large, more local updates can still improve the convergence rate.

**Analysis for Partial Participation.** We also use the new assumption to derive the convergence upper bound for partial participation. At each round, $M$ workers are uniformly sampled with replacement. The result provides the insights into the relationship between local updates and partial participation. It is worth noting that the technique for partial participation in existing works cannot be directly applied in our analysis since the Lipschitz gradient (see Assumption 3.1) is often assumed for each local objective function in the literature. Therefore, we develop new techniques to incorporate the partial participation using $L_h$ and $L_g$, which can be found in Appendix B.

**Theorem 4.5** (Partial Participation). *Consider uniformly sampling $M$ ($1 \leq M \leq N$) workers in each round of FedAvg. Assuming Assumptions 3.2, 3.3, 4.1, 4.2 hold, when $\gamma \leq \frac{1}{30(L_h + L_g)I}$ and $\gamma\eta \leq \frac{1}{4IL_g}$, after $R$ rounds of FedAvg, we have*

$$\min_{r \in [R]} \mathbb{E}\left\|\nabla f(\bar{\mathbf{x}}^r)\right\|^2 = \mathcal{O}\Bigg(\underbrace{\frac{\mathcal{F}}{\gamma\eta RI} + \frac{\gamma\eta L_g \sigma^2}{M}}_{\text{error caused by SGD}} + \underbrace{\frac{\gamma\eta L_g I \zeta^2}{M}}_{\text{error caused by p.p.}} + \underbrace{\gamma^2 L_h^2 (I-1)^2 \zeta^2 + \gamma^2 L_h^2 (I-1)\sigma^2}_{\text{error caused by local updates}}\Bigg),$$

(14)

*where "p.p." means partial participation.*

Compared with Theorem 4.3, there are two differences in the convergence bound. First, the variance term in the error caused by SGD depends on $M$. This means that more workers sampled in each round can reduce the variance. Second, there is an extra term $\frac{\gamma\eta L_g I \zeta^2}{M}$ in the convergence bound for partial participation, which denotes the error caused by partial participation. In the literature (Yang et al., 2020), this term is often multiplied by $\tilde{L}$. In (14), this term depends on $L_g$ and not on $L_h$. This means that a small $L_h$ cannot reduce the error caused by partial participation, which can be shown explicitly by the following corollary.

**Corollary 4.6.** *Consider uniformly sampling $M$ workers during each round in FedAvg. With $\gamma\eta = \sqrt{\frac{M\mathcal{F}}{L_g IR(\sigma^2 + I\zeta^2)}}$, there exists $\gamma \leq \frac{1}{30(L_h + L_g)I}$ such that*

$$\min_{r \in [R]} \mathbb{E}\left\|\nabla f(\bar{\mathbf{x}}^r)\right\|^2 = \mathcal{O}\left(\sqrt{\frac{\mathcal{F}L_g \zeta^2}{RM}} + \sqrt{\frac{\mathcal{F}L_g \sigma^2}{RIM}} + \frac{\zeta^2 + \sigma^2/I}{RIN}\right).$$

(15)

It can be seen from Corollary 4.6 that increasing $I$ can still reduce the convergence error. However, the error caused by partial participation, which is shown by the first term in (15), cannot be reduced by increasing $I$. This is because that $L_h$ characterizes the difference between the averaged model over all workers and the centralized model (we will formally explain this property in Section 5). However, with partial participation, the global model on the server becomes a stochastic estimate of the average models over all workers since only a subset of workers are randomly sampled. The variance caused by the sampling strategy is not characterized by $L_h$. In addition, the dominant term in (15) becomes $\mathcal{O}\left(\sqrt{1/RM}\right)$. This means that given the sampling strategy, to achieve a small convergence error, performing a large number of aggregations is necessary. However, increasing $I$ can still accelerate the convergence since the other two terms are reduced.

## 5 DISCUSSIONS

In this section, first, we show that the global Lipshictz gradient (Assumption 4.1) and the heterogeneity-driven Lipschitz constant (Assumption 4.2) used in this paper are weaker than the commonly used local Lipschitz gradient assumption (Assumption 3.1). Second, we explain the significance of $L_h$ by showing its ability to characterize the difference between the virtual averaged model (defined in (5)) and the centralized model. Third, we consider a special case of $L_h = 0$, demonstrating that under such conditions, local SGD can outperform mini-batch SGD.

**Assumptions in this paper are weaker.** In the following proposition, we show that Assumptions 4.1 and 4.2 are weaker than the commonly used Assumption 3.1 in the literature.

**Proposition 5.1.** *If Assumption 3.1 holds, then Assumption 4.1 holds by choosing $L_g = \tilde{L}$ and Assumption 4.2 holds by choosing $L_h = \tilde{L}$.*

Proposition 5.1 also shows how the information about the data heterogeneity contained in $\tilde{L}$ is captured. The information about the smoothness of the gradients remains in $L_g$, which does not change with the data heterogeneity, while $L_h$ characterizes the effect of data heterogeneity. In addition, Proposition 5.1 implies that $L_g \leq \tilde{L}$ and $L_h \leq \tilde{L}$. However, as shown in Table 1, $L_h$ can be much smaller than $\tilde{L}$. We examine the intricate relationship between $L_h$ and $\tilde{L}$ through a deeper analysis of the quadratic[3] (potentially non-convex) objective function:

$$F_i(\mathbf{x}) = \tfrac{1}{2}\mathbf{x}^{\mathrm{T}}\mathbf{A}_i\mathbf{x} + \mathbf{b}_i^{\mathrm{T}}\mathbf{x} + \mathbf{c}_i. \tag{16}$$

By (1), we directly obtain that the global objective function is given by $f(\mathbf{x}) = \tfrac{1}{2}\mathbf{x}^{\mathrm{T}}\mathbf{A}\mathbf{x} + \mathbf{b}^{\mathrm{T}}\mathbf{x} + \mathbf{c}$, where $\mathbf{A} = \frac{1}{N}\sum_{i=1}^{N}\mathbf{A}_i$ and $\mathbf{b} = \frac{1}{N}\sum_{i=1}^{N}\mathbf{b}_i$. In this case, we can derive the explicit forms of $L_h$ and $\tilde{L}$ as shown by the following proposition.

**Proposition 5.2.** *For quadratic objective functions defined in (16), Assumptions 3.1 and 4.2 hold with $\tilde{L} = \max_{i\in[N]}|\lambda(\mathbf{A}_i)|$, $L_h = 2 \cdot \max_{i\in[N]}|\lambda(\mathbf{A}_i - \mathbf{A})|$, respectively, where $|\lambda(\mathbf{A})|$ denotes the largest absolute value of the eigenvalues of $\mathbf{A}$.*

From Proposition 5.2, it can be seen that both $L_h$ and $\tilde{L}$ capture the properties of Hessian matrices for quadratic objective functions. The heterogeneity-driven Lipschitz constant $L_h$ characterizes the largest eigenvalue of the "deviation" of $\{\mathbf{A}_i\}$ from the global Hessian matrix $\mathbf{A}$, while $\tilde{L}$ characterizes the largest eigenvalue of $\{\mathbf{A}_i\}$ themselves. It can be observed that when $\mathbf{A}_i = \mathbf{A}, \forall i$, which means that the difference of local Hessian matrices is zero, Assumption 4.2 holds with $L_h = 0$. Note that, at the same time, we can pick an $\mathbf{A}_i$ such that $\tilde{L} = \max_{i\in[N]}|\lambda(\mathbf{A}_i)|$ is much larger than zero. This observation shows that while the difference among Hessian matrices of local objective functions, shown by $L_h$, can be small, the eigenvalues of the individual Hessian matrix, shown by $\tilde{L}$ can still be very large.

**Explanation of $L_h$.** Assumption 4.2 captures the difference between the averaged model and centralized model, which can be seen from the following proposition. At the $k$th iteration of the $r$th round, we consider the virtual averaged model in (5) and the centralized model in (6).

**Proposition 5.3.** *Given the virtual averaged model at the $r$th round and $k$th iteration $\hat{\mathbf{x}}^{r,k}$, we have*

$$\left\|\mathbb{E}[\hat{\mathbf{x}}^{r,k+1}|\hat{\mathbf{x}}^{r,k}] - \mathbb{E}[\mathbf{x}_c^{r,k+1}|\hat{\mathbf{x}}^{r,k}]\right\|^2 \leq \gamma^2 \cdot \tfrac{L_h^2}{N}\sum_{i=1}^{N}\left\|\mathbf{x}_i^{r,k} - \hat{\mathbf{x}}^{r,k}\right\|^2. \tag{17}$$

Proposition 5.3 shows that although the difference among local models, captured by $\left\|\mathbf{x}_i^{r,k} - \hat{\mathbf{x}}^{r,k}\right\|^2$ (depends on both $\zeta$ and $\sigma$), can be large after multiple local iterations, the difference between the averaged model and centralized model can still be small if $L_h$ is small. This means that while the variance among local models depends on $\zeta$ and $\sigma$, $L_h$ determines how the averaged model is affected by this variance among local models, which is consistent with the theoretical results in Theorem 4.3. Now we show that when $L_h = 0$, $\zeta$ can be arbitrarily large.

**Proposition 5.4.** *For quadratic objective functions defined in (16), when $\zeta = 0$, Assumption 4.2 holds with $L_h = 0$, while when $L_h = 0$, $\zeta$ can be arbitrarily large.*

Proposition 5.4 shows that $L_h = 0$ is not a sufficient condition for $\zeta = 0$, which implies that only using $\zeta$ can overestimate the effect of the data heterogeneity. This is because that as we have seen in Proposition 5.2, for quadratic objective functions, the key effect of heterogeneity on the local updates is shown on the difference between $\mathbf{A}$ and $\mathbf{A}_i$ while $\zeta$ depends not only on the difference between $\mathbf{A}$ and $\mathbf{A}_i$ but also on the difference between $\mathbf{b}$ and $\mathbf{b}_i$. In addition, we notice that in multi-label learning (Zhang & Zhou, 2014), when $\mathbf{A} = \mathbf{A}_i$, $\mathbf{b}$ can be very different from $\mathbf{b}_i$ since

---

[3]Here we do not assume the Hessian matrix is positive definite so that the quadratic objective function can be non-convex.

data examples sharing the same feature can have different labels. This means that $L_h = 0$ but $\zeta > 0$ is possible in practice.

**Extended discussion about Local SGD v.s. Mini-batch SGD.** In the following theorem, we consider the case of $L_h = 0$, by which we show that local SGD can outperform mini-batch SGD even when $\zeta$ is arbitrarily large. Instead of directly applying $L_h = 0$ to Theorem 4.3, we develop a new technique for Theorem 5.5. The difference on the techniques can be shown by the requirement on the learning rate, which no longer depends on $I$ while in Theorem 4.3, it depends on $I$. In Theorem 4.3, it is shown that when $L_h = 0$, two-sided learning rates do not have advantage over a single learning rate for non-convex objective functions. Without loss of generality, we consider $\eta = 1$ in the following.

**Theorem 5.5** (Special Case of $L_h = 0$). *For quadratic objective functions defined in (16), with a common Hessian $\mathbf{A} = \mathbf{A}_i, \forall i$, if $\gamma \leq \frac{1}{|\lambda(\mathbf{A})|}$ and $\eta = 1$, for local SGD with $I$ local iterations,*

$$\min_{r \in [R], k \in [I]} \mathbb{E}\left[\left\|\nabla f(\hat{\mathbf{x}}^{r,k})\right\|^2\right] = \mathcal{O}\left(\frac{\mathcal{F}}{\gamma RI} + \frac{\gamma L_g}{N}\sigma^2\right); \tag{18}$$

*for mini-batch SGD with the batch size $I$,*

$$\min_{r \in [R], k \in [I]} \mathbb{E}\left[\left\|\nabla f(\hat{\mathbf{x}}^{r,k})\right\|^2\right] = \mathcal{O}\left(\frac{\mathcal{F}}{\gamma R} + \frac{\gamma L_g}{NI}\sigma^2\right). \tag{19}$$

**A fair comparison between local SGD and mini-batch SGD.** In Theorem 5.5, the cost of communication and computation is the same for both local SGD and mini-batch SGD since the number of aggregations is $R$ and the total number of gradients sampled is $NRI$ for both algorithms. The restriction for the learning rate is also the same. Comparing (18) with (19), we see that the difference is on the place where $I$ appears. For local SGD, $I$ is in the first term of (18), which means that local SGD uses more computation to reduce the error caused by the initialization. For mini-batch SGD, $I$ is in the second term of (19), which means that mini-batch SGD uses more computation to reduce the error caused by the variance. When the variance $\sigma^2$ is small (i.e., $\sigma^2 \to 0$), the first terms of (18) and (19) dominate, and it becomes beneficial to choose $\gamma$ to be as large as possible, so we can choose $\gamma = \frac{1}{|\lambda(\mathbf{A})|}$ for both cases. Then, as $\sigma^2 \to 0$, the convergence rate of local SGD goes to $\mathcal{O}(\frac{1}{RI})$ while the convergence rate of mini-batch SGD goes to $\mathcal{O}(\frac{1}{R})$. This implies that when $\sigma^2$ is small, the speed of convergence for local SGD can be much faster than that for mini-batch SGD, which will be validated in the experiments in the next section. The discussion on the comparison between local SGD and mini-batch SGD for non-convex quadratic objective functions with $L_h \neq 0$ can be found in Section A.2.

## 6 EXPERIMENTS

For the non-IID setting, the data on each worker is sampled in two steps. First, $X\%$ of the data on one worker is sampled from a single label. Then we uniformly partition the remaining dataset into all workers and we say that the percentage of heterogeneous data on this worker is $X\%$. Additional experimental details and results can be found in Appendix C.

**Results with MNIST dataset.** In Table 1, a two-layer neural network with cross-entropy loss and a linear regression model with mean squared error (MSE) is trained with the MNIST dataset (LeCun et al., 1998). The MNIST dataset is partitioned into 10 workers.

**Results with CIFAR-10 dataset.** A CNN model with cross-entropy loss is trained with the CIFAR-10 dataset (Krizhevsky & Hinton, 2009). The CIFAR-10 dataset is partitioned into 100 workers, and we randomly sample 10 workers in each round. The results for the general non-convex functions with partial participation are shown in Table 2 and Figure 2. In Table 2, it can be seen that $L_h$ is far smaller than $\tilde{L}$. In Corollary 4.6, it is shown that when $L_h$ is small, increasing $I$ can reduce the convergence error. This is validated by experimental results in Figure 2. It can be observed that for both $50\%$ and $75\%$ of heterogeneous data, $I = 80$ is the best curve and increasing $I$ can accelerate the convergence.

**Results with synthetic data.** For the special case of $L_h = 0$, we construct quadratic examples to validate the insights from Theorem 5.5. We construct the objective function as $F_i(\mathbf{x}) = \frac{1}{2}\|\mathbf{U}\mathbf{x} - \mathbf{v}_i\|^2$, where $\mathbf{U} \in \mathbb{R}^{100 \times 100}$, $\mathbf{v}_i \in \mathbb{R}^{100}$. Each column of $\mathbf{U}$ and $\mathbf{v}_i$ are sampled from

Table 2: Estimated $L_h$, $\tilde{L}$, $L_g$ for a CNN model trained with the CIFAR-10 dataset.

| Obj. Function | CNN | | | |
|---|---|---|---|---|
| Heterogeneity | 25% | 50% | 75% | 100% |
| $\tilde{L}$ | $447.59 \pm 22.27$ | $898.49 \pm 38.57$ | $1131.36 \pm 47.82$ | $1662.24 \pm 62.18$ |
| $L_h$ | $0.96 \pm 0.13$ | $1.21 \pm 0.19$ | $1.63 \pm 0.26$ | $2.15 \pm 0.34$ |
| $L_g$ | $323.35 \pm 15.36$ | $323.35 \pm 15.36$ | $323.35 \pm 15.36$ | $323.35 \pm 15.36$ |

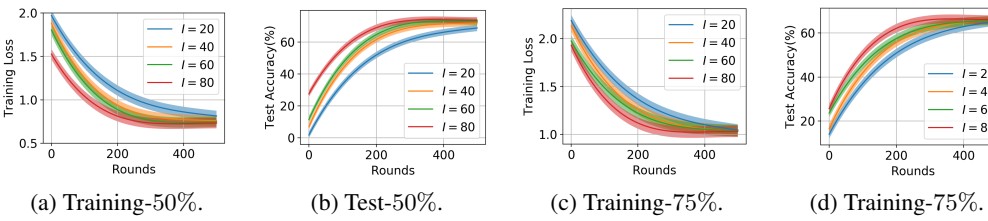

|       (a) Training-50%.       |       (b) Test-50%.       |       (c) Training-75%.       |       (d) Training-75%.       |

Figure 2: Results with CNN. The dataset is CIFAR-10. The learning rates are chosen as $\eta = 2$ and $\gamma = 0.05$. Results for 50% of the heterogeneous data are shown in (a) and (b). Results for 75% of the heterogeneous data are shown in (c) and (d).

a normal distribution $\mathcal{N}(\mathbf{0}, \mathbf{I})$. In this case, the gradient divergence is $\|\mathbf{U}(\mathbf{v}_i - \mathbf{v})\|^2 > 0$. Table 3 shows the results for quadratic objective functions. To distinguish the number of local updates from the mini-batch size in the experiments, we use a separate variable $s$ to indicate the mini-batch size. Theorem 4.3 shows that when $L_h = 0$, using two-sided learning rates does not have advantages over a single learning rate. This is validated by the experiments shown in Table 3, where there is no difference among the results with different learning rates when keeping the product of learning rates. Comparing results with $I = 1$, $I = 5$, and $I = 10$ with $s = 1$ in Table 3, it can be seen that more local updates can reduce the communication cost, which validates the results in Theorem 5.5. By the comparison between the results of $I = 1, s = 5$ and $I = 5, s = 1$ and the comparison between the results of $I = 1, s = 10$ and $I = 10, s = 1$, we can see that keeping the number of gradients sampled in one round the same, local SGD ($I > 1$) converges faster than mini-batch SGD ($I = 1$) when $\sigma^2$ is small, which validates the discussion for Theorem 5.5.

Table 3: Special case of $L_h = 0$ with the quadratic objective functions. $I = 1$ is equivalent to mini-batch SGD. The number of rounds is the communication rounds needed for achieving a target function value of 0.8. For $(\eta, \gamma)$, we fix $I = 10$ and for $(I, s)$, we fix $\eta = 1, \gamma = 0.005$.

| $(\eta, \gamma)$ | $(1, 0.005)$ | $(2, 0.0025)$ | $(5, 0.001)$ | $(10, 0.0005)$ |
|---|---|---|---|---|
| Number of Rounds | $86 \pm 1.6$ | $86 \pm 1.6$ | $86 \pm 1.6$ | $86 \pm 1.6$ |
| $(I, s)$ | $(1,1)\&(1,5)$ | $(1,10)$ | $(5,1)$ | $(10,1)$ |
| Number of Rounds | $927 \pm 3.4$ | $925 \pm 1.7$ | $187 \pm 2.3$ | $95 \pm 2.4$ |

# 7 CONCLUSIONS AND FUTURE WORKS

In this paper, we bridged the gap between the pessimistic theoretical results and the good experimental performance for FL algorithms by introducing a new theoretical perspective of the data heterogeneity, which is shown by the proposed heterogeneity-driven Lipschitz constant $L_h$. Using the new assumption, we developed a novel convergence analysis for FedAvg and identified the regions where local updates can help to improve the convergence even when data are highly heterogeneous. Our convergence upper bounds for both full participation and partial participation can be better compared to the state of the art in the literature. In addition, we showed that the new assumptions used in this paper are weaker. Moreover, we discussed the significance of the proposed heterogeneity-driven Lipschitz constant, by which, we successfully identified a region where local SGD can out perform mini-batch SGD. It is worth mentioning that our new assumption and analysis have the potential to be applied in the convergence analysis of other FL algorithms besides FedAvg. The reason is that this new assumption can be employed in a pivotal step shared in existing literature for the convergence analysis of various FL algorithms. More detailed discussion can be found in Appendix A.1. Hence, our future works include leveraging our new assumption and analysis to enhance the convergence upper bound of other FL algorithms.

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

# APPENDIX

This Appendix is composed of three sections. Section A.1 provides more details of related works and additional discussions on the theoretical results. Section B provides proofs for theorems, corollaries and propositions in the main paper. Section C provides additional details and results of experiments.

# A ADDITIONAL DISCUSSIONS

## A.1 ADDITIONAL DETAILS OF RELATED WORKS

There have been a considerable amount of works analyzing the convergence rate of federated learning algorithms (not limited to FedAvg), with non-convex objective functions (Haddadpour, Farzin et al., 2019; Yu et al., 2019b; Wang & Joshi, 2019; Karimireddy et al., 2020; Reddi et al., 2020). A key step shared by these analyses is to relate the difference of gradients,

$$\left\| \frac{1}{N} \sum_{i=1}^{N} \nabla F_i(\mathbf{x}_i) - \nabla f(\bar{\mathbf{x}}) \right\|,$$

to the model divergence,

$$\frac{1}{N} \sum_{i=1}^{N} \left\| \mathbf{x}_i - \bar{\mathbf{x}} \right\|,$$

which can be found, for example, in inequality (10) in the supplementary of Yu et al. (2019b), the inequality (6) in the supplementary of Reddi et al. (2020), and the proof of Lemma 19 in Karimireddy et al. (2020). In this step, the local Lipschitz gradient assumption is often applied, which amplifies the effect of data heterogeneity. In this paper, the heterogeneity-driven Lipschitz constant $L_h$ is applied in this step so that the convergence error is much smaller than that based on $\tilde{L}$, since it can be seen in Table 1 that $L_h$ is often far smaller than $\tilde{L}$. Therefore, we believe that our techniques can also be applied to other works to improve the convergence analysis.

There are two papers (Wang et al., 2022; Das et al., 2022) closely related to our work. Both works assume the Lipschitz gradient for each local objective function while we only assume it for the global objective function. Therefore, the information about data heterogeneity contained in $\tilde{L}$ is not characterized in either work. Wang et al. (2022) try to re-characterize the data heterogeneity by extending the single gradient divergence assumption ((4) in Wang et al. (2022)) to the averaged gradient divergence assumption ((15) in Wang et al. (2022)). Wang et al. (2022) consider the convex objective function and their analysis cannot guarantee convergence to a stationary point while we consider general non-convex objective function and our results can guarantee convergence to a stationary point. Das et al. (2022) introduce a parameter $\alpha$ in the process of relating the difference of gradients to the model divergence, which can be covered by $L_h$ in this paper. But $\alpha$ cannot cover what $L_h$ can show since they still assume Lipschitz gradient for each local objective function. They only use $\alpha$ as an intermediate step instead of theoretically analyzing the effect of data heterogeneity. In their theoretical results, the convergence error increases with $I$ even when $\alpha = 0$.

In the following, we provide more details about the difference between Wang et al. (2022)) and our paper. In Wang et al. (2022)), a new metric for data heterogeneity, $\rho$, the average drift at optimum, is proposed. The definition of $\rho$ is

$$\left\| \frac{1}{\gamma I} \left( \frac{1}{N} \sum_{i=1}^{N} \mathbf{x}_i^{r,I} - \bar{\mathbf{x}}^r \right) \right\| = \rho. \tag{A.1}$$

We discuss the difference between Wang et al. (2022)) and our paper in the following three aspects.

First, the new metric $\rho$ in Wang et al. (2022)) focuses on the difference on models while in our paper, we still focus on the difference on the gradients. The key insight in Wang et al. (2022)) is that since $\rho$ is small, when the global model is $\mathbf{x}^*$, after multiple local updates, the averaged model does not change significantly. In our paper, the key insight is that since $L_h$ can be small, the difference

between the current global gradient and the current averaged local gradients can be small. In Wang et al. (2022)), the gradient divergence is not used in the analysis. In our analysis, we still use the gradient divergence jointly with the proposed $L_h$ to measure the data heterogeneity.

Second, in Wang et al. (2022)), it is only empirically shown that $\rho$ can be small. In our paper, we not only empirically show that $L_h$ can be small, but also provide an analytical example. Our quadratic example can be non-convex, a case which $\rho$ cannot cover.

Third, one weakness of using $\rho$ is that in the convergence bound in Wang et al. (2022)), the convergence error shown by $\rho$ cannot vanish. This means that by choosing $\gamma = \frac{1}{\sqrt{R}}$, when $R$ goes to infinity, the convergence bound cannot guarantee that FedAvg can converge to the local minima of the global objective function. On the contrary, our convergence bound can guarantee the convergence to the local minima of the global objective function, which is shown by Corollary 4.4.

### A.2 MORE DISCUSSIONS ON THE THEORETICAL RESULTS

In this section, we provide a comparison between mini-batch SGD and local SGD for quadratic objective functions with $L_h > 0$. For simplicity, we consider that all $\mathbf{A}_i$'s are invertible.

**Theorem A.1** (Quadratic Objective Functions with $L_h \neq 0$). *With $\gamma \leq \{\frac{1}{L}, \frac{1}{IL_h}\}$, for local SGD with non-convex quadratic functions, we have*

$$\frac{1}{T}\sum_{t=0}^{T-1}\mathbb{E}\left\|\nabla f(\hat{\mathbf{x}}^t)\right\|^2 = \mathcal{O}\left(\frac{\mathcal{F}}{\gamma RI} + \frac{\gamma L_g\sigma^2}{2N} + \gamma^2 L_h^2(I-1)^2\zeta^2 + \gamma^2 L_h^2 I\sigma^2\right). \quad (A.2)$$

The proof can be found in Section B.13. Compared with the theoretical results for general non-convex objective functions, the main improvement is on the choice of learning rate. We develop new techniques in the proof to achieve the improvement on the learning rate, which fully takes advantage of the properties of quadratic objective functions. For the ease of comparison, the convergence bound for mini-batch SGD is provided as follows. With $\alpha \leq \frac{1}{L_g}$, for mini-batch SGD, we have

$$\frac{1}{T}\sum_{t=0}^{T-1}\mathbb{E}\left\|\nabla f(\hat{\mathbf{x}}^t)\right\|^2 = \mathcal{O}\left(\frac{\mathcal{F}}{\alpha R} + \frac{\alpha L_g\sigma^2}{2NI}\right). \quad (A.3)$$

First, we consider the simplest case, $\sigma = 0$.

**Corollary A.2** ($\sigma = 0$ for Quadratic Objective Functions). *When $\sigma = 0$ and $I \leq \frac{1}{L_h}$, with $\gamma = \frac{1}{(RI)^{\frac{1}{3}}}$, for local SGD, we have*

$$\frac{1}{T}\sum_{t=0}^{T-1}\mathbb{E}\left\|\nabla f(\hat{\mathbf{x}}^t)\right\|^2 = \mathcal{O}\left(\frac{\mathcal{F}}{(RI)^{\frac{2}{3}}} + \frac{\zeta^2}{(RI)^{\frac{2}{3}}}\right), \quad (A.4)$$

*while for mini-batch SGD, we have*

$$\frac{1}{T}\sum_{t=0}^{T-1}\mathbb{E}\left\|\nabla f(\hat{\mathbf{x}}^t)\right\|^2 = \mathcal{O}\left(\frac{\mathcal{F}L_g}{R}\right). \quad (A.5)$$

In this case, when $I^{\frac{1}{3}}L_g \geq R^{\frac{1}{3}}$, with non-convex objective functions, the convergence rate of local SGD is better than that of mini-batch SGD.

Second, we consider the case where $\sigma$ is small.

**Corollary A.3** ($\sigma$ is small). *When $\sigma^2 \leq \frac{2N\mathcal{F}}{\gamma^2 L_g RI}$ and $I \leq \frac{1}{L_h}$, with $\gamma = \frac{1}{(RI)^{\frac{1}{3}}}$, for local SGD, we have*

$$\frac{1}{T}\sum_{t=0}^{T-1}\mathbb{E}\left\|\nabla f(\hat{\mathbf{x}}^t)\right\|^2 = \mathcal{O}\left(\frac{\mathcal{F}}{(RI)^{\frac{2}{3}}} + \frac{\zeta^2}{(RI)^{\frac{2}{3}}} + \frac{2N\mathcal{F}}{L_g RI^2}\right), \quad (A.6)$$

*while for mini-batch SGD, we have*

$$\frac{1}{T} \sum_{t=0}^{T-1} \mathbb{E} \left\| \nabla f(\hat{\mathbf{x}}^t) \right\|^2 = \mathcal{O} \left( \frac{N\mathcal{F}}{L_g(RI)^{\frac{1}{3}}} \right). \tag{A.7}$$

When $(RI)^{\frac{1}{3}} \leq \frac{L_g}{N}$, with non-convex objective functions, the convergence rate of local SGD is better than that of mini-batch SGD. From above two Corollaries, it can be seen that when $L_h$ is small enough and the stochatstic noise is small, local SGD can be better than mini-batch SGD.

## B  PROOFS

The description of FedAvg with two-sided learning rates can be found in Algorithm 1. For full participation, we have $\mathcal{S}_r = \{1, 2, \ldots, N\}, \forall r$ and $M = N$. For partial participation, we have $M < N$.

---

**Algorithm 1:** FedAvg with two-sided learning rates

**Input:** $\gamma, \eta, \bar{\mathbf{x}}^0, I$
**Output:** Global averaged model $\bar{\mathbf{x}}^R$

1 **for** $r = 0$ *to* $R - 1$ **do**
2      Sample a subset of workers $\mathcal{S}_r, |\mathcal{S}_r| = M$;
3      Distribute the current global model $\bar{\mathbf{x}}^r$ to workers in $\mathcal{S}_r$;
4      **for** *Each worker $i$ in $\mathcal{S}_r$, in parallel* **do**
         /* Local Update Phase */
5          $k = 0$;
6          **while** $k < I$ **do**
7              Sample the stochastic gradient $\mathbf{g}_i(\mathbf{x}_i^{r,k})$;
8              Update the local model
9              $\mathbf{x}_i^{r,k+1} \leftarrow \mathbf{x}_i^{r,k} - \gamma \mathbf{g}_i(\mathbf{x}_i^{r,k})$;
10              $k \leftarrow k + 1$;
11          Send $\Delta_i^r \leftarrow \bar{\mathbf{x}}^r - \mathbf{x}_i^{r,I}$ to the server;
     /* Global Update Phase */
12      Update the global model
13      $\bar{\mathbf{x}}^{r+1} \leftarrow \bar{\mathbf{x}}^r - \eta \cdot \frac{1}{M} \sum_{i \in \mathcal{S}_r} \Delta_i^r$;

---

### B.1  TECHNICAL NOVELTIES

Before proceeding to the proof of our theoretical results, we summarize the technical novelties as follows, which provides a reference for our techniques.

(1) We need to develop new techniques to incorporate Assumptions 4.1 and 4.2. In the proof of Theorem 4.3 shown in the Appendix, we need to characterize the difference between local gradients. In the literature, this is done by applying the local Lipschitz constant in shown in Assumption 3.1 in the main paper. In our paper, since Assumption 3.1 is replaced by our newly introduced Assumption 4.2, the proof techniques in the literature cannot be applied. It requires developing new proof techniques to use Assumption 4.2 as shown in the proof of Lemma B.1-B.3. For example, in Lemma B.1, due to the application of Assumptions 4.1 and 4.2, we have to deal with a new term, the local gradient deviation $\| \frac{1}{N} \sum_{i=1}^{N} \nabla F_i(\mathbf{x}_i) - \nabla F_j(\mathbf{x}_j) \|^2$, which is not shown in other techniques. Another example is in the proof of Theorem 4.5. Due to that we only use the global Lipschitz gradient assumption, we have to derive a new method to bound and incorporate the sampling related term $\mathbb{E} \| \frac{1}{M} \sum_{i \in \mathcal{S}_r} \sum_{k=0}^{I} \mathbf{g}_i(\mathbf{x}_r^{r,k}) \|^2$, which can be seen from (B.33) to (B.40).

(2) In addition to using Assumption 4.2 to characterize the new convergence rate of FedAvg, we also validate this assumption from the theoretical perspective. We develop the proof for Proposition 5.1-5.3 in Appendix.

(3) In Theorem 5.5, since we use iteration-by-iteration analysis in the proof, the learning rate is not a function of $I$. It can be seen that in the literature, such as Theorem IV in Karimireddy et al. (2020), for quadratic objective functions, the learning rate is upper bounded by $I$. The advantage of that $\gamma$ is not a function of $I$ can be explained as follows. In Theorem 5.5, to obtain the optimal dependence on $R$ and $I$, we can choose $\gamma = \frac{1}{\sqrt{RI}}$. This requires that $\frac{1}{\sqrt{RI}} \leq \frac{1}{L_g}$, which means that $I$ can be as large as possible. However, if $\gamma \leq \frac{1}{IL_g}$, we will have $\frac{1}{\sqrt{RI}} \leq \frac{1}{IL_g}$ such that $I \leq \frac{R}{L_g^2}$, which means that to achieve the convergence rate of $\mathcal{O}(\frac{1}{\sqrt{RI}})$, $I$ cannot be arbitrarily large. Therefore, the range of the learning rate in Theorem 5.5 significantly enhances the convergence analysis.

## B.2 Additional Lemmas

In the proof, we use $\mathbf{x}_i$ to denote the local model of worker $i$ regardless of the number of iterations, and use $\bar{\mathbf{x}} := \frac{1}{N} \sum_{i=1}^{N} \mathbf{x}_i$ to denote the averaged model. Following lemmas are useful in the proof for main theorems.

**Lemma B.1** (Local Gradient Deviation). *With Assumption 3.3, 4.1 and 4.2, we have*

$$\frac{1}{N} \sum_{j=1}^{N} \left\| \frac{1}{N} \sum_{i=1}^{N} \nabla F_i(\mathbf{x}_i) - \nabla F_j(\mathbf{x}_j) \right\|^2 \leq 3(L_h^2 + L_g^2) \cdot \frac{1}{N} \sum_{j=1}^{N} \|\bar{\mathbf{x}} - \mathbf{x}_j\|^2 + 3\zeta^2. \tag{B.1}$$

**Lemma B.2** (Model Divergence). *With $\gamma \leq \frac{1}{30(L_h+L_g)I}$, we have*

$$\sum_{k=0}^{I-1} \frac{L_h^2}{N} \sum_{i=1}^{N} \mathbb{E} \left\| \mathbf{x}_i^{r,k} - \hat{\mathbf{x}}^{r,k} \right\|^2 \leq 3c(I-1)^3 \gamma^2 L_h^2 \zeta^2 + c(I-1)^2 \gamma^2 L_h^2 \sigma^2, \tag{B.2}$$

where $c = 3$ and $\hat{\mathbf{x}}^{r,k} = \frac{1}{N} \sum_{i=1}^{N} \mathbf{x}_i^{r,k}$.

**Lemma B.3** (The Change of Averaged Models). *With $\gamma \leq \frac{1}{3IL_g}$, at $r$th round, we have*

$$\mathbb{E} \left\| \hat{\mathbf{x}}^{r,k} - \bar{\mathbf{x}}^r \right\|^2 \leq 5(I-1) \cdot \frac{\gamma^2 \sigma^2}{N} + 30I\gamma^2 \sum_{k=0}^{I-1} \frac{L_h^2}{N} \sum_{i=1}^{N} \mathbb{E} \left\| \mathbf{x}_i^{r,k} - \hat{\mathbf{x}}^{r,k} \right\|^2$$
$$+ 30I(I-1)\gamma^2 \mathbb{E} \left\| \nabla f(\bar{\mathbf{x}}^r) \right\|^2. \tag{B.3}$$

## B.3 Proof of Lemma B.1

We start with the LHS of the inequality in Lemma B.1.

$$\frac{1}{N} \sum_{j=1}^{N} \left\| \frac{1}{N} \sum_{i=1}^{N} \nabla F_i(\mathbf{x}_i) - \nabla F_j(\mathbf{x}_j) \right\|^2$$

$$= \frac{1}{N} \sum_{j=1}^{N} \left\| \frac{1}{N} \sum_{i=1}^{N} \nabla F_i(\mathbf{x}_i) - \nabla f(\bar{\mathbf{x}}) + \nabla f(\bar{\mathbf{x}}) - \nabla f(\mathbf{x}_j) + \nabla f(\mathbf{x}_j) - \nabla F_j(\mathbf{x}_j) \right\|^2$$

$$\leq 3 \left\| \frac{1}{N} \sum_{i=1}^{N} \nabla F_i(\mathbf{x}_i) - \nabla f(\bar{\mathbf{x}}) \right\|^2 + 3 \cdot \frac{1}{N} \sum_{j=1}^{N} \|\nabla f(\bar{\mathbf{x}}) - \nabla f(\mathbf{x}_j)\|^2 + 3 \cdot \frac{1}{N} \sum_{j=1}^{N} \|\nabla f(\mathbf{x}_j) - \nabla F_j(\mathbf{x}_j)\|^2$$

$$\overset{(a)}{\leq} 3 \left\| \frac{1}{N} \sum_{i=1}^{N} \nabla F_i(\mathbf{x}_i) - \nabla f(\bar{\mathbf{x}}) \right\|^2 + 3L^2 \cdot \frac{1}{N} \sum_{j=1}^{N} \|\bar{\mathbf{x}} - \mathbf{x}_j\|^2 + 3\zeta^2$$

$$\overset{(b)}{\leq} 3 \cdot \frac{D^2}{N} \sum_{i=1}^{N} \|\bar{\mathbf{x}} - \mathbf{x}_i\|^2 + 3L^2 \cdot \frac{1}{N} \sum_{j=1}^{N} \|\bar{\mathbf{x}} - \mathbf{x}_j\|^2 + 3\zeta^2$$

$$= 3(D^2 + L^2) \cdot \frac{1}{N} \sum_{j=1}^{N} \|\bar{\mathbf{x}} - \mathbf{x}_j\|^2 + 3\zeta^2, \tag{B.4}$$

where $(a)$ is due to Assumption 3.3 and 4.1 and $(b)$ is due to Assumption 4.2.

### B.4 PROOF OF LEMMA B.2

At $r$th round, we have

$$\frac{L_h^2}{N} \sum_{i=1}^{N} \mathbb{E} \left\| \mathbf{x}_i^{r,k} - \hat{\mathbf{x}}^{r,k} \right\|^2$$

$$= \frac{\gamma^2 L_h^2}{N} \sum_{i=1}^{N} \mathbb{E} \left\| \sum_{m=0}^{k-1} \left( \mathbf{g}_i(\mathbf{x}_i^{r,m}) - \frac{1}{N} \sum_{j=1}^{N} \mathbf{g}_j(\mathbf{x}_j^{r,m}) \right) \right\|^2$$

$$= \frac{\gamma^2 L_h^2}{N} \sum_{i=1}^{N} \mathbb{E} \left\| \sum_{m=0}^{k-1} \left( \mathbf{g}_i(\mathbf{x}_i^{r,m}) - \nabla F_i(\mathbf{x}_i^{r,m}) + \nabla F_i(\mathbf{x}_i^{r,m}) \right. \right.$$

$$\left. \left. \cdot - \frac{1}{N} \sum_{j=1}^{N} \nabla F_j(\mathbf{x}_j^{r,m}) + \frac{1}{N} \sum_{j=1}^{N} \nabla F_j(\mathbf{x}_j^{r,m}) - \frac{1}{N} \sum_{j=1}^{N} \mathbf{g}_j(\mathbf{x}_j^{r,m}) \right) \right\|^2$$

$$\leq 2 \cdot \frac{\gamma^2 L_h^2}{N} \sum_{i=1}^{N} \mathbb{E} \left\| \sum_{m=0}^{k-1} \left( \nabla F_i(\mathbf{x}_i^{r,m}) - \frac{1}{N} \sum_{j=1}^{N} \nabla F_j(\mathbf{x}_j^{r,m}) \right) \right\|^2$$

$$+ 2 \cdot \frac{\gamma^2 L_h^2}{N} \sum_{i=1}^{N} \mathbb{E} \left\| \sum_{m=0}^{k-1} \left( \mathbf{g}_i(\mathbf{x}_i^{r,m}) - \nabla F_i(\mathbf{x}_i^{r,m}) + \frac{1}{N} \sum_{j=1}^{N} \nabla F_j(\mathbf{x}_j^{r,m}) - \frac{1}{N} \sum_{j=1}^{N} \mathbf{g}_j(\mathbf{x}_j^{r,m}) \right) \right\|^2$$

$$\overset{(a)}{\leq} 2 \cdot \frac{\gamma^2 L_h^2}{N} \sum_{i=1}^{N} \mathbb{E} \left\| \sum_{m=0}^{k-1} \left( \nabla F_i(\mathbf{x}_i^{r,m}) - \frac{1}{N} \sum_{j=1}^{N} \nabla F_j(\mathbf{x}_j^{r,m}) \right) \right\|^2$$

$$+ 2 \cdot \frac{\gamma^2 L_h^2}{N} \sum_{i=1}^{N} \mathbb{E} \left\| \sum_{m=0}^{k-1} \left( \mathbf{g}_i(\mathbf{x}_i^{r,m}) - \nabla F_i(\mathbf{x}_i^{r,m}) \right) \right\|^2$$

$$\leq 2 \cdot \frac{\gamma^2 L_h^2}{N} \sum_{i=1}^{N} \mathbb{E} \left\| \sum_{m=0}^{k-1} \left( \nabla F_i(\mathbf{x}_i^{r,m}) - \frac{1}{N} \sum_{j=1}^{N} \nabla F_j(\mathbf{x}_j^{r,m}) \right) \right\|^2 + 2\gamma^2 L_h^2 k \sigma^2$$

$$\leq 2k \cdot \frac{\gamma^2 L_h^2}{N} \cdot \sum_{i=1}^{N} \sum_{m=0}^{k-1} \mathbb{E} \left\| \nabla F_i(\mathbf{x}_i^{r,m}) - \frac{1}{N} \sum_{j=1}^{N} \nabla F_j(\mathbf{x}_j^{r,m}) \right\|^2 + 2\gamma^2 L_h^2 k \sigma^2$$

$$\overset{(b)}{\leq} 2k\gamma^2 L_h^2 \sum_{m=0}^{k-1} \left( 3(L_h^2 + L_g^2) \frac{1}{N} \sum_{k=1}^{N} \mathbb{E} \left\| \hat{\mathbf{x}}^{r,m} - \mathbf{x}_k^{r,m} \right\|^2 + 3\zeta^2 \right) + 2\gamma^2 L_h^2 k \sigma^2$$

$$= 6k\gamma^2 L_h^2 (L_h^2 + L_g^2) \sum_{m=0}^{k-1} \frac{1}{N} \sum_{i=1}^{N} \mathbb{E} \left\| \hat{\mathbf{x}}^{r,m} - \mathbf{x}_i^{r,m} \right\|^2 + 6k^2 \gamma^2 L_h^2 \zeta^2 + 2\gamma^2 L_h^2 k \sigma^2, \tag{B.5}$$

where $(a)$ is due to $\frac{1}{N} \sum_{i=1}^{N} \|\mathbf{y}_i - \bar{\mathbf{y}}\|^2 \leq \frac{1}{N} \sum_{i=1}^{N} \|\mathbf{y}_i\|^2$ and we let $\mathbf{y}_i = \sum_{m=0}^{k-1} [\mathbf{g}_i(\mathbf{x}_i^{r,m}) - \nabla F_i(\mathbf{x}_i^{r,m})]$, and $(b)$ is due to Lemma B.1.

Note that when $k = I$, we have $\mathbf{x}_i^{r,k} = \mathbf{x}_i^{r+1,0} = \bar{\mathbf{x}}^{r+1}$ and when $k = 0$, we have $\mathbf{x}_i^{r,k} = \bar{\mathbf{x}}^r$. So we have $\left\| \mathbf{x}_i^{r,I} - \hat{\mathbf{x}}^{r,I} \right\|^2 = 0$, for $k = 0, I$. Then sum over $k$ for one round on both sides, we have

$$\sum_{k=1}^{I} \frac{L_h^2}{N} \sum_{i=1}^{N} \mathbb{E} \left\| \mathbf{x}_i^{r,k} - \hat{\mathbf{x}}^{r,k} \right\|^2$$

$$\leq \sum_{k=1}^{I} \left( 6k\gamma^2 L_h^2 (L_h^2 + L_g^2) \sum_{m=0}^{k-1} \frac{1}{N} \sum_{i=1}^{N} \mathbb{E} \left\| \hat{\mathbf{x}}^{r,m} - \mathbf{x}_i^{r,m} \right\|^2 + 6k^2 \gamma^2 L_h^2 \zeta^2 + 2\gamma^2 L_h^2 k \sigma^2 \right)$$

$$\leq 3\gamma^2 L_h^2 (L_h^2 + L_g^2) I(I-1) \sum_{m=0}^{I-1} \frac{1}{N} \sum_{i=1}^{N} \mathbb{E} \left\| \mathbf{x}_i^{r,m} - \hat{\mathbf{x}}^{r,m} \right\|^2$$
$$+ 6(I-1)^3 \gamma^2 L_h^2 \zeta^2 + 2(I-1)^2 \gamma^2 L_h^2 \sigma^2. \tag{B.6}$$

Move the first term on RHS of (B.6) to LHS, we have

$$\left( L_h^2 - 3\gamma^2 L_h^2 (L_h^2 + L_g^2) I(I-1) \right) \sum_{k=0}^{I-1} \frac{1}{N} \sum_{i=1}^{N} \mathbb{E} \left\| \mathbf{x}_i^{r,k} - \hat{\mathbf{x}}^{r,k} \right\|^2$$
$$\leq 6(I-1)^3 \gamma^2 L_h^2 \zeta^2 + 2(I-1)^2 \gamma^2 L_h^2 \sigma^2. \tag{B.7}$$

With $\gamma \leq \frac{1}{30(L_h + L_g)I}$, we have

$$L_h^2 - 3\gamma^2 L_h^2 (L_h^2 + L_g^2) I(I-1) > 0. \tag{B.8}$$

Since $\frac{2}{1 - 3\gamma^2 (L_h^2 + L_g^2) I(I-1)} < 3$, we can choose $c = 3$ such that

$$\sum_{k=0}^{I-1} \frac{L_h^2}{N} \sum_{i=1}^{N} \mathbb{E} \left\| \mathbf{x}_i^{r,k} - \hat{\mathbf{x}}^{r,k} \right\|^2 \leq 3c(I-1)^3 \gamma^2 L_h^2 \zeta^2 + c(I-1)^2 \gamma^2 L_h^2 \sigma^2. \tag{B.9}$$

### B.5 PROOF OF LEMMA B.3

At $r$th round, for $k = 0$, we have

$$\mathbb{E} \left\| \hat{\mathbf{x}}^{r,k} - \bar{\mathbf{x}}^r \right\|^2 = 0. \tag{B.10}$$

At $r$th round, for $1 \leq k \leq I-1$, we have

$$\mathbb{E} \left\| \hat{\mathbf{x}}^{r,k} - \bar{\mathbf{x}}^r \right\|^2$$
$$= \mathbb{E} \left\| \hat{\mathbf{x}}^{r,k-1} - \frac{\gamma}{N} \sum_{i=1}^{N} \mathbf{g}_i(\mathbf{x}_i^{r,k-1}) - \bar{\mathbf{x}}^r \right\|^2$$
$$= \mathbb{E} \left\| \hat{\mathbf{x}}^{r,k-1} - \bar{\mathbf{x}}^r - \gamma \left( \frac{1}{N} \sum_{i=1}^{N} \mathbf{g}_i(\mathbf{x}_i^{r,k-1}) - \frac{1}{N} \sum_{i=1}^{N} \nabla F_i(\mathbf{x}_i^{r,k-1}) + \frac{1}{N} \sum_{i=1}^{N} \nabla F_i(\mathbf{x}_i^{r,k-1}) \right. \right.$$
$$\left. \left. - \nabla f(\hat{\mathbf{x}}^{r,k-1}) + \nabla f(\hat{\mathbf{x}}^{r,k-1}) - \nabla f(\bar{\mathbf{x}}^r) + \nabla f(\bar{\mathbf{x}}^r) \right) \right\|^2$$
$$\overset{(a)}{\leq} \left( 1 + \frac{1}{2I-1} \right) \mathbb{E} \left\| \hat{\mathbf{x}}^{r,k-1} - \bar{\mathbf{x}}^r \right\|^2 + \frac{\gamma^2 \sigma^2}{N} + 6I\gamma^2 \mathbb{E} \left\| \frac{1}{N} \sum_{i=1}^{N} \nabla F_i(\mathbf{x}_i^{r,k-1}) - \nabla f(\hat{\mathbf{x}}^{r,k-1}) \right\|^2$$
$$+ 6I\gamma^2 \mathbb{E} \left\| \nabla f(\hat{\mathbf{x}}^{r,k-1}) - \nabla f(\bar{\mathbf{x}}^r) \right\|^2 + 6I\gamma^2 \mathbb{E} \| \nabla f(\bar{\mathbf{x}}^r) \|^2 \tag{B.11}$$
$$\overset{(b)}{\leq} \left( 1 + \frac{1}{2I-1} + 6I\gamma^2 L_g^2 \right) \mathbb{E} \left\| \hat{\mathbf{x}}^{r,k-1} - \bar{\mathbf{x}}^r \right\|^2 + \frac{\gamma^2 \sigma^2}{N} + \frac{6I\gamma^2 L_h^2}{N} \sum_{i=1}^{N} \mathbb{E} \left\| \mathbf{x}_i^{r,k-1} - \hat{\mathbf{x}}^{r,k-1} \right\|^2$$
$$+ 6I\gamma^2 \mathbb{E} \| \nabla f(\bar{\mathbf{x}}^r) \|^2 \tag{B.12}$$
$$\overset{(c)}{\leq} 5(I-1) \cdot \frac{\gamma^2 \sigma^2}{N} + 30I\gamma^2 \sum_{k=0}^{I-1} \frac{L_h^2}{N} \sum_{i=1}^{N} \mathbb{E} \left\| \mathbf{x}_i^{r,k} - \hat{\mathbf{x}}^{r,k} \right\|^2 + 30I(I-1)\gamma^2 \mathbb{E} \| \nabla f(\bar{\mathbf{x}}^r) \|^2, \tag{B.13}$$

where $(a)$ is due to that $\| \mathbf{x} + \mathbf{y} \|^2 \leq (1+p) \| \mathbf{x} \|^2 + (1 + \frac{1}{p}) \| \mathbf{y} \|^2, \forall p > 0$, $(b)$ is due to Assumption 4.1 and 4.2 and $(c)$ is due to $(1 + \frac{1}{q})^q < e, \forall q > 0$, where $e$ is the natural exponent.

### B.6 PROOF OF THEOREM 4.3

With Assumption 4.1, we have

$$
\mathbb{E}\left[f(\bar{\mathbf{x}}^{r+1})\right] \leq \mathbb{E}\left[f(\bar{\mathbf{x}}^r)\right] - \gamma\eta\mathbb{E}\left\langle \nabla f(\bar{\mathbf{x}}^r), \frac{1}{N}\sum_{i=1}^{N}\sum_{k=0}^{I-1}\mathbf{g}_i(\mathbf{x}_i^{r,k})\right\rangle + \frac{\gamma^2\eta^2 L_g}{2}\mathbb{E}\left\|\frac{1}{N}\sum_{i=1}^{N}\sum_{k=0}^{I-1}\mathbf{g}_i(\mathbf{x}_i^{r,k})\right\|^2
$$

$$
= \mathbb{E}\left[f(\bar{\mathbf{x}}^r)\right] - \gamma\eta\mathbb{E}\left\langle \nabla f(\bar{\mathbf{x}}^r), \frac{1}{N}\sum_{i=1}^{N}\sum_{k=0}^{I-1}\nabla F_i(\mathbf{x}_i^{r,k})\right\rangle + \frac{\gamma^2\eta^2 L_g}{2}\mathbb{E}\left\|\frac{1}{N}\sum_{i=1}^{N}\sum_{k=0}^{I-1}\mathbf{g}_i(\mathbf{x}_i^{r,k})\right\|^2.
$$

$$(\text{B.14})$$

The second term in the RHS of (B.14) can be computed as follows.

$$
-\gamma\eta\mathbb{E}\left\langle \nabla f(\bar{\mathbf{x}}^r), \frac{1}{N}\sum_{i=1}^{N}\sum_{k=0}^{I-1}\nabla F_i(\mathbf{x}_i^{r,k})\right\rangle
$$

$$
= -\frac{\gamma\eta}{I}\mathbb{E}\left\langle I\nabla f(\bar{\mathbf{x}}^r), \frac{1}{N}\sum_{i=1}^{N}\sum_{k=0}^{I-1}\nabla F_i(\mathbf{x}_i^{r,k})\right\rangle
$$

$$
= \frac{\gamma\eta}{2I}\mathbb{E}\left\|\frac{1}{N}\sum_{i=1}^{N}\sum_{k=0}^{I-1}\left(\nabla F_i(\mathbf{x}_i^{r,k}) - \nabla f(\bar{\mathbf{x}}^r)\right)\right\|^2 - I^2\|\nabla f(\bar{\mathbf{x}}^r)\|^2 - \left\|\frac{1}{N}\sum_{i=1}^{N}\sum_{k=0}^{I-1}\nabla F_i(\mathbf{x}_i^{r,k})\right\|^2
$$

$$
= \frac{\gamma\eta}{2I}\left\{\mathbb{E}\left\|\sum_{k=0}^{I-1}\left(\frac{1}{N}\sum_{i=1}^{N}\nabla F_i(\mathbf{x}_i^{r,k}) - \nabla f(\hat{\mathbf{x}}^{r,k})\right) + \sum_{k=0}^{I-1}\left(\nabla f(\hat{\mathbf{x}}^{r,k}) - \nabla f(\bar{\mathbf{x}}^r)\right)\right\|^2\right.
$$

$$
\left. - I^2\mathbb{E}\|\nabla f(\bar{\mathbf{x}}^r)\|^2 - \mathbb{E}\left\|\frac{1}{N}\sum_{i=1}^{N}\sum_{k=0}^{I-1}\nabla F_i(\mathbf{x}_i^{r,k})\right\|^2\right\}
$$

$$
\leq \frac{\gamma\eta}{2I}\left\{2I\sum_{k=0}^{I-1}\mathbb{E}\left\|\frac{1}{N}\sum_{i=1}^{N}\nabla F_i(\mathbf{x}_i^{r,k}) - \nabla f(\hat{\mathbf{x}}^{r,k})\right\|^2 + 2I\sum_{k=0}^{I-1}\mathbb{E}\|\nabla f(\hat{\mathbf{x}}^{r,k}) - \nabla f(\bar{\mathbf{x}}^r)\|^2\right.
$$

$$
\left. - I^2\mathbb{E}\|\nabla f(\bar{\mathbf{x}}^r)\|^2 - \mathbb{E}\left\|\frac{1}{N}\sum_{i=1}^{N}\sum_{k=0}^{I-1}\nabla F_i(\mathbf{x}_i^{r,k})\right\|^2\right\}
$$

$$
\overset{(a)}{\leq} \frac{\gamma\eta}{2I}\left\{\frac{2IL_h^2}{N}\sum_{k=0}^{I-1}\sum_{i=1}^{N}\mathbb{E}\left\|\mathbf{x}_i^{r,k} - \hat{\mathbf{x}}^{r,k}\right\|^2 + 2IL_g^2\sum_{k=0}^{I-1}\mathbb{E}\left\|\hat{\mathbf{x}}^{r,k} - \bar{\mathbf{x}}^r\right\|^2\right.
$$

$$
\left. - I^2\mathbb{E}\|\nabla f(\bar{\mathbf{x}}^r)\|^2 - \mathbb{E}\left\|\frac{1}{N}\sum_{i=1}^{N}\sum_{k=0}^{I-1}\nabla F_i(\mathbf{x}_i^{r,k})\right\|^2\right\},
$$

$$(\text{B.15})$$

where $(a)$ is due to Assumption 4.1 and Assumption 4.2.

The third term in the RHS of (B.14) can be computed as follows.

$$
\frac{\gamma^2\eta^2 L_g}{2}\mathbb{E}\left\|\frac{1}{N}\sum_{i=1}^{N}\sum_{k=0}^{I-1}\mathbf{g}_i(\mathbf{x}_i^{r,k})\right\|^2
$$

$$
= \frac{\gamma^2\eta^2 L_g}{2}\left\|\frac{1}{N}\sum_{i=1}^{N}\sum_{k=0}^{I-1}\nabla F_i(\mathbf{x}_i^{r,k})\right\|^2 + \frac{\gamma^2\eta^2 L_g}{2}\mathbb{E}\left\|\frac{1}{N}\sum_{i=1}^{N}\sum_{k=0}^{I-1}\left[\mathbf{g}_i(\mathbf{x}_i^{r,k}) - \nabla F_i(\mathbf{x}_i^{r,k})\right]\right\|^2
$$

$$
\leq \frac{\gamma^2\eta^2 L_g}{2}\left\|\frac{1}{N}\sum_{i=1}^{N}\sum_{k=0}^{I-1}\nabla F_i(\mathbf{x}_i^{r,k})\right\|^2 + \frac{\gamma^2\eta^2 I L_g\sigma^2}{2N}.
$$

$$(\text{B.16})$$

Substitute (B.15) and (B.16) to (B.14), we have

$$\mathbb{E}\left[f(\bar{\mathbf{x}}^{r+1})\right]$$

$$\leq \mathbb{E}\left[f(\bar{\mathbf{x}}^r)\right] + \frac{\gamma\eta L_h^2}{N}\sum_{k=0}^{I-1}\sum_{i=1}^{N}\mathbb{E}\left\|\mathbf{x}_i^{r,k} - \hat{\mathbf{x}}^{r,k}\right\|^2 + \gamma\eta L_g^2\sum_{k=0}^{I-1}\mathbb{E}\left\|\hat{\mathbf{x}}^{r,k} - \bar{\mathbf{x}}^r\right\|^2$$

$$- \frac{\gamma\eta I}{2}\mathbb{E}\left\|\nabla f(\bar{\mathbf{x}}^r)\right\|^2 - \frac{\gamma\eta}{2}\left(\frac{1}{I} - \gamma\eta L_g\right)\mathbb{E}\left\|\frac{1}{N}\sum_{i=1}^{N}\sum_{k=0}^{I-1}\nabla F_i(\mathbf{x}_i^{r,k})\right\|^2 + \frac{\gamma^2\eta^2 I L_g\sigma^2}{2N}$$

$$\overset{(a)}{\leq} \mathbb{E}\left[f(\bar{\mathbf{x}}^r)\right] + \frac{\gamma\eta L_h^2}{N}\sum_{k=0}^{I-1}\sum_{i=1}^{N}\mathbb{E}\left\|\mathbf{x}_i^{r,k} - \hat{\mathbf{x}}^{r,k}\right\|^2 + \gamma\eta L_g^2\sum_{k=0}^{I-1}\mathbb{E}\left\|\hat{\mathbf{x}}^{r,k} - \bar{\mathbf{x}}^r\right\|^2$$

$$- \frac{\gamma\eta I}{2}\mathbb{E}\left\|\nabla f(\bar{\mathbf{x}}^r)\right\|^2 + \frac{\gamma^2\eta^2 I L_g\sigma^2}{2N}$$

$$\overset{(b)}{\leq} \mathbb{E}\left[f(\bar{\mathbf{x}}^r)\right] + \frac{\gamma\eta L_h^2}{N}\sum_{k=0}^{I-1}\sum_{i=1}^{N}\mathbb{E}\left\|\mathbf{x}_i^{r,k} - \hat{\mathbf{x}}^{r,k}\right\|^2 - \frac{\gamma\eta I}{2}\mathbb{E}\left\|\nabla f(\bar{\mathbf{x}}^r)\right\|^2 + \frac{\gamma^2\eta^2 I L_g\sigma^2}{2N}$$

$$+ \gamma\eta L_g^2 I\left(5(I-1)\frac{\gamma^2\sigma^2}{N} + 30I\gamma^2\sum_{k=0}^{I-1}\frac{L_h^2}{N}\sum_{i=1}^{N}\mathbb{E}\left\|\mathbf{x}_i^{r,k} - \hat{\mathbf{x}}^{r,k}\right\|^2 + 30I(I-1)\gamma^2\mathbb{E}\left\|\nabla f(\bar{\mathbf{x}}^r)\right\|^2\right)$$

$$\leq \mathbb{E}\left[f(\bar{\mathbf{x}}^r)\right] + 2\gamma\eta\sum_{k=0}^{I-1}\frac{L_h^2}{N}\sum_{i=1}^{N}\mathbb{E}\left\|\mathbf{x}_i^{r,k} - \hat{\mathbf{x}}^{r,k}\right\|^2 - \frac{\gamma\eta I}{4}\mathbb{E}\left\|\nabla f(\bar{\mathbf{x}}^r)\right\|^2 + \frac{\gamma^2\eta^2 I L_g\sigma^2}{N}$$

$$\overset{(c)}{\leq} \mathbb{E}\left[f(\bar{\mathbf{x}}^r)\right] + 2\gamma\eta\left[3c(I-1)^3\gamma^2 L_h^2\zeta^2 + c(I-1)^2\gamma^2 L_h^2\sigma^2\right] - \frac{\gamma\eta I}{4}\mathbb{E}\left\|\nabla f(\bar{\mathbf{x}}^r)\right\|^2 + \frac{\gamma^2\eta^2 I L_g\sigma^2}{N},$$
(B.17)

where $(a)$ is due to $\gamma\eta < \frac{1}{4IL}$, $(b)$ is due to Lemma B.3 and $(c)$ is due to Lemma B.2.

Rearrange the above inequality and average over $r$, we obtain

$$\min_{r\in[R]}\mathbb{E}\left\|\nabla f(\bar{\mathbf{x}}^r)\right\|^2 \leq \frac{1}{R}\sum_{r=0}^{R-1}\mathbb{E}\left\|\nabla f(\bar{\mathbf{x}}^r)\right\|^2 \leq \frac{4[f(\mathbf{x}^0) - f^*]}{\gamma\eta RI} + \frac{\gamma\eta L_g\sigma^2}{N}$$
$$+ 24c\gamma^2 L_h^2(I-1)^2\zeta^2 + 8c\gamma^2 L_h^2(I-1)\sigma^2.$$
(B.18)

Then we have

$$\min_{r\in[R]}\mathbb{E}\left\|\nabla f(\bar{\mathbf{x}}^r)\right\|^2 = \mathcal{O}\left(\frac{f(\mathbf{x}^0) - f^*}{\gamma\eta RI} + \frac{\gamma\eta L_g\sigma^2}{N} + \gamma^2 L_h^2(I-1)^2\zeta^2 + \gamma^2 L_h^2(I-1)\sigma^2\right).$$
(B.19)

### B.7 PROOF OF COROLLARY 4.4 AND COROLLARY 4.6

When $I \geq \frac{1}{L_h}$, we let $\gamma = \frac{1}{L_h I}\frac{1}{\sqrt{RIN}}$, then we can verify that

$$\gamma = \frac{1}{L_h I}\frac{1}{\sqrt{RIN}} \leq \frac{1}{\sqrt{RIN}} \leq \frac{1}{30(L_g + L_h)I},$$
(B.20)

which satisfy the condition in Theorem 4.3 and implies $I \leq \frac{RN}{900(L_g + L_h)^2}$. Hence the range of $I$ when $I \geq \frac{1}{L_h}$ is given by

$$I \in \left(\frac{1}{L_h}, \frac{RN}{900(L_g + L_h)^2}\right).$$
(B.21)

In this case, by plugging in $\gamma = \frac{1}{L_h I}\frac{1}{\sqrt{RIN}}$, the last two terms of (12) in Theorem 4.3 can be upper bounded as

$$\gamma^2 L_h^2(I-1)^2\zeta^2 + \gamma^2 L_h^2(I-1)\sigma^2 \leq \frac{\zeta^2}{RIN} + \frac{\sigma^2/I}{RIN} = \frac{\zeta^2 + \sigma^2/I}{RIN}.$$
(B.22)

When $I < \frac{1}{L_h}$, let $\frac{1}{L_h I} \cdot \frac{1}{\sqrt{RIN}}$, then we have

$$\gamma = \frac{1}{\sqrt{RIN}} \leq \frac{1}{30(L_g + L_h)I},$$

which satisfy the condition in Theorem 4.3 and implies $I \leq \frac{RN}{900(L_g + L_h)^2}$. In this case, by plugging in $\gamma = \frac{1}{L_h I} \frac{1}{\sqrt{RIN}}$, the last two terms of (12) in Theorem 4.3 can be upper bounded as

$$\gamma^2 L_h^2 (I-1)^2 \zeta^2 + \gamma^2 L_h^2 (I-1) \sigma^2 \leq \gamma^2 \zeta^2 + \gamma^2 \cdot \frac{\sigma^2}{I} = \frac{\zeta^2 + \sigma^2/I}{RIN}. \tag{B.23}$$

Hence, the condition of $I$ to make the upper bound of the last two terms of (12) in Theorem 4.3 is $I \leq \frac{RN}{900(L_g + L_h)^2}$.

Next, we compute the the first two terms of (12) in Theorem 4.3. When $I \leq \frac{R\sigma^2}{16\mathcal{F}NL_g}$, let $\gamma\eta = \sqrt{\frac{\mathcal{F}N}{RIL_g\sigma^2}}$. Then the first two terms of (12) in Theorem 4.3 can be computed as

$$\frac{f(\boldsymbol{x}_0) - f^*}{\gamma\eta RI} + \frac{\gamma\eta L_g \sigma^2}{N} = \sqrt{\frac{\mathcal{F}L_g\sigma^2}{RIN}}. \tag{B.24}$$

Hence, we finish the proof of Corollary 4.4. Corollary 4.6 can be proved using similar approach.

## B.8 PROOF OF THEOREM 4.5

Consider the partial participation shown in Algorithm 1. In each round, $M$ workers are uniformly sampled with replacement. Then $\forall r, k$, we have

$$\mathbb{E}_{\mathcal{S}_r}\left[\frac{1}{M}\sum_{j\in\mathcal{S}_r}\nabla F_j(\mathbf{x}_j^{r,k})\right] = \frac{1}{N}\sum_{i=1}^{N}\nabla F_i(\mathbf{x}_j^{r,k}). \tag{B.25}$$

With Assumption 4.1, after one round of FedAvg, we have

$$\mathbb{E}\left[f(\bar{\mathbf{x}}^{r+1})\right] \leq \mathbb{E}\left[f(\bar{\mathbf{x}}^{r})\right] - \gamma\eta\mathbb{E}\left\langle\nabla f(\bar{\mathbf{x}}^r), \frac{1}{M}\sum_{i\in\mathcal{S}_r}\sum_{k=0}^{I-1}\mathbf{g}_i(\mathbf{x}_i^{r,k})\right\rangle + \frac{\gamma^2\eta^2 L_g}{2}\mathbb{E}\left\|\frac{1}{M}\sum_{i\in\mathcal{S}_r}\sum_{k=0}^{I-1}\mathbf{g}_i(\mathbf{x}_i^{r,k})\right\|^2$$

$$= \mathbb{E}\left[f(\bar{\mathbf{x}}^{r})\right] - \gamma\eta\mathbb{E}\left\langle\nabla f(\bar{\mathbf{x}}^r), \frac{1}{N}\sum_{i=1}^{N}\sum_{k=0}^{I-1}\nabla F_i(\mathbf{x}_i^{r,k})\right\rangle + \frac{\gamma^2\eta^2 L_g}{2}\mathbb{E}\left\|\frac{1}{M}\sum_{i\in\mathcal{S}_r}\sum_{k=0}^{I-1}\mathbf{g}_i(\mathbf{x}_i^{r,k})\right\|^2. \tag{B.26}$$

It can be seen that the inner-product term is the same as that in (B.15). So we have

$$-\gamma\eta\mathbb{E}\left\langle\nabla f(\bar{\mathbf{x}}^r), \frac{1}{N}\sum_{i=1}^{N}\sum_{k=0}^{I-1}\nabla F_i(\mathbf{x}_i^{r,k})\right\rangle$$

$$\leq \frac{\gamma\eta}{2I}\left\{\frac{2IL_h^2}{N}\sum_{k=0}^{I-1}\sum_{i=1}^{N}\mathbb{E}\left\|\mathbf{x}_i^{r,k} - \hat{\mathbf{x}}^{r,k}\right\|^2 + 2IL_g^2\sum_{k=0}^{I-1}\mathbb{E}\left\|\hat{\mathbf{x}}^{r,k} - \bar{\mathbf{x}}^r\right\|^2\right.$$

$$\left. - I^2\mathbb{E}\left\|\nabla f(\bar{\mathbf{x}}^r)\right\|^2 - \mathbb{E}\left\|\frac{1}{N}\sum_{i=1}^{N}\sum_{k=0}^{I-1}\nabla F_i(\mathbf{x}_i^{r,k})\right\|^2\right\}. \tag{B.27}$$

In this case, $\mathbf{x}_i^{r,k}, i \notin \mathcal{S}_r$ is the virtual local model on worker $i$, which cannot be seen in the system. The virtual local model in mainly used for theoretical analysis. For the third term in the RHS of

(B.26), we have

$$
\mathbb{E}\left\|\frac{1}{M}\sum_{i\in\mathcal{S}_r}\sum_{k=0}^{I-1}\mathbf{g}_i(\mathbf{x}_i^{r,k})\right\|^2
$$

$$
=\mathbb{E}\left\|\frac{1}{M}\sum_{i\in\mathcal{S}_r}\sum_{k=0}^{I-1}\left[\mathbf{g}_i(\mathbf{x}_i^{r,k})-\nabla F_i(\mathbf{x}_i^{r,k})+\nabla F_i(\mathbf{x}_i^{r,k})\right]\right\|^2
$$

$$
=\mathbb{E}\left\|\frac{1}{M}\sum_{i\in\mathcal{S}_r}\sum_{k=0}^{I-1}\left[\mathbf{g}_i(\mathbf{x}_i^{r,k})-\nabla F_i(\mathbf{x}_i^{r,k})\right]\right\|^2+\mathbb{E}\left\|\frac{1}{M}\sum_{i\in\mathcal{S}_r}\sum_{k=0}^{I-1}\nabla F_i(\mathbf{x}_i^{r,k})\right\|^2
$$

$$
\leq\frac{I\sigma^2}{M}+\mathbb{E}\left\|\frac{1}{M}\sum_{i\in\mathcal{S}_r}\sum_{k=0}^{I-1}\nabla F_i(\mathbf{x}_i^{r,k})\right\|^2. \tag{B.28}
$$

For simplicity, we use $Q_i$ to denote the sum of expected gradients of worker $i$ during $r$th round in the following. Then for the second term in the RHS of (B.28), we have

$$
\mathbb{E}\left\|\frac{1}{M}\sum_{i\in\mathcal{S}_r}\sum_{k=0}^{I-1}\nabla F_i(\mathbf{x}_i^{r,k})\right\|^2=\mathbb{E}\left\|\frac{1}{M}\sum_{i\in\mathcal{S}_r}Q_i\right\|^2
$$

$$
=\mathbb{E}\left\|\frac{1}{M}\sum_{i\in\mathcal{S}_r}Q_i-\frac{1}{N}\sum_{j=1}^{N}Q_j+\frac{1}{N}\sum_{j=1}^{N}Q_j\right\|^2
$$

$$
\overset{(a)}{=}\mathbb{E}\left\|\frac{1}{M}\sum_{i\in\mathcal{S}_r}Q_i-\frac{1}{N}\sum_{j=1}^{N}Q_j\right\|^2+\mathbb{E}\left\|\frac{1}{N}\sum_{j=1}^{N}Q_j\right\|^2, \tag{B.29}
$$

where $(a)$ is due to $\mathbb{E}_{\mathcal{S}_r}\left[\frac{1}{M}\sum_{i\in\mathcal{S}_r}Q_i\right]=\frac{1}{N}\sum_{j=1}^{N}Q_j$ by (B.25). Further we have

$$
\mathbb{E}\left\|\frac{1}{M}\sum_{i\in\mathcal{S}_r}Q_i-\frac{1}{N}\sum_{j=1}^{N}Q_j\right\|^2
$$

$$
=\mathbb{E}\left[\frac{1}{M^2}\sum_{i\in\mathcal{S}_r}\left\|Q_i-\frac{1}{N}\sum_{j=1}^{N}Q_j\right\|^2+\frac{1}{M^2}\sum_{i,j\in\mathcal{S}_r,i\neq j}\left\langle Q_i-\frac{1}{N}\sum_{m=1}^{N}Q_m,Q_j-\frac{1}{N}\sum_{m=1}^{N}Q_m\right\rangle\right]
$$

$$
\overset{(a)}{=}\frac{1}{M^2}\sum_{i\in\mathcal{S}_r}\mathbb{E}\left\|Q_i-\frac{1}{N}\sum_{j=1}^{N}Q_j\right\|^2
$$

$$
=\frac{1}{M^2}\sum_{i\in\mathcal{S}_r}\left[\mathbb{E}\left\|Q_i\right\|^2-\mathbb{E}\left\|\frac{1}{N}\sum_{j=1}^{N}Q_j\right\|^2\right]
$$

$$
=\frac{1}{MN}\sum_{i=1}^{N}\mathbb{E}\left\|Q_i\right\|^2-\frac{1}{M}\mathbb{E}\left\|\frac{1}{N}\sum_{j=1}^{N}Q_j\right\|^2, \tag{B.30}
$$

where $(a)$ is due to that the sampling is with replacement so $i$th sampling and $j$th sampling are independent. Then we have

$$
\mathbb{E}\left\|\frac{1}{M}\sum_{i\in\mathcal{S}_r}Q_i\right\|^2=\frac{1}{MN}\sum_{i=1}^{N}\mathbb{E}\left\|Q_i\right\|^2+\frac{M-1}{M}\mathbb{E}\left\|\frac{1}{N}\sum_{j=1}^{N}Q_j\right\|^2. \tag{B.31}
$$

Substituting above results back to (B.28), we obtain

$$\mathbb{E}\left\|\frac{1}{M}\sum_{i\in\mathcal{S}_r}\sum_{k=0}^{I-1}\mathbf{g}_i(\mathbf{x}_i^{r,k})\right\|^2 \leq \frac{I\sigma^2}{M} + \frac{1}{MN}\sum_{i=1}^{N}\mathbb{E}\left\|\sum_{k=0}^{I-1}\nabla F_i(\mathbf{x}_i^{r,k})\right\|^2 + \frac{M-1}{M}\mathbb{E}\left\|\frac{1}{N}\sum_{j=1}^{N}\sum_{k=0}^{I-1}\nabla F_j(\mathbf{x}_j^{r,k})\right\|^2.$$
(B.32)

For the second term of (B.32), we have

$$\mathbb{E}\left\|\sum_{k=0}^{I-1}\nabla F_i(\mathbf{x}_i^{r,k})\right\|^2$$

$$= \mathbb{E}\left\|\sum_{k=0}^{I-1}\nabla F_i(\mathbf{x}_i^{r,k}) - \nabla f(\mathbf{x}_i^{r,k}) + \nabla f(\mathbf{x}_i^{r,k}) - \nabla f(\hat{\mathbf{x}}^{r,k}) + \nabla f(\hat{\mathbf{x}}^{r,k}) - \nabla f(\bar{\mathbf{x}}^r) + \nabla f(\bar{\mathbf{x}}^r)\right\|^2$$

$$\overset{(a)}{\leq} 4I^2\zeta^2 + 4L_g^2 I\sum_{k=0}^{I-1}\mathbb{E}\left\|\mathbf{x}_i^{r,k} - \hat{\mathbf{x}}^{r,k}\right\|^2 + 4L_g^2 I\sum_{k=0}^{I-1}\mathbb{E}\left\|\hat{\mathbf{x}}^{r,k} - \bar{\mathbf{x}}^r\right\|^2 + 4I^2\mathbb{E}\left\|\nabla f(\bar{\mathbf{x}}^r)\right\|^2,$$
(B.33)

where $(a)$ is due to Assumption 3.3 and Assumption 4.1. Substituting back and rearranging, we have

$$\frac{\gamma^2\eta^2 L_g}{2}\mathbb{E}\left\|\frac{1}{M}\sum_{i\in\mathcal{S}_r}\sum_{k=0}^{I-1}\mathbf{g}_i(\mathbf{x}_i^{r,k})\right\|^2 \leq \frac{\gamma^2\eta^2 L_g I\sigma^2}{2M} + \frac{\gamma^2\eta^2 L_g(M-1)}{2M}\mathbb{E}\left\|\frac{1}{N}\sum_{i=1}^{N}\sum_{k=0}^{I-1}\nabla F_i(\mathbf{x}_i^{r,k})\right\|^2$$

$$+ \frac{2\gamma^2\eta^2 L_g I^2\zeta^2}{M} + \frac{2\gamma^2\eta^2 L_g^3 I}{MN}\sum_{i=1}^{N}\sum_{k=0}^{I-1}\mathbb{E}\left\|\mathbf{x}_i^{r,k} - \hat{\mathbf{x}}^{r,k}\right\|^2$$

$$+ \frac{2\gamma^2\eta^2 L_g^3 I}{MN}\sum_{i=1}^{N}\sum_{k=0}^{I-1}\mathbb{E}\left\|\hat{\mathbf{x}}^{r,k} - \bar{\mathbf{x}}^r\right\|^2 + \frac{2\gamma^2\eta^2 L_g I^2}{M}\mathbb{E}\left\|\nabla f(\bar{\mathbf{x}}^r)\right\|^2.$$
(B.34)

Substituting all terms back to (B.26), we have

$$\mathbb{E}\left[f(\bar{\mathbf{x}}^{r+1})\right] \leq \mathbb{E}\left[f(\bar{\mathbf{x}}^r)\right] - \left(\frac{\gamma\eta I}{2} - \frac{2\gamma^2\eta^2 L_g I^2}{M}\right)\mathbb{E}\left\|\nabla f(\bar{\mathbf{x}}^r)\right\|^2$$

$$- \left(\frac{\gamma\eta}{2I} - \frac{\gamma^2\eta^2 L_g(M-1)}{2M}\right)\mathbb{E}\left\|\frac{1}{N}\sum_{i=1}^{N}\sum_{k=0}^{I-1}\nabla F_i(\mathbf{x}_i^{r,k})\right\|^2$$

$$+ \frac{\gamma^2\eta^2 L_g I\sigma^2}{2M} + \frac{2\gamma^2\eta^2 L_g I^2\zeta^2}{M} + \left(\gamma\eta L_h^2 + \frac{2\gamma^2\eta^2 L_g^3 I}{M}\right)\cdot\frac{1}{N}\sum_{k=0}^{I-1}\sum_{i=1}^{N}\mathbb{E}\left\|\mathbf{x}_i^{r,k} - \hat{\mathbf{x}}^{r,k}\right\|^2$$

$$+ \left(\gamma\eta L_g^2 + \frac{2\gamma^2\eta^2 L_g^3 I}{M}\right)\sum_{k=0}^{I-1}\mathbb{E}\left\|\hat{\mathbf{x}}^{r,k} - \bar{\mathbf{x}}^r\right\|^2.$$
(B.35)

With $\gamma\eta \leq \frac{1}{4IL_g}$ and Lemma B.3, we have

$$\mathbb{E}\left[f(\bar{\mathbf{x}}^{r+1})\right] \leq \mathbb{E}\left[f(\bar{\mathbf{x}}^r)\right] - \left(\frac{\gamma\eta I}{2} - \frac{2\gamma^2\eta^2 L_g I^2}{M}\right)\mathbb{E}\left\|\nabla f(\bar{\mathbf{x}}^r)\right\|^2 + \frac{\gamma^2\eta^2 L_g I\sigma^2}{2M} + \frac{2\gamma^2\eta^2 L_g I^2\zeta^2}{M}$$

$$+ \left(\gamma\eta L_h^2 + \frac{2\gamma^2\eta^2 L_g^3 I}{M}\right)\cdot\frac{1}{N}\sum_{k=0}^{I-1}\sum_{i=1}^{N}\mathbb{E}\left\|\mathbf{x}_i^{r,k} - \hat{\mathbf{x}}^{r,k}\right\|^2$$

$$+ \left(\gamma\eta I L_g^2 + \frac{2\gamma^2\eta^2 L_g^3 I^2}{M}\right)$$

$$\cdot\left(5(I-1)\cdot\frac{\gamma^2\sigma^2}{N} + 30I\gamma^2\sum_{k=0}^{I-1}\frac{L_h^2}{N}\sum_{i=1}^{N}\mathbb{E}\left\|\mathbf{x}_i^{r,k} - \hat{\mathbf{x}}^{r,k}\right\|^2 + 30I(I-1)\gamma^2\mathbb{E}\left\|\nabla f(\bar{\mathbf{x}}^r)\right\|^2\right).$$
(B.36)

With $\gamma\eta \leq \frac{1}{4IL_g}$ and $\gamma < \frac{1}{30(L_g+L_h)I}$, we have

$$\mathbb{E}\left[f(\bar{\mathbf{x}}^{r+1})\right] \leq \mathbb{E}\left[f(\bar{\mathbf{x}}^r)\right] - \frac{\gamma\eta I}{8}\mathbb{E}\left\|\nabla f(\bar{\mathbf{x}}^r)\right\|^2 + \frac{\gamma^2\eta^2 L_g I\sigma^2}{2M} + \frac{2\gamma^2\eta^2 L_g I^2\zeta^2}{M} + \frac{\gamma\eta\sigma^2}{N}$$
$$+ \left(2\gamma\eta L_h^2 + \frac{2\gamma^2\eta^2 L_g^3 I}{M}\right) \cdot \frac{1}{N}\sum_{k=0}^{I-1}\sum_{i=1}^{N}\mathbb{E}\left\|\mathbf{x}_i^{r,k} - \hat{\mathbf{x}}^{r,k}\right\|^2. \tag{B.37}$$

With Lemma B.2, we have

$$\mathbb{E}\left[f(\bar{\mathbf{x}}^{r+1})\right] \leq \mathbb{E}\left[f(\bar{\mathbf{x}}^r)\right] - \frac{\gamma\eta I}{8}\mathbb{E}\left\|\nabla f(\bar{\mathbf{x}}^r)\right\|^2 + \frac{\gamma^2\eta^2 L_g I\sigma^2}{2M} + \frac{2\gamma^2\eta^2 L_g I^2\zeta^2}{M} + \frac{\gamma\eta\sigma^2}{N}$$
$$+ \left(2\gamma\eta L_h^2 + \frac{2\gamma^2\eta^2 L_g^3 I}{M}\right) \cdot \left(3c(I-1)^3\gamma^2\zeta^2 + c(I-1)^2\gamma^2\sigma^2\right). \tag{B.38}$$

Then we obtain

$$\min_{r\in[R]}\mathbb{E}\left\|\nabla f(\bar{\mathbf{x}}^r)\right\|^2 \leq \frac{1}{R}\sum_{r=0}^{R-1}\mathbb{E}\left\|\nabla f(\bar{\mathbf{x}}^r)\right\|^2 \leq \frac{8(f^0-f^*)}{\gamma\eta IR} + \frac{4\gamma\eta L_g\sigma^2}{M} + \frac{16\gamma\eta L_g I\zeta^2}{M}$$
$$+ \left(16L_h^2 + \frac{16\gamma\eta L_g^3 I}{M}\right) \cdot \left(3c(I-1)^2\gamma^2\zeta^2 + c(I-1)\gamma^2\sigma^2\right). \tag{B.39}$$

Rearrange,

$$\min_{r\in[R]}\mathbb{E}\left\|\nabla f(\bar{\mathbf{x}}^r)\right\|^2 = \mathcal{O}\left(\frac{(f^0-f^*)}{\gamma\eta IR} + \frac{\gamma\eta L_g\sigma^2}{M} + \frac{\gamma\eta L_g I\zeta^2}{M} + \gamma^2 L_h^2(I-1)\sigma^2 + \gamma^2 L_h^2(I-1)^2\zeta^2\right). \tag{B.40}$$

### B.9 PROOF OF PROPOSITION 5.1

First, using $\nabla f(\mathbf{x}) = \frac{1}{N}\sum_{i=1}^{N}\nabla F_i(\mathbf{x})$, it is straightforward to show that Assumption 3.1 implies Assumption 4.1 holds by choosing $L_g = \tilde{L}$.

Second, we can see that

$$\left\|\frac{1}{N}\sum_{i=1}^{N}\nabla F_i(\mathbf{x}_i) - \nabla f(\bar{\mathbf{x}})\right\|^2 = \left\|\frac{1}{N}\sum_{i=1}^{N}[]\nabla F_i(\mathbf{x}_i) - \nabla F_i(\bar{\mathbf{x}})]\right\|^2$$
$$\leq \frac{1}{N}\sum_{i=1}^{N}\left\|\nabla F_i(\mathbf{x}_i) - \nabla F_i(\bar{\mathbf{x}})\right\|^2$$
$$\overset{(a)}{\leq} \frac{\tilde{L}^2}{N}\sum_{i=1}^{N}\left\|\mathbf{x}_i - \bar{\mathbf{x}}\right\|^2, \tag{B.41}$$

where $(a)$ is due to Assumption 3.1. By choosing $L_h = \tilde{L}$, Assumption 4.2 holds. $\qquad\square$

### B.10 PROOF OF PROPOSITION 5.2

For quadratic functions, we have

$$\nabla F_i(\mathbf{x}) = \mathbf{A}_i\mathbf{x} + \mathbf{b}_i, \mathbf{x} \in \mathbb{R}^d. \tag{B.42}$$

Let $\bar{\mathbf{A}} := \frac{1}{N} \sum_{i=1}^{N} \mathbf{A}_i$ and $\bar{\mathbf{b}} := \frac{1}{N} \sum_{i=1}^{N} \mathbf{b}_i$. We have

$$\left\| \frac{1}{N} \sum_{i=1}^{N} \nabla F_i(\mathbf{x}_i) - \nabla f(\bar{\mathbf{x}}) \right\|^2$$

$$= \left\| \frac{1}{N} \sum_{i=1}^{N} (\mathbf{A}_i \mathbf{x}_i + \mathbf{b}_i) - (\bar{\mathbf{A}} \bar{\mathbf{x}} + \bar{\mathbf{b}}) \right\|^2$$

$$= \left\| \frac{1}{N} \sum_{i=1}^{N} \mathbf{A}_i \mathbf{x}_i - \bar{\mathbf{A}} \bar{\mathbf{x}} \right\|^2$$

$$= \left\| \frac{1}{N^2} \sum_{i=1}^{N} \sum_{j=1}^{N} (\mathbf{A}_i \mathbf{x}_i - \mathbf{A}_i \mathbf{x}_j) \right\|^2$$

$$= \left\| \frac{1}{N^2} \sum_{i=1}^{N} \sum_{j=1}^{N} \left[ (\mathbf{A}_i - \bar{\mathbf{A}})(\mathbf{x}_i - \bar{\mathbf{x}}) - (\mathbf{A}_i - \bar{\mathbf{A}})(\mathbf{x}_j - \bar{\mathbf{x}}) \right] \right\|^2$$

$$\leq 2 \left\| \frac{1}{N^2} \sum_{i=1}^{N} \sum_{j=1}^{N} (\mathbf{A}_i - \bar{\mathbf{A}})(\mathbf{x}_i - \bar{\mathbf{x}}) \right\|^2 + 2 \left\| \frac{1}{N^2} \sum_{i=1}^{N} \sum_{j=1}^{N} (\mathbf{A}_i - \bar{\mathbf{A}})(\mathbf{x}_j - \bar{\mathbf{x}}) \right\|^2$$

$$\leq \frac{2}{N^2} \sum_{i=1}^{N} \sum_{j=1}^{N} \left\| (\mathbf{A}_i - \bar{\mathbf{A}})(\mathbf{x}_i - \bar{\mathbf{x}}) \right\|^2 + \frac{2}{N^2} \sum_{i=1}^{N} \sum_{j=1}^{N} \left\| (\mathbf{A}_i - \bar{\mathbf{A}})(\mathbf{x}_j - \bar{\mathbf{x}}) \right\|^2$$

$$\overset{(a)}{\leq} \frac{2|\lambda_{\text{diff}}|_{\max}^2}{N^2} \sum_{i=1}^{N} \sum_{j=1}^{N} \left\| \mathbf{x}_i - \bar{\mathbf{x}} \right\|^2 + \frac{2|\lambda_{\text{diff}}|_{\max}^2}{N^2} \sum_{i=1}^{N} \sum_{j=1}^{N} \left\| \mathbf{x}_j - \bar{\mathbf{x}} \right\|^2$$

$$\leq \frac{4|\lambda_{\text{diff}}|_{\max}^2}{N^2} \sum_{i=1}^{N} \left\| \mathbf{x}_i - \bar{\mathbf{x}} \right\|^2, \tag{B.43}$$

where $(a)$ is due to Cauchy's inequality and $|\lambda_{\text{diff}}| := \max_{i \in [N]} |\lambda(\mathbf{A}_i - \bar{\mathbf{A}})|$.

## B.11 PROOF OF PROPOSITION 5.3

Recall that $\hat{\mathbf{x}}^{r,k}$ is the virtual averaged model defined in (5). During one local iteration, we have

$$\mathbb{E}[\hat{\mathbf{x}}^{r,k+1} | \hat{\mathbf{x}}^{r,k}] = \hat{\mathbf{x}}^{r,k} - \gamma \cdot \frac{1}{N} \sum_{i=1}^{N} \nabla F_i(\mathbf{x}_i^{r,k}). \tag{B.44}$$

Using (6), if we use centralized update at this iteration, we have

$$\mathbb{E}[\mathbf{x}_c^{r,k+1} | \hat{\mathbf{x}}^{r,k}] = \hat{\mathbf{x}}^{r,k} - \gamma \nabla f(\hat{\mathbf{x}}^{r,k}). \tag{B.45}$$

Using Assumption 4.2, we obtain

$$\left\| \mathbb{E}[\hat{\mathbf{x}}^{r,k+1} | \hat{\mathbf{x}}^{r,k}] - \mathbb{E}[\mathbf{x}_c^{r,k+1} | \hat{\mathbf{x}}^{r,k}] \right\|^2 = \gamma^2 \left\| \frac{1}{N} \sum_{i=1}^{N} \nabla F_i(\mathbf{x}_i^{r,k}) - \nabla f(\hat{\mathbf{x}}^{r,k}) \right\|^2$$

$$\leq \gamma^2 \cdot \frac{L_h^2}{N} \sum_{i=1}^{N} \left\| \mathbf{x}_i^{r,k} - \hat{\mathbf{x}}^{r,k} \right\|^2. \tag{B.46}$$

## B.12 PROOF OF THEOREM 5.5

It can be observed that for quadratic objective functions when $\mathbf{A}_i = \mathbf{A}, \forall i$, we have $L_h = 0$ and $L_g = |\lambda(\mathbf{A})|$. In this section, we use $t$ to denote the index of the total number of iterations and $\hat{\mathbf{x}}^t$ is

defined as

$$\hat{\mathbf{x}}^t = \begin{cases} \hat{\mathbf{x}}^{r,k}, t = rI + k, k \neq 0, \\ \bar{\mathbf{x}}^r, t = rI. \end{cases}$$

With Assumption 4.1, after one local iteration, we have

$$\mathbb{E}\left[f(\hat{\mathbf{x}}^{t+1})\right] \leq \mathbb{E}\left[f(\hat{\mathbf{x}}^t)\right] - \gamma\mathbb{E}\left\langle\nabla f(\hat{\mathbf{x}}^t), \frac{1}{N}\sum_{i=1}^N \mathbf{g}_i(\mathbf{x}_i^t)\right\rangle + \frac{\gamma^2 L_g}{2}\mathbb{E}\left\|\frac{1}{N}\sum_{i=1}^N \mathbf{g}_i(\mathbf{x}_i^t)\right\|^2$$

$$= \mathbb{E}\left[f(\hat{\mathbf{x}}^t)\right] - \gamma\mathbb{E}\left\langle\nabla f(\hat{\mathbf{x}}^t), \frac{1}{N}\sum_{i=1}^N \nabla F_i(\mathbf{x}_i^t)\right\rangle + \frac{\gamma^2 L_g}{2}\mathbb{E}\left\|\frac{1}{N}\sum_{i=1}^N \mathbf{g}_i(\mathbf{x}_i^t)\right\|^2. \tag{B.47}$$

For the second term in the RHS of (B.47), we have

$$-\gamma\mathbb{E}\left\langle\nabla f(\hat{\mathbf{x}}^t), \frac{1}{N}\sum_{i=1}^N \nabla F_i(\mathbf{x}_i^t)\right\rangle$$

$$= \frac{\gamma}{2}\left(\mathbb{E}\left\|\frac{1}{N}\sum_{i=1}^N \nabla F_i(\mathbf{x}_i^t) - \nabla f(\hat{\mathbf{x}}^t)\right\|^2 - \mathbb{E}\left\|\nabla f(\hat{\mathbf{x}}^t)\right\|^2 - \mathbb{E}\left\|\frac{1}{N}\sum_{i=1}^N \nabla F_i(\mathbf{x}_i^t)\right\|^2\right)$$

$$\leq \frac{\gamma}{2}\left(\frac{L_h^2}{N}\sum_{i=1}^N \mathbb{E}\left\|\mathbf{x}_i^t - \hat{\mathbf{x}}^t\right\|^2 - \mathbb{E}\left\|\nabla f(\hat{\mathbf{x}}^t)\right\|^2 - \mathbb{E}\left\|\frac{1}{N}\sum_{i=1}^N \nabla F_i(\mathbf{x}_i^t)\right\|^2\right). \tag{B.48}$$

For the third term of (B.47), we have

$$\frac{\gamma^2 L_g}{2}\mathbb{E}\left\|\frac{1}{N}\sum_{i=1}^N \mathbf{g}_i(\mathbf{x}_i^t)\right\|^2 \leq \frac{\gamma^2 L_g}{2}\mathbb{E}\left\|\frac{1}{N}\sum_{i=1}^N \nabla F_i(\mathbf{x}_i^t)\right\|^2 + \frac{\gamma^2 L_g \sigma^2}{2N}. \tag{B.49}$$

Substitute (B.48) and (B.49) back to (B.47), we obtain

$$\mathbb{E}\left[f(\hat{\mathbf{x}}^{t+1})\right]$$

$$\leq \mathbb{E}\left[f(\hat{\mathbf{x}}^t)\right] + \frac{\gamma L_h^2}{2N}\sum_{i=1}^N \mathbb{E}\left\|\mathbf{x}_i^t - \hat{\mathbf{x}}^t\right\|^2 - \frac{\gamma}{2}\mathbb{E}\left\|\nabla f(\hat{\mathbf{x}}^t)\right\|^2$$

$$- \left(\frac{\gamma}{2} - \frac{\gamma^2 L_g}{2}\right)\mathbb{E}\left\|\frac{1}{N}\sum_{i=1}^N \nabla F_i(\mathbf{x}_i^t)\right\|^2 + \frac{\gamma^2 L_g \sigma^2}{2N}$$

$$\overset{(a)}{\leq} \mathbb{E}\left[f(\hat{\mathbf{x}}^t)\right] + \frac{\gamma L_h^2}{2N}\sum_{i=1}^N \mathbb{E}\left\|\mathbf{x}_i^t - \hat{\mathbf{x}}^t\right\|^2 - \frac{\gamma}{2}\mathbb{E}\left\|\nabla f(\hat{\mathbf{x}}^t)\right\|^2 + \frac{\gamma^2 L_g \sigma^2}{2N}, \tag{B.50}$$

where $(a)$ is due to $\gamma < \frac{1}{L_g}$. Rearrange the above inequality with $L_h = 0$, we have

$$\mathbb{E}\left\|\nabla f(\hat{\mathbf{x}}^t)\right\|^2 \leq \frac{2\mathbb{E}\left[f(\hat{\mathbf{x}}^t)\right] - 2\mathbb{E}f(\hat{\mathbf{x}}^{t+1})}{\gamma} + \frac{L_h^2}{N}\sum_{i=1}^N \mathbb{E}\left\|\mathbf{x}_i^t - \hat{\mathbf{x}}^t\right\|^2 + \frac{\gamma L_g \sigma^2}{N}$$

$$= \frac{2\mathbb{E}\left[f(\hat{\mathbf{x}}^t)\right] - 2\mathbb{E}f(\hat{\mathbf{x}}^{t+1})}{\gamma} + \frac{\gamma L_g \sigma^2}{N}. \tag{B.51}$$

Take the average over $t$ on both sides, we obtain

$$\min_{t\in[T]}\mathbb{E}\left\|\nabla f(\hat{\mathbf{x}}^t)\right\|^2 \leq \frac{1}{T}\sum_{t=0}^{T-1}\mathbb{E}\left\|\nabla f(\hat{\mathbf{x}}^t)\right\|^2 \leq \frac{2f(\hat{\mathbf{x}}^t) - 2f^*}{\gamma T} + \frac{\gamma L_g \sigma^2}{N}. \tag{B.52}$$

### B.13 PROOF OF THEOREM A.1

For quadratic objective functions, the global objective functions is

$$f(\mathbf{x}) = \frac{1}{2}\mathbf{x}^T \mathbf{A}\mathbf{x} + \mathbf{b}^T \mathbf{x} + c. \tag{B.53}$$

The local objective function of worker $i$ is

$$F_i(\mathbf{x}) = \frac{1}{2}\mathbf{x}^T \mathbf{A}_i \mathbf{x} + \mathbf{b}_i^T \mathbf{x} + c_i, \tag{B.54}$$

where $\mathbf{A} = \frac{1}{N}\sum_{i=1}^{N}\mathbf{A}_i$, $\mathbf{b} = \frac{1}{N}\sum_{i=1}^{N}\mathbf{b}_i$ and $c = \frac{1}{N}\sum_{i=1}^{N}c_i$. The local stochastic gradient is

$$\mathbf{g}_i(\mathbf{x}) = \mathbf{A}_i\mathbf{x} + \mathbf{b}_i + \xi_i, \tag{B.55}$$

where $\mathbb{E}[\xi] = 0$ and $\mathbb{E}[\xi^2] = \sigma^2$. By Assumption 3.3, we have

$$\|\nabla F_i(\mathbf{x}) - \nabla f(\mathbf{x})\|^2 = \|(\mathbf{A}_i - \mathbf{A})\mathbf{x} + (\mathbf{b}_i - \mathbf{b})\|^2 \le \zeta^2, \forall i, \mathbf{x}. \tag{B.56}$$

By Proposition 5.2, we have $\tilde{L} = \max_i |\lambda(\mathbf{A}_i)|$, $L_g = |\lambda(\mathbf{A})|$ and $L_h = 2\max_i |\lambda(\mathbf{A} - \mathbf{A}_i)|$.

Now we start the proof. During local updates, we have

$$\mathbf{x}_i^{r,k+1} = \mathbf{x}_i^{r,k} - \gamma(\mathbf{A}_i\mathbf{x}_i^{r,k} + \mathbf{b}_i + \xi_{i,k}) = (\mathbf{I} - \gamma\mathbf{A}_i)\mathbf{x}_i^{r,k} - \gamma\mathbf{b}_i - \gamma\xi_{i,k}$$

$$= (\mathbf{I} - \gamma\mathbf{A}_i)^k\bar{\mathbf{x}}^r - \gamma\sum_{l=0}^{k-1}(\mathbf{I} - \gamma\mathbf{A}_i)^l(\mathbf{b}_i + \xi_{i,l})$$

$$= (\mathbf{I} - \gamma\mathbf{A}_i)^k\bar{\mathbf{x}}^r - \gamma\sum_{l=0}^{k-1}(\mathbf{I} - \gamma\mathbf{A}_i)^l\mathbf{b}_i - \gamma\sum_{l=0}^{k-1}(\mathbf{I} - \gamma\mathbf{A}_i)^l\xi_{i,l}$$

$$\overset{(a)}{=} \left[\mathbf{I} - \gamma\sum_{l=0}^{k-1}(\mathbf{I} - \gamma\mathbf{A}_i)^k\mathbf{A}_i\right]\bar{\mathbf{x}}^r - \gamma\sum_{l=0}^{k-1}(\mathbf{I} - \gamma\mathbf{A}_i)^l\mathbf{b}_i - \gamma\sum_{l=0}^{k-1}(\mathbf{I} - \gamma\mathbf{A}_i)^l\xi_{i,l}$$

$$= \bar{\mathbf{x}}^r - \gamma\sum_{l=0}^{k-1}(\mathbf{I} - \gamma\mathbf{A}_i)^l[\mathbf{A}_i\bar{\mathbf{x}}^r + \mathbf{b}_i] - \gamma\sum_{l=0}^{k-1}(\mathbf{I} - \gamma\mathbf{A}_i)^l\xi_{i,l}$$

$$= \bar{\mathbf{x}}^r - \gamma\sum_{l=0}^{k-1}(\mathbf{I} - \gamma\mathbf{A}_i)^l\nabla F_i(\bar{\mathbf{x}}^r) - \gamma\sum_{l=0}^{k-1}(\mathbf{I} - \gamma\mathbf{A}_i)^l\xi_{i,l}, \tag{B.57}$$

where $(a)$ is due to $\sum_{l=0}^{k-1}(\mathbf{I} - \gamma\mathbf{A}_i)^l = \frac{1}{\gamma}\left[\mathbf{I} - (\mathbf{I} - \gamma\mathbf{A}_i)^k\right]\mathbf{A}_i^{-1}$.

Then for the model divergence, we have

$$\mathbb{E}\left\|\mathbf{x}_i^{r,k} - \hat{\mathbf{x}}^{r,k}\right\|^2$$

$$\overset{(a)}{=} \mathbb{E}\left\|\gamma\sum_{l=0}^{k-1}(\mathbf{I} - \gamma\mathbf{A}_i)^l\nabla F_i(\bar{\mathbf{x}}^r) - \gamma\cdot\frac{1}{N}\sum_{j=1}^{N}\sum_{l=0}^{k-1}(\mathbf{I} - \gamma\mathbf{A}_j)^l\nabla F_j(\bar{\mathbf{x}}^r)\right\|^2$$

$$+ \mathbb{E}\left\|\gamma\sum_{l=0}^{k-1}(\mathbf{I} - \gamma\mathbf{A}_i)^l\xi_{i,l} - \gamma\cdot\frac{1}{N}\sum_{j=1}^{N}\sum_{l=0}^{k-1}(\mathbf{I} - \gamma\mathbf{A}_j)^l\xi_{j,l}\right\|^2, \tag{B.58}$$

where $(a)$ is due to $\mathbb{E}[\xi_{i,l}] = 0$.

For the first term in the RHS of (B.58), with $\gamma \leq \frac{1}{|\lambda(\mathbf{A}_i)|}$, we have

$$
\mathbb{E}\left\| \gamma \sum_{l=0}^{k-1}(\boldsymbol{I}-\gamma\mathbf{A}_i)^l \nabla F_i(\bar{\mathbf{x}}^r) - \gamma \cdot \frac{1}{N}\sum_{j=1}^{N}\sum_{l=0}^{k-1}(\boldsymbol{I}-\gamma\mathbf{A}_j)^l \nabla F_j(\bar{\mathbf{x}}^r) \right\|^2
$$

$$
= \mathbb{E}\left\| \gamma \sum_{l=0}^{k-1}(\boldsymbol{I}-\gamma\mathbf{A}_i)^l \nabla F_i(\bar{\mathbf{x}}^r) - \gamma \sum_{l=0}^{k-1}(\boldsymbol{I}-\gamma\mathbf{A}_i)^l \nabla f(\bar{\mathbf{x}}^r) + \gamma \sum_{l=0}^{k-1}(\boldsymbol{I}-\gamma\mathbf{A}_i)^l \nabla f(\bar{\mathbf{x}}^r) \right.
$$

$$
- \gamma \sum_{l=0}^{k-1}(\boldsymbol{I}-\gamma\mathbf{A})^l \nabla f(\bar{\mathbf{x}}^r) + \gamma \sum_{l=0}^{k-1}(\boldsymbol{I}-\gamma\mathbf{A})^l \nabla f(\bar{\mathbf{x}}^r) - \gamma \cdot \frac{1}{N}\sum_{j=1}^{N}\sum_{l=0}^{k-1}(\boldsymbol{I}-\gamma\mathbf{A}_j)^l \nabla f(\bar{\mathbf{x}}^r)
$$

$$
\left. + \gamma \cdot \frac{1}{N}\sum_{j=1}^{N}\sum_{l=0}^{k-1}(\boldsymbol{I}-\gamma\mathbf{A}_j)^l \nabla f(\bar{\mathbf{x}}^r) - \gamma \cdot \frac{1}{N}\sum_{j=1}^{N}\sum_{l=0}^{k-1}(\boldsymbol{I}-\gamma\mathbf{A}_j)^l \nabla F_j(\bar{\mathbf{x}}^r) \right\|^2
$$

$$
\leq 4\mathbb{E}\left\| \gamma \sum_{l=0}^{k-1}(\boldsymbol{I}-\gamma\mathbf{A}_i)^l \nabla F_i(\bar{\mathbf{x}}^r) - \gamma \sum_{l=0}^{k-1}(\boldsymbol{I}-\gamma\mathbf{A}_i)^l \nabla f(\bar{\mathbf{x}}^r) \right\|^2
$$

$$
+ 4\mathbb{E}\left\| \gamma \sum_{l=0}^{k-1}(\boldsymbol{I}-\gamma\mathbf{A}_i)^l \nabla f(\bar{\mathbf{x}}^r) - \gamma \sum_{l=0}^{k-1}(\boldsymbol{I}-\gamma\mathbf{A})^l \nabla f(\bar{\mathbf{x}}^r) \right\|^2
$$

$$
+ 4\mathbb{E}\left\| \gamma \sum_{l=0}^{k-1}(\boldsymbol{I}-\gamma\mathbf{A})^l \nabla f(\bar{\mathbf{x}}^r) - \gamma \cdot \frac{1}{N}\sum_{j=1}^{N}\sum_{l=0}^{k-1}(\boldsymbol{I}-\gamma\mathbf{A}_j)^l \nabla f(\bar{\mathbf{x}}^r) \right\|^2
$$

$$
+ 4\mathbb{E}\left\| \gamma \cdot \frac{1}{N}\sum_{j=1}^{N}\sum_{l=0}^{k-1}(\boldsymbol{I}-\gamma\mathbf{A}_j)^l \nabla f(\bar{\mathbf{x}}^r) - \gamma \cdot \frac{1}{N}\sum_{j=1}^{N}\sum_{l=0}^{k-1}(\boldsymbol{I}-\gamma\mathbf{A}_j)^l \nabla F_j(\bar{\mathbf{x}}^r) \right\|^2. \tag{B.59}
$$

For the first term in RHS of (B.59), we have

$$
4\mathbb{E}\left\| \gamma \sum_{l=0}^{k-1}(\boldsymbol{I}-\gamma\mathbf{A}_i)^l \nabla F_i(\bar{\mathbf{x}}^r) - \gamma \sum_{l=0}^{k-1}(\boldsymbol{I}-\gamma\mathbf{A}_i)^l \nabla f(\bar{\mathbf{x}}^r) \right\|^2
$$

$$
\leq 4\gamma^2 k \sum_{l=0}^{k-1}\left\|(\boldsymbol{I}-\gamma\mathbf{A}_i)^l\right\|^2 \mathbb{E}\left\|\nabla F_i(\bar{\mathbf{x}}^r) - \nabla f(\bar{\mathbf{x}}^r)\right\|^2
$$

$$
\overset{(a)}{\leq} 4\gamma^2 k^2 \zeta^2, \tag{B.60}
$$

where $(a)$ is due to $\|\boldsymbol{I}-\gamma\mathbf{A}_i\|^2 \leq 1$ since we have $\gamma < \frac{1}{L}$.

For the second term in RHS of (B.59), we have

$$
4\mathbb{E}\left\| \gamma \sum_{l=0}^{k-1}(\boldsymbol{I}-\gamma\mathbf{A}_i)^l \nabla f(\bar{\mathbf{x}}^r) - \gamma \sum_{l=0}^{k-1}(\boldsymbol{I}-\gamma\mathbf{A})^l \nabla f(\bar{\mathbf{x}}^r) \right\|^2
$$

$$
\leq 4\gamma^2 k \sum_{l=0}^{k-1}\left\|(\boldsymbol{I}-\gamma\mathbf{A}_i)^l - (\boldsymbol{I}-\gamma\mathbf{A})^l\right\|^2 \mathbb{E}\left\|\nabla f(\bar{\mathbf{x}}^r)\right\|^2
$$

$$
\overset{(a)}{\leq} 4\gamma^2 k \sum_{l=0}^{k-1}\gamma^2 L_h^2 l^2 \mathbb{E}\left\|\nabla f(\bar{\mathbf{x}}^r)\right\|^2
$$

$$
= 4\gamma^4 L_h^2 k^4 \mathbb{E}\left\|\nabla f(\bar{\mathbf{x}}^r)\right\|^2, \tag{B.61}
$$

where $(a)$ is due to

$$
\left\|(\boldsymbol{I}-\gamma\mathbf{A}_i)^l - (\boldsymbol{I}-\gamma\mathbf{A})^l\right\|^2 \overset{(b)}{\leq} l^2 \left\|\boldsymbol{I}-\gamma\mathbf{A}_i - \boldsymbol{I} + \gamma\mathbf{A}\right\|^2 = \gamma^2 l^2 \left\|\mathbf{A}_i - \mathbf{A}\right\|^2 \leq \gamma^2 L_h^2 l^2. \tag{B.62}
$$

Now we prove $(b)$. Let $\mathbf{B}_i = \boldsymbol{I} - \gamma \mathbf{A}_i$, we have $\|\mathbf{B}_i^l\| \leq 1$ since $\gamma \leq \frac{1}{L}$. When $l$ is an even, we have

$$
\|\mathbf{B}_i^l - \mathbf{B}^l\| = \|(\mathbf{B}_i^{\frac{l}{2}} - \mathbf{B}^{\frac{l}{2}})(\mathbf{B}_i^{\frac{l}{2}} + \mathbf{B}^{\frac{l}{2}})\| \leq \|\mathbf{B}_i^{\frac{l}{2}} - \mathbf{B}^{\frac{l}{2}}\|\|\mathbf{B}_i^{\frac{l}{2}} + \mathbf{B}^{\frac{l}{2}}\|
$$
$$
\leq \|\mathbf{B}_i^{\frac{l}{2}} - \mathbf{B}^{\frac{l}{2}}\|(\|\mathbf{B}_i^{\frac{l}{2}}\| + \|\mathbf{B}^{\frac{l}{2}}\|) \leq 2\|\mathbf{B}_i^{\frac{l}{2}} - \mathbf{B}^{\frac{l}{2}}\|. \tag{B.63}
$$

When $l$ is odd, we have

$$
\|\mathbf{B}_i^l - \mathbf{B}^l\| = \|(\mathbf{B}_i - \mathbf{B})(\mathbf{B}_i^{l-1} + \mathbf{B}_i^{l-2}\mathbf{B} + \mathbf{B}_i^{l-3}\mathbf{B}^2 + \ldots + \mathbf{B}^{l-1})\|
$$
$$
\leq \|\mathbf{B}_i - \mathbf{B}\|\|\mathbf{B}_i^{l-1} + \mathbf{B}_i^{l-2}\mathbf{B} + \mathbf{B}_i^{l-3}\mathbf{B}^2 + \ldots + \mathbf{B}^{l-1}\|
$$
$$
\leq \|\mathbf{B}_i - \mathbf{B}\|(\|\mathbf{B}_i^{l-1}\| + \|\mathbf{B}_i^{l-2}\|\|\mathbf{B}\| + \|\mathbf{B}_i^{l-3}\|\|\mathbf{B}^2\| + \ldots + \|\mathbf{B}^{l-1}\|)
$$
$$
\leq l\|\mathbf{B}_i - \mathbf{B}\|. \tag{B.64}
$$

By recursion, when $l$ is even, we always have $\|\mathbf{B}_i^l - \mathbf{B}^l\| \leq l\|\mathbf{B}_i - \mathbf{B}\|$.

Taking all terms back to (B.59), we can obtain

$$
\mathbb{E}\left\|\gamma\sum_{l=0}^{k-1}(\boldsymbol{I} - \gamma\mathbf{A}_i)^l \nabla F_i(\bar{\mathbf{x}}^r) - \gamma \cdot \frac{1}{N}\sum_{j=1}^{N}\sum_{l=0}^{k-1}(\boldsymbol{I} - \gamma\mathbf{A}_j)^l \nabla F_j(\bar{\mathbf{x}}^r)\right\|^2
$$
$$
\leq 8\gamma^2(k-1)^2\zeta^2 + 8\gamma^4 k^4 L_h^2 \mathbb{E}\|\nabla f(\bar{\mathbf{x}}^r)\|^2. \tag{B.65}
$$

For the second term in RHS of (B.58), we have

$$
\mathbb{E}\left\|\gamma\sum_{l=0}^{k-1}(\boldsymbol{I} - \gamma\mathbf{A}_i)^l \xi_{i,l} - \gamma \cdot \frac{1}{N}\sum_{j=1}^{N}\sum_{l=0}^{k-1}(\boldsymbol{I} - \gamma\mathbf{A}_j)^l \xi_{j,l}\right\|^2
$$
$$
= \mathbb{E}\left\|\gamma\sum_{l=0}^{k-1}(\boldsymbol{I} - \gamma\mathbf{A}_i)^l \xi_{i,l}\right\|^2 + \mathbb{E}\left\|\gamma \cdot \frac{1}{N}\sum_{j=1}^{N}\sum_{l=0}^{k-1}(\boldsymbol{I} - \gamma\mathbf{A}_j)^l \xi_{j,l}\right\|^2
$$
$$
\leq \gamma^2\sum_{l=0}^{k-1}\left\|(\boldsymbol{I} - \gamma\mathbf{A}_i)^l\right\|^2 \sigma^2 + \gamma^2\sum_{l=0}^{k-1}\left\|\frac{1}{N}\sum_{j=1}^{N}\sum_{l=0}^{k-1}(\boldsymbol{I} - \gamma\mathbf{A}_j)^l\right\|^2 \sigma^2
$$
$$
\overset{(a)}{\leq} 2\gamma^2 k\sigma^2, \tag{B.66}
$$

where $(a)$ is due to $\|\boldsymbol{I} - \gamma\mathbf{A}_i\|^2 \leq 1$ since we have $\gamma < \frac{1}{L}$.

With (B.50), we have
$$
\mathbb{E}\left[f(\hat{\mathbf{x}}^{t+1})\right]
$$
$$
\leq \mathbb{E}\left[f(\hat{\mathbf{x}}^t)\right] + \frac{\gamma L_h^2}{2N}\sum_{i=1}^{N}\mathbb{E}\left\|\mathbf{x}_i^t - \hat{\mathbf{x}}^t\right\|^2 - \frac{\gamma}{2}\mathbb{E}\left\|\nabla f(\hat{\mathbf{x}}^t)\right\|^2 - \left(\frac{\gamma}{2} - \frac{\gamma^2 L_g}{2}\right)\mathbb{E}\left\|\frac{1}{N}\sum_{i=1}^{N}\nabla F_i(\mathbf{x}_i^t)\right\|^2 + \frac{\gamma^2 L_g\sigma^2}{2N}
$$
$$
\leq \mathbb{E}\left[f(\hat{\mathbf{x}}^t)\right] + \frac{\gamma L_h^2}{2}\left(8\gamma^2(I-1)^2\zeta^2 + 8\gamma^4 I^4 L_h^2 \mathbb{E}\|\nabla f(\bar{\mathbf{x}}^r)\|^2 + 2\gamma^2 I\sigma^2\right)
$$
$$
- \frac{\gamma}{2}\mathbb{E}\left\|\nabla f(\hat{\mathbf{x}}^t)\right\|^2 + \frac{\gamma^2 L_g\sigma^2}{2N}
$$
$$
= \mathbb{E}\left[f(\hat{\mathbf{x}}^t)\right] + \gamma\left(4\gamma^2 L_h^2(I-1)^2\zeta^2 + \gamma^2 L_h^2 I\sigma^2\right) - \frac{1}{2}[\gamma - 4\gamma^5 L_h^4 I^4]\mathbb{E}\left\|\nabla f(\hat{\mathbf{x}}^t)\right\|^2 + \frac{\gamma^2 L_g\sigma^2}{2N}
$$
$$
\overset{(a)}{\leq} \mathbb{E}\left[f(\hat{\mathbf{x}}^t)\right] + \gamma\left(4\gamma^2 L_h^2(I-1)^2\zeta^2 + \gamma^2 L_h^2 I\sigma^2\right) - \frac{\gamma}{4}\mathbb{E}\left\|\nabla f(\hat{\mathbf{x}}^t)\right\|^2 + \frac{\gamma^2 L_g\sigma^2}{2N} \tag{B.67}
$$

where $(a)$ is due to $\gamma < \frac{1}{2IL_h}$. Rearranging the above inequality, we obtain

$$
\frac{1}{T}\sum_{t=0}^{T-1}\mathbb{E}\left\|\nabla f(\hat{\mathbf{x}}^t)\right\|^2 \leq \frac{4\mathcal{F}}{\gamma T} + \frac{2\gamma L_g\sigma^2}{2N} + 16\gamma^2 L_h^2(I-1)^2\zeta^2 + 4\gamma^2 L_h^2 I\sigma^2, \tag{B.68}
$$

where the learning rate satisfies $\gamma \leq \min\{\frac{1}{IL}, \frac{1}{L}\}$ and $T = RI$.

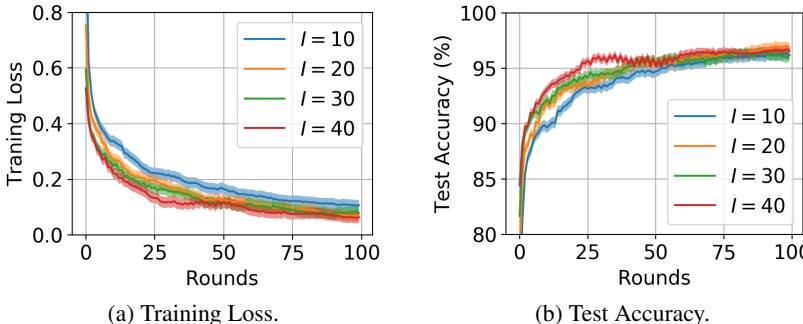

(a) Training Loss.

(b) Test Accuracy.

Figure C.1: Results with MNIST dataset. The model is a two-layer neural network with the cross-entropy loss. The percentage of heterogeneous data is $50\%$. The learning rates are chosen as $\eta = 2$ and $\gamma = 0.1$.

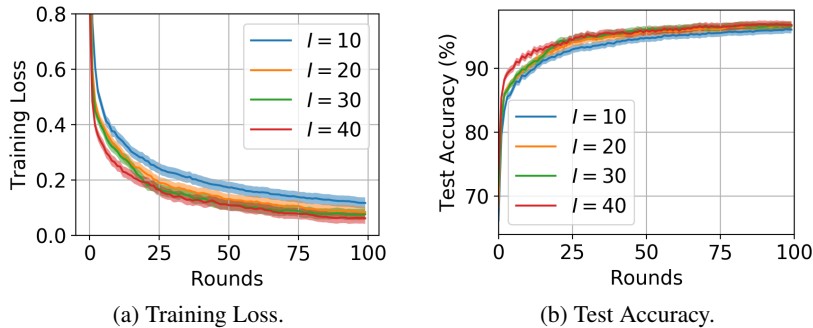

(a) Training Loss.

(b) Test Accuracy.

Figure C.2: Results with MNIST dataset. The model is a two-layer neural network with the cross-entropy loss. The percentage of heterogeneous data is $75\%$. The learning rates are chosen as $\eta = 2$ and $\gamma = 0.1$.

## C ADDITIONAL DETAILS AND RESULTS OF EXPERIMENTS

In this section, we provide additional details of our experiments. More experimental results are provided for full participation with the MNIST dataset and the CINIC-10 dataset (Darlow et al., 2018).

**Environment.** All our experiments are implemented in PyTorch and run on a server with four NVIDIA 2080Ti GPUs. The mini-batch size of SGD is 20. We run each experiment 5 times then plot their average and the stadard deviation.

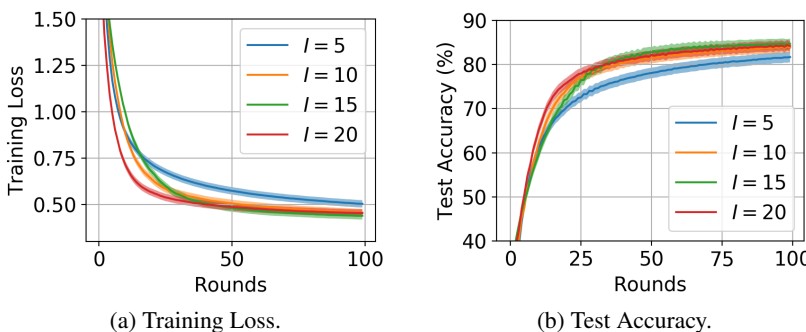

(a) Training Loss.

(b) Test Accuracy.

Figure C.3: Results with MNIST dataset. The model is linear regression with the MSE loss. The percentage of heterogeneous data is $50\%$. The learning rates are chosen as $\eta = 2$ and $\gamma = 0.01$.

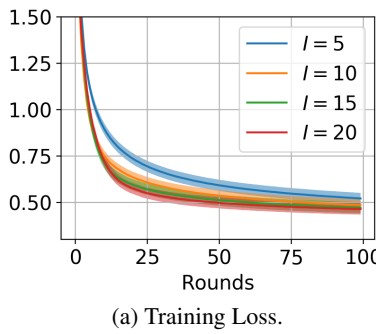 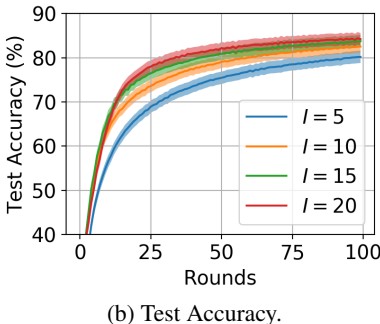

       (a) Training Loss.                  (b) Test Accuracy.

Figure C.4: Results with MNIST dataset. The model is linear regression with the MSE loss. The percentage of heterogeneous data is $75\%$. The learning rates are chosen as $\eta = 2$ and $\gamma = 0.01$.

**Model.** For experimental results with CIFAR-10 dataset in the main paper, we use a CNN model. The structure of the CNN is $5 \times 5 \times 32$ Convolutional $\to 2 \times 2$ MaxPool $\to 5 \times 5 \times 32$ Convolutional $\to 2 \times 2$ MaxPool $\to 4096 \times 512$ Dense $\to 512 \times 128$ Dense $\to 128 \times 10$ Dense $\to$ Softmax. For experimental results with MNIST dataset, we use a two-layer neural network with cross-entropy loss and a linear regression model with MSE loss. For experimental results with CINIC-10 dataset (Darlow et al., 2018), we use VGG-16 with the cross-entropy loss.

**Further explanation of the percentage of heterogeneous data.** For example, the percentage of heterogeneous data is $50\%$ means that $50\%$ of the data on each worker are with the same label, e.g., $50\%$ of the data on worker 1 are with label 1. Another $50\%$ of the data are sampled uniformly from the remaining dataset.

**The estimate of $L_h$.** Let the global model be $\bar{\mathbf{x}}$ and the local models be $\mathbf{x}_i, i = 1, 2, \ldots, N$ in the beginning of a round, then we estimate $L_h$ using the following equations.

$$L_h^2 \approx \frac{\left\| \nabla f(\bar{\mathbf{x}}) - \frac{1}{N} \sum_{i=1}^{N} \nabla f_i(\mathbf{x}_i) \right\|^2}{\frac{1}{N} \sum_{i=1}^{N} \|\mathbf{x}_i - \bar{\mathbf{x}}\|^2}.$$

Starting from a global model that is close to convergence, we perform FedAvg for 10 rounds and estimate $L_h^2$ in each round. Then we use the averaged $L_h^2$ over 10 rounds as the estimate for $L_h^2$. The reason for starting from a global model that is close to convergence is that this can make the variance of the estimate smaller. Similarly, the methods of estimating $L_g$ and $\tilde{L}$ are given by

$$L_g \approx \frac{\|\nabla f(\bar{\mathbf{x}}) - \nabla f(\bar{\mathbf{y}})\|}{\|\bar{\mathbf{x}} - \bar{\mathbf{y}}\|},$$

$$\tilde{L} \approx \max_i \frac{\|\nabla F_i(\bar{\mathbf{x}}) - \nabla F_i(\mathbf{x}_i)\|}{\|\bar{\mathbf{x}} - \mathbf{x}_i\|}.$$

The results of estimating $L_h$, $L_g$ and $\tilde{L}$ with linear regression and MSE loss can be found in Table C.1.

Table C.1: Estimated $L_h, \tilde{L}, L_g$ with the MNIST dataset.

| Obj. Function | Linear Regression | | | |
|---|---|---|---|---|
| Heterogeneity | 25% | 50% | 75% | 100% |
| $\tilde{L}$ | $2010.51 \pm 34.68$ | $3577.35 \pm 47.32$ | $20563.42 \pm 112.87$ | $25402.19 \pm 143.94$ |
| $L_h$ | $226.15 \pm 10.13$ | $916.20 \pm 23.45$ | $3172.41 \pm 57.79$ | $4610.54 \pm 66.13$ |
| $L_g$ | $869.07 \pm 18.46$ | $869.07 \pm 18.46$ | $869.07 \pm 18.46$ | $869.07 \pm 18.46$ |

**Additional Experimental Results.** We partition the MNIST dataset into 10 workers. During each round, all workers will perform the local updates. Results with a two-layer neural network and the cross-entropy loss are shown in Figure C.1 and C.2. As shown in Table 1 of the main paper, $L_h$ is

Table C.2: Training loss/test accuracy of FedAvg with MNIST dataset and MLP model when fixing the product of $R$ and $I$. The percentage of the heterogeneous data is $50\%$.

| Training Loss/Test Accuracy | $I = 10$ | $I = 20$ | $I = 30$ | $I = 40$ |
|---|---|---|---|---|
| $RI = 120$ | 0.34/88.84% | 0.34/89.38% | 0.35/89.70% | 0.35/89.45% |
| $RI = 240$ | 0.21/93.76% | 0.27/91.61% | 0.29/90.96% | 0.25/90.91% |
| $RI = 360$ | 0.19/94.01% | 0.20/93.72% | 0.23/92.57% | 0.23/92.80% |
| $RI = 480$ | 0.17/94.46% | 0.18/94.27% | 0.21/93.4% | 0.23/92.84% |
| $RI = 600$ | 0.15/95.05% | 0.15/95.47% | 0.16/94.63% | 0.18/94.22% |
| $RI = 720$ | 0.12/96.04% | 0.14/95.14% | 0.15/94.68% | 0.17/94.13% |
| $RI = 840$ | 0.12/95.90% | 0.12/95.71% | 0.14/95.06% | 0.14/95.17% |
| $RI = 960$ | 0.10/96.49% | 0.11/95.95% | 0.13/95.50% | 0.12/96.06% |

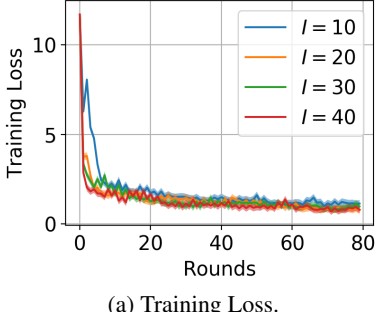

(a) Training Loss.

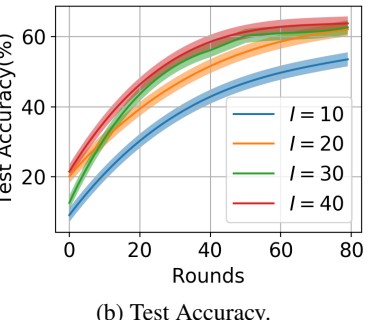

(b) Test Accuracy.

Figure C.5: Results with CINIC-10 dataset. The model is VGG-16. The percentage of heterogeneous data is $50\%$. The learning rates are chosen as $\eta = 2$ and $\gamma = 0.01$.

very small in this case. In Corollary 4.4, with full participation, it is shown that when $L_h$ is small, increasing $I$ can improve the convergence even when data are highly heterogeneous. As shown in both Figure C.1 and C.2, the curve with the largest number of local iterations, $I = 40$, converges the fastest and achieves best accuracy, which validates Corollary 4.4. Results with linear regression and the MSE loss are shown in Figure C.1 and C.2. Since $L_h$ and $L_g$ are larger compared to that of the two-layer neural network, a smaller $\gamma$ and smaller $I$'s are chosen according to Corollary 4.4. It can be seen in both Figure C.3 and C.4, the curve with the largest number of local iterations, $I = 20$ converges the fastest and achieves the best accuracy.

In order to verify the insight of fixing $RI$, we ran more experiment as shown in Table C.2. In Table C.2, we provide the training loss and the test accuracy for the different choices of $I$ by fixing $RI$. It can be seen that by varying $I$, the training loss and testing accuracy do stay almost the same if $RI$ is given. This verifies our finding and cannot be explained by the existing theory in the literature. In addition, if we take a closer look, it can be seen that the difference between $I = 10$ and $I = 20$ is negligible. That is, the difference on the training loss is at most $0.06$ and the difference on the test accuracy is at most $1.16\%$. However, another implication of our results is that the number of communication rounds are doubled for $I = 10$ to achieve the almost same testing accuracy. Especially for $RI = 960$, the number of communication rounds is $R = 96$ for $I = 10$ while the number of communication rounds is $R = 48$ for $I = 10$. Moreover, the difference on the training loss between $I = 10$ and $I = 40$ is at most $0.06$ and the difference on the test accuracy is at most $2.85\%$. However, the number of communication rounds of $I = 10$ is four times of that for $I = 40$. We believe that the experimental results can support our theoretical results very well that we can keep the product of $R$ and $I$ while increasing $I$ and decreasing $R$ simultaneously to achieve almost the same convergence rate.

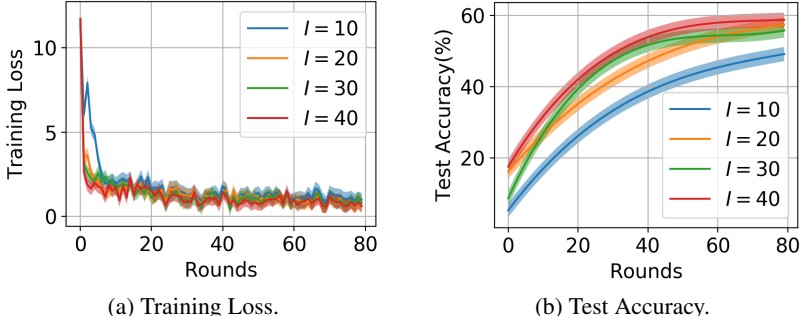

(a) Training Loss.

(b) Test Accuracy.

Figure C.6: Results with CINIC-10 dataset. The model is VGG-16. The percentage of heterogeneous data is $75\%$. The learning rates are chosen as $\eta = 2$ and $\gamma = 0.01$.

