# OpenReview forum: "A New Theoretical Perspective on Data Heterogeneity in Federated Averaging"
_ICLR.cc/2024/Conference — Submitted to ICLR 2024_

### Official Review · Reviewer_wiji · 2023-10-22

**Soundness:** 2 fair
**Presentation:** 3 good
**Contribution:** 2 fair
**Rating:** 5
**Confidence:** 4

**Summary:**

This paper addresses the challenge of data heterogeneity in federated learning, where theoretical analyses have previously been pessimistic about the convergence error due to local updates. Empirical studies, however, suggest that increasing the number of local updates can enhance convergence rates and reduce communication costs when dealing with heterogeneous data. To reconcile these disparities, the paper introduces a new theoretical perspective on data heterogeneity in federated averaging (FedAvg) with non-convex objective functions. They propose a novel assumption called the "heterogeneity-driven Lipschitz assumption" to better account for data heterogeneity's impact on local updates. They replace the commonly used local Lipschitz constant with the heterogeneity-driven Lipschitz constant and the global Lipschitz constant in their convergence analysis. This change results in better convergence bounds for both full and partial participation, compared to previous FedAvg analyses. These findings suggest that increasing local updates can improve convergence rates even with highly heterogeneous data. Furthermore, the paper identifies a region where FedAvg, also known as local SGD, can outperform mini-batch SGD, even when the gradient divergence is substantial.

**Strengths:**

1. **Novel Perspective on Data Heterogeneity**: The paper introduces a fresh perspective on the impact of data heterogeneity on the federated averaging (FedAvg) algorithm by introducing the concept of the "heterogeneity-driven Lipschitz constant." This innovative viewpoint sheds light on how data diversity affects the convergence behavior of the algorithm.

2. **Weaker Assumptions**: The paper's assumptions are demonstrated to be weaker than those in existing literature.

3. **Convergence Analysis for Non-Convex Objective Functions**: The paper extends the analysis of FedAvg to encompass general non-convex objective functions.

4. **Partial Participation Inclusion**: The paper incorporates partial participation scenarios, where only a subset of workers are involved in each round of local updates.

5. **Identification of Competitive Region**: The paper identifies a specific region in which local stochastic gradient descent (SGD) can outperform the widely used mini-batch SGD.

6. **Experimental Validation**: The paper backs its theoretical results with experimental validation. This empirical validation shows the corresponding Lipschitz constants.

**Weaknesses:**

I have several concerns about the paper:

1. **Lack of Clarity in Technical Challenges and Limited Technical Contributions**: The authors do not sufficiently show the technical challenges associated with the convergence analysis based on the new assumption. The paper could benefit from a more detailed statement of challenges from the new Lipschitz assumption, especially in comparison to the conventional local Lipschitz assumption (assumption 1). This lack of clarity leaves room for misinterpretation and raises questions about the significance of the proposed approach. As I understand, the main difference in the proof seems to revolve around using this new assumption to bound the discrepancy between the aggregated gradients and the groundtruth gradients (eq 11). In contrast, other analyses have traditionally employed the local Lipschitz assumption (assumption 1). A clearer statement of how this change significantly enhances the convergence analysis would be beneficial.

2. **Experiment Results Below Expectation**: The experimental results presented in the paper fall short of expectations, particularly in comparison to prior works. In many previous studies, training and test accuracy consistently reach high levels, such as 90% for MNIST and 80% for CIFAR10 datasets. The paper's results, on the other hand, demonstrate lower performance. This discrepancy might be attributed to the specific models employed, namely linear regression and simple CNN. The choice of these relatively basic models may limit the generalizability of the findings, and more comprehensive experiments with diverse model architectures could provide a more robust assessment of the proposed approach. In addition, I believe the estimation of Lipschitz is a highlight for experiments. So I hope the author can provide a more in-depth discussion on how Lipschitz constants are estimated, the justification, and present results across a broader range of datasets and models to reinforce the paper's findings.

**Questions:**

see above

---

> ### Author Response · Authors · 2023-11-20
> **Responses to Reviewer wiji**
>
> We thank Reviewer wiji for the questions and the suggestions. Our responses are as follows.
>
> 1. **About the technical novelty.** We agree with Reviewer wiji that the main technical novelty in the proof for Theorem 4.3 and 4.5 is about developing new techniques to incorporate Assumption 4.1 and 4.2. In addition, we also develop Proposition 5.1-5.3 and Theorem 5.5 to demonstrate the usefulness and the effectiveness of Assumption 4.1 and 4.2. We summarize our technical novelty as follows.
>
> (1) We need to develop new techniques to incorporate Assumption 4.1 and 4.2. In the proof of Theorem 4.3 shown in the Appendix, we need to characterize the difference between local gradients. In the literature, this is done by applying the local Lipschitz constant in shown in Assumption 3.1 in the main paper. In our paper, since Assumption 3.1 is replaced by our newly introduced assumption 4.2, the proof techniques in the literature cannot be applied. It requires developing new proof techniques to use Assumption 4.2 as shown in the proof of Lemma A.1-A.3. For example, in Lemma A.1, due to the application of Assumption 4.1 and 4.2, we have to deal with a new term, the local gradient deviation $\\|\frac{1}{N}\sum_{i=1}^N \nabla F_i(\mathbf{x}_i) - \nabla F_j(\mathbf{x}_j) \\|^2$, which is not shown in other techniques. Another example is in the proof of Theorem 4.5. Due to that we only use the global Lipschitz gradient assumption, we have to derive a new method to bound and incorporate the sampling related term
>
> $$\mathbb{E}\\|\frac{1}{M}\sum_{i\in S_r}\sum_{k=0}^I \mathbf{g}_i(\mathbf{x}_r^{r,k}) \\|^2,$$
>
> which can be seen from (A.27) to (A.35).
>
> (2) In addition to using Assumption 4.2 to characterize the new convergence rate of FedAvg, we also validate this assumption from the theoretical perspective. We develop the proof for Proposition 5.1-5.3 in Appendix.
>
> (3) In Theorem 5.5, since we use iteration-by-iteration analysis in the proof, the learning rate is not a function of $I$. It can be seen that in the literature, such as Theorem IV in [1], for quadratic objective functions, the learning rate is upper bounded by $I$. The advantage of that $\gamma$ is not a function of $I$ can be explained as follows. In Theorem 5.5, to obtain the optimal dependence on $R$ and $I$, we can choose $\gamma = \frac{1}{\sqrt{RI}}$. This requires that $\frac{1}{\sqrt{RI}}\le \frac{1}{L_g}$, which means that $I$ can be as large as possible. However, if $\gamma \le \frac{1}{IL_g}$, we will have $\frac{1}{\sqrt{RI}}\le\frac{1}{IL_g} $ such that $I\le \frac{R}{L_g^2}$, which means that to achieve the convergence rate of $\mathcal{O}(\frac{1}{\sqrt{RI}})$, $I$ cannot be arbitrarily large. Therefore, the range of the learning rate in Theorem 5.5 significantly enhances the convergence analysis.
>
> 2. **About the experiments.** We would like to clarify the following points for our experiments.
>
> (1) About the accuracy. It is common in the literature on the theoretical analysis of FL algorithms [1-4] that accuracy cannot reach $80\\%$ for CIFAR-10 with a highly non-IID partition. This is because that in the literature of theoretical analysis of FL algorithms, the goal of the experiments is mainly to show the effect of some parameters, such as $I$, while other parameters are kept the same instead of being fine-tuned. For the MNIST dataset, the test accuracy can achieve $90\%$ when the percentage of heterogeneous data is $50\\%$. The experimental results for MNIST dataset can be found in Appendix C.
>
> (2) About more experiments with different models and datasets. During these days, we run more experiments with the CINIC-10 dataset and VGG-16. We have added the new experimental results in Appendix C, which can be found in the new PDF file.
>
> (3) About estimating $\tilde{L}$, $L_g$ and $L_h$. We provide the methods of estimating $L_h$, $L_g$ and $\tilde{L}$ in Appendix C.
>
> We will provide more experimental results for (2) and (3) in the subsequent version if the paper is accepted.
>
> [1] Wang, Jianyu, et al. "Tackling the objective inconsistency problem in heterogeneous federated optimization." Advances in neural information processing systems 33 (2020): 7611-7623.
>
> [2] Jhunjhunwala, D., Sharma, P., Nagarkatti, A., \& Joshi, G. (2022, August). Fedvarp: Tackling the variance due to partial client participation in federated learning. In Uncertainty in Artificial Intelligence (pp. 906-916). PMLR.
>
> [3] Khaled, Ahmed, Konstantin Mishchenko, and Peter Richtárik. "Tighter theory for local SGD on identical and heterogeneous data." International Conference on Artificial Intelligence and Statistics. PMLR, 2020.
>
> [4] Reddi, Sashank J., et al. "Adaptive Federated Optimization." International Conference on Learning Representations. 2020.

---

### Official Review · Reviewer_RuXa · 2023-10-23

**Soundness:** 3 good
**Presentation:** 3 good
**Contribution:** 2 fair
**Rating:** 5
**Confidence:** 3

**Summary:**

This paper seeks to explain the benefits of local updates in FedAvg. To that end, it proposes a new but weaker assumption to replace the typically used per-client smoothness assumption. Specifically, it proposes a heterogeneity-driven Lipschitz condition on the averaged gradients (Assumption 4.2). Under this assumption, it derives convergence results for non-convex functions under full-device and partial-device participation. The authors claim that when $L_h$ (i.e., the parameter of Assumption 4.2) is small, the negative effect of multiple local updates is small. Additionally, for quadratics, when $L_h = 0$ and the stochastic gradient variance is small, the authors show that local SGD is better than mini-batch SGD (with the same number of gradient evaluations). There are some experiments to corroborate the theoretical insights.

**Strengths:**

**1.** This paper attempts to explain the benefit of local updates in FedAvg. This is an aspect where there is a gap between theory and practice.

**2.** The heterogeneity-driven Lipschitz assumption (Assumption 4.2) is weaker than per-client smoothness and has a dependence on the heterogeneity of the system. Moreover, the authors empirically show that $L_h$ (the parameter of Assumption 4.2) is small at least on the datasets that they tried.

**3.** For quadratics, in the *very special case* of $L_h = 0$ and when the stochastic gradient variance is small, it is shown that local SGD is better than mini-batch SGD (with the same number of gradient evaluations).

**Weaknesses:**

**1.** After Theorem 4.3 (in "New insights about the effect of data heterogeneity"), the authors write "*A key message is that when $\zeta^2$ is large, as long as $L_h^2$ is small enough, the error caused by local updates can still be small.*" First of all, "small enough" should be precisely quantified. But more importantly, the choice of $\gamma$ in Corollary 4.4 completely nullifies the role of $L_h^2$ in the convergence bound. The bound in Corollary 4.6 is also independent of $L_h^2$. This looks a bit strange (especially after reading the aforementioned statement).

**2.** A result similar to Theorem 4.5 (and Theorem 4.4) is presented in Theorem 1 of [1] (cited below). They use the per-client smoothness assumption (Assumption 3.1 in this paper) though. The authors claim that they "develop new techniques" to obtain Theorem 4.5 but Appendix B does not mention what are the new techniques. Can the authors please elaborate on this? It seems that this paper obtains prior results with better constants but it is not clear to me how significant the technical challenges are in doing so.

**3.** The class of quadratics in Theorem 5.5 (i.e., $A_i = A$ for all $i$) is too restrictive for me. It would have been better to obtain a general result with $A_i \neq A$ and then state how small $L_h = 2 \max_{i} |\lambda(A_i - A)|$ needs to be so that local SGD is better than mini-batch SGD.

Overall, I don't find any of the results or insights of this paper interesting or intriguing enough to endorse it for an ICLR publication at the moment.

[1]: Jhunjhunwala, D., Sharma, P., Nagarkatti, A. and Joshi, G., 2022, August. Fedvarp: Tackling the variance due to partial client participation in federated learning. In Uncertainty in Artificial Intelligence (pp. 906-916). PMLR.

**Questions:**

**1.** Regarding Weakness 1, perhaps the authors should choose a $\gamma$ that is *not* inversely proportional to $L_h$?

**2.** Please answer the question in Weakness 2.

---

> ### Author Response · Authors · 2023-11-20
> **Responses to Reviewer RuXa (Part 1)**
>
> We thank Reviewer RuXa for the questions and the suggestions. Our responses are as follows.
>
> 1. **About the choice of $\gamma$.** First, we quantify $L_h$ in Theorem 4.3 as follows. When $L_h$ is small enough such that $\gamma^2L_h^2\zeta^2\ll \frac{\mathcal{F}}{\gamma \eta R}$, the convergence bound will become smaller when increasing $I$. However, it should be noted that this does not imply that $I$ can be arbitrarily large. As we will show in the following, the upper bound of $I$ depends on $L_h$.
>
> Second, in Corollary 4.4, choosing $\gamma = \frac{1}{IL_h}$ does not mean that the role of $L_h$ is nullified. Instead, a smaller $L_h$ implies that we can choose a larger $I$ in Corollary 4.4. The explanation is as follows.
> In Theorem 4.3, the condition for $\gamma$ is $\gamma \le \frac{1}{30(L_g+L_h)I}$. By this condition, when $L_h \le \frac{1}{I}$, we have
> \begin{align}
>     \gamma = \frac{1}{L_h I}\cdot \frac{1}{\sqrt{RIN}} \le \frac{1}{\sqrt{RIN}} \le \frac{1}{30(L_g+L_h)I}.
> \end{align}
> Rearranging the above inequality, we have $I \le \frac{RN}{900(L_g+L_h)^2}$. Combined with $L_h\le \frac{1}{I}$, we obtain the upper bound of $I$. That is
>
> $$
> I \le \min \left\\{\frac{1}{L_h}, \frac{RN}{900(L_g+L_h)^2}\right\\}.
> $$
>
> It can be seen that a smaller $L_h$ means we can choose a larger $I$ such that Corollary 4.4 still holds.
>
> 2. **About the technical novelty.** It is not straightforward to apply Assumption 4.1 and 4.2 to the convergence analysis for both full participation and partial participation. The technical novelties are summarized as follows.
>
> (1) We need to develop new techniques to incorporate Assumption 4.1 and 4.2. In the proof of Theorem 4.3 shown in the Appendix, we need to characterize the difference between local gradients. In the literature, this is done by applying the local Lipschitz constant in shown in Assumption 3.1 in the main paper. In our paper, since Assumption 3.1 is replaced by our newly introduced assumption 4.2, the proof techniques in the literature cannot be applied. It requires developing new proof techniques to use Assumption 4.2 as shown in the proof of Lemma A.1-A.3. For example, in Lemma A.1, due to the application of Assumption 4.1 and 4.2, we have to deal with a new term, the local gradient deviation $\\|\frac{1}{N}\sum_{i=1}^N \nabla F_i(\mathbf{x}_i) - \nabla F_j(\mathbf{x}_j) \\|^2$, which is not shown in other techniques. Another example is in the proof of Theorem 4.5. Due to that we only use the global Lipschitz gradient assumption, we have to derive a new method to bound and incorporate the sampling related term
>
> $\mathbb{E}\\|\frac{1}{M}\sum_{i\in S_r}\sum_{k=0}^I \mathbf{g}_i(\mathbf{x}_r^{r,k}) \\|^2,$
>
> which can be seen from (A.27) to (A.35).
>
> (2) In addition to using Assumption 4.2 to characterize the new convergence rate of FedAvg, we also validate this assumption from the theoretical perspective. We develop the proof for Proposition 5.1-5.3 in Appendix.
>
> (3) In Theorem 5.5, since we use iteration-by-iteration analysis in the proof, the learning rate is not a function of $I$. It can be seen that in the literature, such as Theorem IV in [1], for quadratic objective functions, the learning rate is upper bounded by $I$. The advantage of that $\gamma$ is not a function of $I$ can be explained as follows. In Theorem 5.5, to obtain the optimal dependence on $R$ and $I$, we can choose $\gamma = \frac{1}{\sqrt{RI}}$. This requires that $\frac{1}{\sqrt{RI}}\le \frac{1}{L_g}$, which means that $I$ can be as large as possible. However, if $\gamma \le \frac{1}{IL_g}$, we will have $\frac{1}{\sqrt{RI}}\le\frac{1}{IL_g} $ such that $I\le \frac{R}{L_g^2}$, which means that to achieve the convergence rate of $\mathcal{O}(\frac{1}{\sqrt{RI}})$, $I$ cannot be arbitrarily large. Therefore, the range of the learning rate in Theorem 5.5 significantly enhances the convergence analysis.

---

> ### Author Response · Authors · 2023-11-20
> **Responses to Reviewer RuXa (Part 2)**
>
> 3. **About the quadratic results.** We would like to clarify the following points for our quadratic results in Theorem 5.5.
>
> (1) Developing Theorem 5.5 is not straightforward, and it provides a better range for $I$. As mentioned in our response to the question about the technical novelty, we use the techniques which are different from those used in the proof for Theorem 4.3. In Theorem 5.5, the range of the learning rate is given by $\gamma\le \frac{1}{|\lambda(\mathbf{A})|}$, which is not a function of $I$. This is not shown in previous works.
> The advantage of the range of the learning rate $\gamma\le \frac{1}{|\lambda(\mathbf{A})|}$ can be shown as follows. To obtain the optimal dependence on $R$ and $I$, we can choose $\gamma = \frac{1}{\sqrt{RI}}$. By the range of the learning rate, this only requires $\frac{1}{\sqrt{RI}}\le \frac{1}{|\lambda(\mathbf{A})|}$, which means we can choose arbitrarily large $I$. Extending $\gamma\le \frac{1}{|\lambda(\mathbf{A})|}$ to non-convex quadratic objectives with $\mathbf{A} \neq \mathbf{A}_i$ can be very challenging since current techniques require $\gamma < \frac{1}{IL_g}$ as shown in Theorem 4.3. This requires us developing new techniques, which could be one of our future work.
>
> (2) The theoretical result in Theorem 4.3 is general and we can use Theorem 4.3 to make a comparison between local SGD and mini-batch SGD when $L_h>0$. Later we can see that the implication and the intuition are similar to what we have observed in Theorem 5.5.
>
> The convergence upper bound for local SGD is
> $$
>     \min_r \mathbb{E}\left\\| \nabla f(\bar{\mathbf{x}}^r) \right\\| = \mathcal{O}\left(\frac{\mathcal{F}}{\gamma\eta RI} + \frac{\gamma\eta L_g \sigma^2}{N}+\gamma^2L_h^2I^2\zeta^2 + \gamma^2 L_h^2 I\sigma^2 \right).
> $$
> The convergence upper bound for mini-batch SGD is
> $$
>     \min_r \mathbb{E}\left\\| \nabla f(\bar{\mathbf{x}}^r) \right\\| = \mathcal{O}\left(\frac{\mathcal{F}}{\alpha R}+ \frac{\alpha L_g \sigma^2}{NI} \right).
> $$
> Here, $\alpha$ denotes the learning rate of mini-batch SGD, distinguishing it from local SGD. Let $\gamma\eta = \alpha$. If $L_h$ is small enough, such that $\gamma^2L_h^2(I^2\zeta^2 + I\sigma^2)\le \frac{\gamma\eta L_g\sigma^2}{N}$, when $\sigma \to 0$, the convergence rate of local SGD goes to $\mathcal{O}(\frac{1}{RI})$ while the convergence rate of mini-batch SGD goes to $\mathcal{O}(\frac{1}{R})$.  Later we can see that the implication and the intuition are similar to what we have observed in Theorem 5.5.
>
> The convergence upper bound for local SGD is
> $$
>     \min_r \mathbb{E}\left\\| \nabla f(\bar{\mathbf{x}}^r) \right\\| = \mathcal{O}\left(\frac{\mathcal{F}}{\gamma\eta RI} + \frac{\gamma\eta L_g \sigma^2}{N}+\gamma^2L_h^2I^2\zeta^2 + \gamma^2 L_h^2 I\sigma^2 \right).
> $$
> The convergence upper bound for mini-batch SGD is
> $$
>     \min_r \mathbb{E}\left\\| \nabla f(\bar{\mathbf{x}}^r) \right\\| = \mathcal{O}\left(\frac{\mathcal{F}}{\alpha R}+ \frac{\alpha L_g \sigma^2}{NI} \right),
> $$
> where we use $\alpha$ to denote the learning rate of mini-batch SGD to distinguish from local SGD. Let $\gamma\eta = \alpha$. If $L_h$ is small enough such that $\gamma^2L_h^2(I^2\zeta^2 + I\sigma^2)\le \frac{\gamma\eta L_g\sigma^2}{N}$, when $\sigma \to 0$, the convergence rate of local SGD goes to $\mathcal{O}(\frac{1}{RI})$ while the convergence rate of mini-batch SGD goes to $\mathcal{O}(\frac{1}{R})$.
>
>
> [1] Karimireddy, Sai Praneeth, et al. "Scaffold: Stochastic controlled averaging for federated learning." International conference on machine learning. PMLR, 2020.

---

> > ### Comment · Reviewer_RuXa · 2023-11-22
> > **Reply to both parts**
> >
> > Thanks for the detailed rebuttal!
> >
> > Regarding 1, I believe there is an error. If $L_h \leq \frac{1}{I}$, then $\frac{1}{L_h I} \geq 1$. Thus, the inequality $\gamma = \frac{1}{L_h I}.\frac{1}{\sqrt{R I N}} \leq \frac{1}{\sqrt{R I N}}$ does *not* hold. Regarding 3, I maintain my stance that $A = A_i$ is too restrictive and not very interesting. Further, if the authors claim that the result in Theorem 4.3 can also show when local SGD is better than mini-batch SGD, I'd like them to quantify this *precisely in the form of a corollary/theorem* by providing choices for $\gamma$, $\eta$, etc. and laying out the condition for $L_h$ in terms of these quantities.
> >
> > I'll keep my score for now.

---

### Official Review · Reviewer_ziE8 · 2023-10-29

**Soundness:** 3 good
**Presentation:** 2 fair
**Contribution:** 2 fair
**Rating:** 6
**Confidence:** 3

**Summary:**

This paper proposes to alternatively measure the heterogeneity in FedAvg by the Lipschitz constant of the averaged gradients instead of the local gradient. The proposed alternative Lipschitz constants are much smaller, thus yielding an improved convergence rate for FedAvg in non-convex optimization. Such improvement is further demonstrated with the example of quadratic functions, the comparison against mini-batch SGD under the same setting, and numerical evaluation.

**Strengths:**

1. It's a good insight, though natural as well, to measure the closeness between the average of local models and the centralized model with the proposed Lipschitzness of the averaged gradients. The proposed measure is overall a valid complement to the other heterogeneity assumptions in the literature.

2. Convergence rates for both full and partial participation are derived.

3. The numerical yield of the proposed Lipschitz constants seems significant.

**Weaknesses:**

1. The convergence analysis seems less technically novel as the result is almost identical to previous derivations, replacing the local Lipschitz constant with the proposed ones.

2. The insight that the average of local models may be close to the centralized model even with the presence of heterogeneity is also observed in Wang et al., 2022, i.e., to instead measure the average drift at optimum (Eq. (15) in Wang et al., 2022). The corresponding discussion seems limited to that Wang et al., 2022 studies convex functions whereas this paper studies non-convex functions, without much comments on how they differ / relate in measuring heterogeneity. As a result, I'm not quite convinced by the statement "both works cannot capture the information on the data heterogeneity contained in the local Lipschitz constant as shown in our paper so their convergence upper bounds can be worse compared to ours."

3. The analysis in the paper seems to me more numerical than theoretical, as the improvement in convergence rates comes purely from improved constants, which are mostly measured numerically. The theoretical arguments that the proposed constants are smaller are also limited to quadratics.

**Questions:**

An improved version of Assumption 3.3 would be to assume bounded gradient divergence at optima only, as in Glasgow et al., 2022, or as Assumption 2.3 in Patel, Kumar Kshitij, et al., 2023. How would this assumption affect the comparison / relation of the proposed measure to previous measures?

Glasgow, Margalit R., Honglin Yuan, and Tengyu Ma. "Sharp bounds for federated averaging (local SGD) and continuous perspective." International Conference on Artificial Intelligence and Statistics. PMLR, 2022.

Patel, Kumar Kshitij, et al. "On the Still Unreasonable Effectiveness of Federated Averaging for Heterogeneous Distributed Learning." Federated Learning and Analytics in Practice: Algorithms, Systems, Applications, and Opportunities. 2023.

---

> ### Author Response · Authors · 2023-11-20
> **Responses to Reviewer ziE8 (Part 1)**
>
> We thank Reviewer ziE8 for the suggestions and the questions. Our responses are as follows.
>
> 1. **About the technical novelty.** The technical novelties are summarized as follows.
>
> (1) We need to develop new techniques to incorporate Assumption 4.1 and 4.2. In the proof of Theorem 4.3 shown in the Appendix, we need to characterize the difference between local gradients. In the literature, this is done by applying the local Lipschitz constant in shown in Assumption 3.1 in the main paper. In our paper, since Assumption 3.1 is replaced by our newly introduced assumption 4.2, the proof techniques in the literature cannot be applied. It requires developing new proof techniques to use Assumption 4.2 as shown in the proof of Lemma A.1-A.3. For example, in Lemma A.1, due to the application of Assumption 4.1 and 4.2, we have to deal with a new term, the local gradient deviation $\\|\frac{1}{N}\sum_{i=1}^N \nabla F_i(\mathbf{x}_i) - \nabla F_j(\mathbf{x}_j) \\|^2$, which is not shown in other techniques. Another example is in the proof of Theorem 4.5. Due to that we only use the global Lipschitz gradient assumption, we have to derive a new method to bound and incorporate the sampling related term
>
> $$\mathbb{E}\\|\frac{1}{M}\sum_{i\in S_r}\sum_{k=0}^I \mathbf{g}_i(\mathbf{x}_r^{r,k}) \\|^2,$$
>
> which can be seen from (A.27) to (A.35).
>
> (2) In addition to using Assumption 4.2 to characterize the new convergence rate of FedAvg, we also validate this assumption from the theoretical perspective. We develop the proof for Proposition 5.1-5.3 in Appendix.
>
> (3) In Theorem 5.5, since we use iteration-by-iteration analysis in the proof, the learning rate is not a function of $I$. It can be seen that in the literature, such as Theorem IV in [1], for quadratic objective functions, the learning rate is upper bounded by $I$. The advantage of that $\gamma$ is not a function of $I$ can be explained as follows. In Theorem 5.5, to obtain the optimal dependence on $R$ and $I$, we can choose $\gamma = \frac{1}{\sqrt{RI}}$. This requires that $\frac{1}{\sqrt{RI}}\le \frac{1}{L_g}$, which means that $I$ can be as large as possible. However, if $\gamma \le \frac{1}{IL_g}$, we will have $\frac{1}{\sqrt{RI}}\le\frac{1}{IL_g} $ such that $I\le \frac{R}{L_g^2}$, which means that to achieve the convergence rate of $\mathcal{O}(\frac{1}{\sqrt{RI}})$, $I$ cannot be arbitrarily large. Therefore, the range of the learning rate in Theorem 5.5 significantly enhances the convergence analysis.

---

> ### Author Response · Authors · 2023-11-20
> **Responses to Reviewer ziE8 (Part 2)**
>
> 2. **About the difference between Wang et al., 2022 and our paper.** In Wang et al., 2022, a new metric for data heterogeneity, $\rho$, the average drift at optimum, is proposed. The definition of $\rho$ is
> \begin{align}
> \left\\|\frac{1}{\gamma I}\left(\frac{1}{N}\sum_{i=1}^N\mathbf{x}_i^{r,I} - \bar{\mathbf{x}}^r \right)\right\\|= \rho.
> \end{align}
> The detailed comparison between Wang et al., 2022 and our paper is in the following.
>
>
> (1) The new metric $\rho$ in Wang et al., 2022 focuses on the difference on models while in our paper, we still focus on the difference on the gradients. The key insight in Wang et al., 2022 is that since $\rho$ is small, when the global model is $\mathbf{x}^*$, after multiple local updates, the averaged model does not change significantly.
> In our paper, the key insight is that since $L_h$ can be small, the difference between the current global gradient and the current averaged local gradients can be small.
> In Wang et al., 2022, the gradient divergence is not used in the analysis.
> In our analysis, we still use the gradient divergence jointly with the proposed $L_h$ to measure the data heterogeneity.
>
> (2) In Wang et al., 2022, it is only empirically shown that $\rho$ can be small. In our paper, we not only empirically show that $L_h$ can be small, but also provide an analytical example. Our quadratic example can be non-convex, a case which $\rho$ cannot cover.
>
>
> (3) One weakness of using $\rho$ is that in the convergence bound in Wang et al., 2022, the convergence error shown by $\rho$ cannot vanish. This means that by choosing $\gamma = \frac{1}{\sqrt{R}}$, when $R$ goes to infinity, the convergence bound cannot guarantee that FedAvg can converge to the local minima of the global objective function. On the contrary, our convergence bound can guarantee the convergence to the local minima of the global objective function, which is shown by Corollary 4.4.
>
>
> 3. **About the proposed $L_h$.** Using $L_h$ can address the unsolved problem in the literature. We observe that existing theoretical analyses are pessimistic about the convergence error caused by local updates; however, in practice, local updates can improve the convergence rate and reduce communication costs. By employing $L_h$, we can bridge the gap between the pessimistic theoretical results and the positive experimental outcomes for federated algorithms, which is \textbf{unsolved} in the literature. This is our main contribution.
>
> Other theoretical contributions include: (i) as shown in Proposition 5.1, we rigorously prove our assumptions are weaker; (ii) there are technical novelties in our proof (see the response point 1 above.)
> (iii) in Theorem 5.5, by extending the discussion to $L_h=0$, we show that local SGD can outperform mini-batch SGD for non-convex quadratic objectives, which is not shown in the literature.
>
> 4. **About the question for the bounded gradient divergence on the optima only.** The bounded gradient divergence on the optima is
>
> $$
>  \\|\nabla f(\mathbf{x}^*) - \nabla F_i(\mathbf{x}^*) \\|^2 = \\| \nabla F_i(\mathbf{x}^*)\\|^2 \le \zeta_*^2.
> $$
>
> We observe that $\zeta_*$ is often used to substitute $\zeta$ in the analysis for convex objective functions in the literature [2-4]. To the best of our knowledge, there is no literature applying $\zeta_*$ in the analysis for non-convex objective functions. The possible reason could be that the convergence upper bound for non-convex objective functions can only guarantee the convergence to one of the local minima. Only assuming gradient divergence on the global minima might not help the convergence analysis. However, we believe that it is possible to use both $L_h$ and $\zeta_*$ in the convergence analysis for convex objectives.  For example, in the proof for heterogeneous data of [2], there are also steps transforming the gradient difference to the model divergence, such as (58) and (59) in [2], where we can use $L_h$ to substitute $\tilde{L}$. Therefore, applying Assumption 4.1 and 4.2 to the convergence analysis for convex objectives can be one of our future work.
>
>
>
> [1] Karimireddy, Sai Praneeth, et al. "Scaffold: Stochastic controlled averaging for federated learning." International conference on machine learning. PMLR, 2020.
>
> [2] Khaled, Ahmed, Konstantin Mishchenko, and Peter Richtárik. "Tighter theory for local SGD on identical and heterogeneous data." International Conference on Artificial Intelligence and Statistics. PMLR, 2020.
>
> [3] Glasgow, Margalit R., Honglin Yuan, and Tengyu Ma. "Sharp bounds for federated averaging (local SGD) and continuous perspective." International Conference on Artificial Intelligence and Statistics. PMLR, 2022.
>
> [4] Patel, Kumar Kshitij, et al. "On the Still Unreasonable Effectiveness of Federated Averaging for Heterogeneous Distributed Learning." Federated Learning and Analytics in Practice: Algorithms, Systems, Applications, and Opportunities. 2023.

---

> > ### Comment · Reviewer_ziE8 · 2023-11-22
> >
> > Thanks for the response. I think the technical challenges and the difference to Wang et al., 2022 have been better addressed. As long as the authors include relevant discussion in the revised version, I would keep my score for their exploration of alternative notions for data heterogeneity.

---

> ### Author Response · Authors · 2023-11-22
>
> Dear Reviewer ziE8,
>
> We would like to thank you for suggestions and your appreciation for our exploration of alternative notations for data heterogeneity. We have uploaded the revised version of our paper, which includes all the discussions and additional experimental results.
>
> Best regards,
> Authors of Paper 4651

---

### Official Review · Reviewer_NMhc · 2023-10-31

**Soundness:** 2 fair
**Presentation:** 3 good
**Contribution:** 2 fair
**Rating:** 3
**Confidence:** 3

**Summary:**

This paper suggest to replace "Local Lipschitz Gradient" with two other assumptions : "Global Lipschitz Gradient" and "Heterogeneity-driven Lipschitz Condition on Averaged Gradients"  to better capture data heterogeneity of clients. The authors show that these assumptions are weaker than the classic assumption, and the constants are smaller. They prove convergence bounds for FedAvg based on the new assumptions.

**Strengths:**

1) The new assumptions are weaker than previous one. The proof of many related works can adopt these assumptions instead of the classic Local Lipschitz Gradient without much change.

2)  The authors build an example that "Heterogeneity-driven Lipschitz constant" ($L_h$) is zero, but "Bounded Gradient Divergence constant"  ($\zeta$)  or local Lipschitz constant ($\tilde{L}$) are large. They show that in practice $L_h$ is much smaller $\tilde{L}$.

**Weaknesses:**

1) The contribution of the paper is limited. The bound is the same as [1]. The only difference is that the Lipschitz constants are replaced, but the bound has the same order and the same limitations as [1] with respect to $R$, $I$, and $N$.

2) The paper claims several times that previous bounds don't improve with more local steps, and sometimes get worse. However, this is not true. For example in [1] or [2] the bounds decrease with the number of local steps. Again, the bound in this paper has the same order with respect to $I$ as [1]. Since, the new assumptions are weaker than the classic one, and in practice $L_h$ is not zero, these new assumptions can't help to get a better bound with respect to $I$ other than improving constants.

3) The paper claims based on the bounds, FedAvg works with large $I$ and small $R$. However, this doesn't happen in practice. Also, in order for the bound to hold, we should have $\eta \gamma \le \frac{1}{4I L_g}$, and in the Corollary 4.4 the bounds $\eta \gamma = \sqrt{\frac{4FN}{RI L_g \sigma^2}}$, which means $O(IN) \le R$. Therefore, with the increase of number of local steps, more rounds is needed for the bound to hold.

[1] Haibo Yang, Minghong Fang, and Jia Liu. Achieving linear speedup with partial worker
participation in non-iid federated learning. In International Conference on Learning
Representations, 2020.

[2] Sai Praneeth Karimireddy, Satyen Kale, Mehryar Mohri, Sashank Reddi, Sebastian Stich, and
Ananda Theertha Suresh. Scaffold: Stochastic controlled averaging for federated learning. In
International Conference on Machine Learning, pp. 5132–5143. PMLR, 2020

**Questions:**

Please discuss the weaknesses.

---

> ### Author Response · Authors · 2023-11-20
> **Responses to Reviewer NMhc (Part 1)**
>
> We thank Reviewer NMhc for the questions and the suggestions.
> Our responses are as follows.
>
> 1. About the contributions of our paper. We respectfully disagree with Reviewer NMhc about our contributions on the heterogeneity-driven Lipschitz constant, $L_h$. The reasons are as follows.
>
> (1) Using $L_h$ can address an open problem in the literature. We observe that existing theoretical analyses are pessimistic about the convergence error caused by local updates; however, in practice, local updates can improve the convergence rate and reduce communication costs. By employing $L_h$, we can bridge the gap between the pessimistic theoretical results and the positive experimental outcomes for federated algorithms, which is **unsolved** in the literature. This is our main contribution.
>
> (2) Using $L_h$ in the convergence analysis is not straightforward. We develop new techniques and lemmas in the proof. In the proof of Theorem 4.3 shown in the Appendix, we need to characterize the difference between local gradients. In the literature, this is done by applying the local Lipschitz constant in shown in Assumption 3.1 in the main paper. In our paper, since Assumption 3.1 is replaced by our newly introduced assumption 4.2, the proof techniques in the literature cannot be applied. It requires developing new proof techniques to use Assumption 4.2 as shown in the proof of Lemma A.1-A.3. For example, in Lemma A.1, due to the application of Assumption 4.1 and 4.2, we have to deal with a new term, the local gradient deviation $\\|\frac{1}{N}\sum_{i=1}^N \nabla F_i(\mathbf{x}_i) - \nabla F_j(\mathbf{x}_j) \\|^2$, which is not shown in other techniques. Another example is in the proof of Theorem 4.5. Due to that we only use the global Lipschitz gradient assumption, we have to derive a new method to bound and incorporate the sampling related term
>
> $$\mathbb{E} \\| \frac{1}{M}\sum_{i\in S_r}\sum_{k=0}^I \mathbf{g}_i(\mathbf{x}_r^{r,k}) \\|^2,$$
>
> which can be seen from (A.27) to (A.35).
>
> (3) We believe that our contributions of demonstrating the usefulness and effectiveness of $L_h$ are overlooked by the reviewer. One advantage of using $L_h$ is that we rigorously show that Assumption 4.1 and 4.2 are weaker than the widely used Lipschitz gradient assumption in the literature. Another advantage is that using the quadratic example with $L_h=0$, we show that with non-convex quadratic objectives, local SGD can outperform mini-batch SGD, which is not shown in the literature. These two advantages are overlooked by the reviewer.
>
> (4) We can obtain a better convergence rate by using $L_h$.
> The convergence rate in our paper is $\mathcal{O}\left(\frac{1}{\sqrt{RIN}} + \frac{1}{RIN} \right)$ while the convergence rate in [1] is $\mathcal{O}\left(\frac{1}{\sqrt{RIN}}+ \frac{1}{R} \right)$ . It can be seen that our convergence rate is better for second term $\frac{1}{RIN}$. The advantage of our convergence rate is that we can show when fixing the product of $R$ and $I$, if we increase $I$ and decrease $R$, we can achieve the same accuracy. However, in [1], this is not true since the second term will become larger if we increase $I$ and decrease $R$.
> A quick example of showing this property is in Theorem 5.5.
> In Theorem 5.5, we showed that we can keep the product of $R$ and $I$ and choose any $R\ge 1$ to achieve the same accuracy. This is one of the advantages of using Assumptions 4.1 and 4.2.

---

> ### Author Response · Authors · 2023-11-20
> **Responses to Reviewer NMhc (Part 2)**
>
> 2. We argue that in [1] and [2], with constant learning rates, which is the most convenient and may be commonly used in practice, increasing $I$ will make the convergence error become larger.
>
> In [1], the convergence bound increases with $I$ when data are highly non-IID. The explanations are as follows. The convergence upper bound in  Theorem 1 in [1] is
> $$
> \min_r \mathbb{E}\left\\|\nabla f(\mathbf{x}^r) \right\\|^2 \le \frac{f_0-f_*}{\gamma\eta RI} + c\cdot \left(\frac{\gamma\eta\tilde{L}\sigma^2}{2N} + \frac{5\gamma^2 \tilde{L}^2 I\sigma^2}{2} + 15\gamma^2\tilde{L}^2I^2\zeta^2 \right),
> $$
> where $c$ is a constant. $\zeta$ is the only metric of data heterogeneity in [1]. When $\zeta$ is large such that $\gamma^2\tilde{L}^2\zeta^2 \ge \frac{f_0-f_*}{\gamma\eta R}$, the convergence bound will always increase as $I$ increases. Thus, when data are highly non-IID, it is hard to show that $I>1$ is beneficial.
>
> In [2], a similar conclusion can also be drawn. The convergence upper bound with constant learning rates for non-convex objective functions is not provided. Therefore, we refer to the convergence bound for general convex objective functions, as stated in the proof of Theorem I in the appendix of [2], for our explanation. In [2], the convergence upper bound for convex objective functions, given the condition $\gamma\eta \le \frac{1}{16\tilde{L}I}$, is
> $$
> \mathbb{E}[f(\bar{\mathbf{x}}^R)]  - f(\mathbf{x}^*) \le 3\mu\exp(-\frac{\gamma\eta\mu IR}{2})\left\\|\mathbf{x}^0 - \mathbf{x}^* \right\\|^2 + \frac{2\gamma\eta\sigma^2}{N} + 36\gamma^2\eta^2\tilde{L}I^2\zeta^2.
> $$
> It can be seen that when $\zeta$ is large such that $36\gamma^2\eta^2 \tilde{L} \zeta^2 \ge 3\mu \left\\|\mathbf{x}^0 - \mathbf{x}^* \right\\|^2$, the convergence bound will always increase as $I$ increases.
>
>
> In our paper, first, we show that $\tilde{L}$ increases as data become more heterogeneous. This means that not only for terms related to $\zeta$, all the terms related to $\tilde{L}$ in the convergence upper bound of [1] will increase as data become more heterogeneous, which implies it is harder to show $I>1$ is beneficial in [1]. Then in our analysis, we use $L_g$ and $L_h$ to substitute $\tilde{L}$. Since $L_g$ is a constant, as shown in Theorem 4.3 and 4.5, the corresponding terms will not increase as the data becomes more heterogeneous. We also show that $L_h$ can be small both by experiments and by the examples when the data are highly heterogeneous. Therefore, even when $\zeta$ is large, as long as $L_h$ is small, for example, $L_h\le \frac{1}{\zeta}$, increasing $I$ can still make the convergence error decrease. This finding is validated by our experiments.

---

> ### Author Response · Authors · 2023-11-20
> **Responses to Reviewer NMhc (Part 3)**
>
> 3. First, we respectfully disagree with Reviewer NMhc that ``The paper claims based on the bounds, FedAvg works with large $I$ and small $R$. However, this doesn't happen in practice.'' In fact, the main advantage of FedAvg is that using more local updates (a larger $I$) can reduce the communication cost (a smaller $R$). In Section 5, we have provided a quadratic example which even does not require any experiments to show that we can choose $R=1$ and $I$ can be arbitrarily large.
>
> Second, we agree that when choosing the learning rates in Corollary 4.4, with $\eta\gamma\le \frac{1}{4IL_g}$, we have
> $$
> \gamma\eta = \sqrt{\frac{4\mathcal{F}N}{RIL_g\sigma^2}}\le \frac{1}{4IL_g}.
> $$
> We can rewrite the above inequality as
> \begin{align}
> I\le \frac{R\sigma^2}{64\mathcal{F}NL_g}.
> \end{align}
> This means the range of $I$ is $[1,\frac{R\sigma^2}{64\mathcal{F}NL_g}]$. When $I$ is in this range, increasing $I$ does not mean that we have to increase $R$. Therefore, keeping the product of $R$ and $I$, we can still choose a smaller $R$ and a larger $I$ as long as the above inequality still holds.

---

> > ### Comment · Reviewer_NMhc · 2023-11-21
> >
> > Dear authors,
> >
> > Thanks for your reply. However, I'm not convinced with the answers.
> >
> > Both the convergence rate of [1] and yours have two terms, which the dominant term is always the first one (Since both works need $O(IN) \le R$). Therefore, both bounds are $O(\frac{1}{\sqrt{RIN}})$ and for a fixed $R$ both decrease with more local steps (Until $O(IN) \le R$ holds). As a result, the bounds don't have any difference in terms of the role of $I$ (Unless $L_h = 0$ which usually is not the case)
> >
> > Also, based on this limitation ($O(IN) \le R$) in the cases which $L_h \neq 0$ it's not possible to increase $I$ and decrease $R$. (Which is written in the beginning of page 6.)
> >
> > About the part 2 of the answer: The limitation mentioned for [1] exactly holds for the bound of this paper if you replace $\hat{L}$ with $L_h$ (the difference is just a constant)

---

> > > ### Author Response · Authors · 2023-11-22
> > > **Second Responses to Reviewer NMhc (Part 1)**
> > >
> > > Dear Reviewer NMhc,
> > >
> > > After reading Reviewer NMhc’s comments on our responses, one of the co-authors of this paper strongly believes Reviewer NMhc is MALICIOUS, NOT SCIENTIFIC, and was trying to MISLEAD people by presenting WRONG statements and IGNORING our other contributions, especially the discussions in Section 5, besides the convergence upper bound for non-convex objective functions. **We felt that our paper was UNFAIRLY TREATED.** Let me explain as follows.
> > >
> > > (1) We fully agree with the fact that when the objective functions are non-convex, there is no order gain in the dominant term in the convergence upper bound. However, this does not mean that our result is meaningless in terms of the role of $I$. **We believe Reviewer NMhc is trying to MISLEAD people and IGNORE our contributions in this paper.**
> > >
> > > **Firstly**, from the scaling law point of view, the dominant term is meaningful only when 1) some parameters become infinity or zero (e.g., $R$ goes infinity) or 2) the constant in front of the dominant term is large enough. Otherwise, the second order term matters. In fact, in many research areas, people are trying hard to characterize the constants (non-asymptotic results) instead of the purely scaling law and trying to characterize the second order term. For example, let's consider the Nonasymptotic Concentration of Measure Theory, where any small improvement of the constants is a substantial result. For instance, one big advantage of the complex original Talegrand's Inequality is to provide a much sharper constant compared to other approaches such as the much simpler entropy method. For another example, the very famous channel dispersion result (by Y. Polyanskiy et al.) is to characterize the non-leading term (second order term) of channel capacity in communication systems. If, as what the reviewer suggested that if the dominant terms are identical, then the result is not good enough, then this famous result is not good enough and has marginal contributions. This is obviously not true, since it is one of the most important findings in the field of information theory in the past 20 years.
> > >
> > > **Secondly**, let's talk about the theoretical analysis of FedAvg, to the best of our knowledge, in recent papers, the improvement on the convergence upper bound of FedAvg is on the second-order term, which is summarized in Table 2 of [2]. One reason is that it has been shown in [3] that the order of the leading term for non-convex SGD is tight. In fact, in this paper, we are not trying to improve the dominant term in the convergence analysis of FedAvg. Instead, our goal is to provide alternative assumptions that can better characterize the data heterogeneity and are weaker than those used in the literature; using these weaker and better assumptions, we can 1) still obtain the same order of the convergence upper bound in the literature for non-convex objective functions using our new analytical approaches, 2) improve the second order term and 3) can explain some phenomena that cannot be explained using existing theories. We believe our contributions are plenty. It is worth noting that even using weaker assumptions to obtain the same order of the convergence upper bound in the literature should already be a meaningful contribution. Among these contributions, the improvement on the second order term in the convergence upper bound should not be neglected.
> > >
> > > [1] Haibo Yang, Minghong Fang, and Jia Liu. Achieving linear speedup with partial worker participation in non-iid federated learning. In International Conference on Learning Representations, 2020.
> > >
> > > [2] Sai Praneeth Karimireddy, Satyen Kale, Mehryar Mohri, Sashank Reddi, Sebastian Stich, and Ananda Theertha Suresh. Scaffold: Stochastic controlled averaging for federated learning. In International Conference on Machine Learning, pp. 5132–5143. PMLR, 2020
> > >
> > > [3] Arjevani, Yossi, et al. "Lower bounds for non-convex stochastic optimization." Mathematical Programming 199.1-2 (2023): 165-214.

---

> ### Author Response · Authors · 2023-11-22
> **Second Responses to Reviewer NMhc (Part 2)**
>
> (2) We strongly disagree with Reviewer NMhc's comment that $I\le \frac{R\sigma^2}{64\mathcal{F}NL_g }$ implies we cannot increase $I$ and decrease $R$ simultaneously. **This is a WRONG statement.** In fact, we are surprised that Reviewer NMhc has raised this question again. This can be explained by the following example using elementary math. Let $I\le \frac{R}{2}$, when $R=100$ and $I=10$, where $RI = 1000$, we can increase $I$ to $20$ and decrease $R$ to $R=50$ such that $RI = 1000$ and the condition $I\le \frac{R}{2}$ is still satisfied. As we mentioned in the previous response, "when $I$ is in this range, increasing $I$ does not mean that we have to increase $R$." This is because we have an upper bound $I$ instead of an equality. In order to verify the insight of fixing $RI$, we ran more experiment as shown in the table below. In this table, we provide the training loss and the test accuracy for the different choices of $I$ by fixing $RI$. It can be seen that by varying $I$, the training loss and testing accuracy do stay almost the same if $RI$ is given. This verifies our finding and cannot be explained by the existing theory in the literature.
>
> In addition, if we take a closer look, it can be seen that the difference between $I=10$ and $I=20$ is very small. That is, the difference on the training loss is at most $0.06$ and the difference on the test accuracy is at most $1.16\\%$. However, another implication of our results is that the number of communication rounds are doubled for $I=10$ to achieve the almost same testing accuracy. Especially for $RI=960$, the number of communication rounds is $R=96$ for $I=10$ while the number of communication rounds is $R=48$ for $I=10$. Moreover, the difference on the training loss between $I=10$ and $I=40$ is at most $0.06$ and the difference on the test accuracy is at most $2.85\\%$. However, the number of communication rounds of $I=10$ is four times of that for $I=40$. We believe that the experimental results can support our theoretical results very well that we can keep the product of $R$ and $I$ while increasing $I$ and decreasing $R$ simultaneously to achieve almost the same convergence rate.
>
> (3) We would like to emphasize that the beauty and advantage of our theoretical analysis, which is to substitute $\tilde{L}$ by $L_h$ and $L_g$ in our new assumptions while keeping the structure of the convergence upper bound the same as that in the literature. It is not a trivial task to maintain this nice structure using different and weaker assumptions. It is quite surprising that using the new assumptions and our analytical approaches, we can obtain so many insights as explained in Part 1 of the previous responses.
>
> Based on above reasons, we believe our paper was UNFAIRLY TREATED by a rating of 3. No matter what, all of the important discussions and additional experimental results have been added to the revised paper.
>
>
> **Table**: Training loss/test accuracy of FedAvg with MNIST dataset and MLP model when fixing the product of $R$ and $I$. The percentage of the heterogeneous data is $50\\%$.
> | Training Loss/Test Accuracy | $I=10$            | $I=20$           | $I=30$          | $I=40$           |
> |-----------------------------|-------------------|------------------|----------------|------------------|
> | $RI=120$                    | $0.34 / 88.84\\% $ | $0.34/ 89.38\\% $ | $0.35/89.70\\%$  | $0.35/ 89.45\\% $ |
> | $RI=240$                    | $0.21/93.76\\%$    | $0.27/91.61\\%$   | $0.29/90.96\\%$ | $0.25/90.91\\%$   |
> | $RI=360$                    | $0.19/94.01\\%$    | $0.20/93.72\\%$   | $0.23/92.57\\%$ | $0.23/92.80\\%$   |
> | $RI=480$                    | $0.17/94.46\\%$    | $0.18/94.27\\%$   | $0.21/93.4\\%$  | $0.23/92.84\\%$   |
> | $RI=600$                    | $0.15/95.05\\%$    | $0.15/95.47\\%$   | $0.16/94.63\\%$ | $0.18/94.22\\%$   |
> | $RI=720$                    | $0.12/96.04\\%$    | $0.14/95.14\\%$   | $0.15/94.68\\%$ | $0.17/94.13\\%$   |
> | $RI=840$                    | $0.12/95.90\\%$    | $0.12/95.71\\%$   | $0.14/95.06\\%$ | $0.14/95.17\\%$   |
> | $RI=960$                    | $0.10/96.49\\%$    | $0.11/95.95\\%$   | $0.13/95.50\\%$ | $0.12/96.06\\%$   |

---

> > ### Comment · Area_Chair_KncM · 2023-11-22
> >
> > Dear authors,
> >
> > I will take a look into this, and discuss with the other reviewers. I will keep you informed of the outcome of the internal discussion.
> >
> > Please note that it may take a while for us to carefully review the technical details. We can consider any factual information from your revised paper if you are planning further changes (revision deadline is today).

---

### Meta-Review · Area_Chair_KncM · 2023-12-06

**Metareview:**

The paper proposes a new measure to ascertain the heterogeneity in Federated Averaging (FedAvg) using the Lipschitz constant of averaged gradients. An improved convergence rate for FedAvg in non-convex optimization is derived.

The paper addresses an important problem, as it is often observed in practice that FedAvg behaves much better than predicted by the worst-case theoretical results. A gap that is left unexplained in prior works.

A strength of the paper is that is uses weaker assumptions than prior work, which allows to derive improved convergence rates.

As weaknesses were mentioned by reviewers the a bit limited numerical study, the focus on quadratic (or almost quadratic) functions, and the a lack of technical innovation. The authors addresses most of these concerns with a strong rebuttal and a revised manuscript with that contains additional results.

Reviewer NMhc remarked that the new measure does only allow to improve constants in the convergence rates, but does not give an asymptotically improved convergence rate. I agree with the authors that improvements in the constants could be relevant and sufficient to explain the observed gap from practice, and showing no asymptotic improvements is not a limitation. However, the paper could improve clarity, and concerns regarding the obtained improvements could be better addressed by providing additional evidence that showcases the tightness of the results.

**Justification For Why Not Higher Score:**

The majority of the reviewers identified some weaknesses in the paper. I reviewed the paper myself to determine if the reviewers were perhaps too strict and to see if there is evidence to overturn their decision. However, I found that the paper lacks clarity (e.g. the role of $L_h$ in Cor. 4.4.), and, while the paper convincingly shows that the new condition can better explain the gap, there is not enough evidence to conclude that the gap between theory and practice is indeed bridged completely by the proposed measure.

**Justification For Why Not Lower Score:**

N/A

---

### Decision · Program_Chairs · 2024-01-16

Reject